# Mitochondrial integrated stress response controls lung epithelial cell fate

SeungHye Han[1✉], Minho Lee[2], Youngjin Shin[2], Regina Giovanni[1], Ram P. Chakrabarty[1], Mariana M. Herrerias[1], Laura A. Dada[1], Annette S. Flozak[1], Paul A. Reyfman[1], Basil Khuder[1], Colleen R. Reczek[1], Lin Gao[3], José Lopéz-Barneo[3], Cara J. Gottardi[1], G. R. Scott Budinger[1] & Navdeep S. Chandel[1,4✉]

Alveolar epithelial type 1 (AT1) cells are necessary to transfer oxygen and carbon dioxide between the blood and air. Alveolar epithelial type 2 (AT2) cells serve as a partially committed stem cell population, producing AT1 cells during postnatal alveolar development and repair after influenza A and SARS-CoV-2 pneumonia[1–6]. Little is known about the metabolic regulation of the fate of lung epithelial cells. Here we report that deleting the mitochondrial electron transport chain complex I subunit $Ndufs2$ in lung epithelial cells during mouse gestation led to death during postnatal alveolar development. Affected mice displayed hypertrophic cells with AT2 and AT1 cell features, known as transitional cells. Mammalian mitochondrial complex I, comprising 45 subunits, regenerates $NAD^+$ and pumps protons. Conditional expression of yeast NADH dehydrogenase (NDI1) protein that regenerates $NAD^+$ without proton pumping[7,8] was sufficient to correct abnormal alveolar development and avert lethality. Single-cell RNA sequencing revealed enrichment of integrated stress response (ISR) genes in transitional cells. Administering an ISR inhibitor[9,10] or $NAD^+$ precursor reduced ISR gene signatures in epithelial cells and partially rescued lethality in the absence of mitochondrial complex I function. Notably, lung epithelial-specific loss of mitochondrial electron transport chain complex II subunit $Sdhd$, which maintains $NAD^+$ regeneration, did not trigger high ISR activation or lethality. These findings highlight an unanticipated requirement for mitochondrial complex I-dependent $NAD^+$ regeneration in directing cell fate during postnatal alveolar development by preventing pathological ISR induction.

During mammalian lung development, the lung traverses through morphologically distinct developmental stages characterized by the progressive commitment of airway and alveolar epithelial progenitors to mature cell fates[11]. While the conducting airways develop prenatally during branching morphogenesis, development of the alveoli begins perinatally but is incomplete at birth, continuing for four to five weeks in mice and at least three years in humans. Although many of the molecular and transcriptional signals necessary for lung development have been elucidated[11,12], the mechanisms by which metabolic cues may direct these processes remains unknown.

Development across organs is characterized by early reliance on glycolysis that progressively shifts toward oxidative phosphorylation with support from fatty acid oxidation[13]. Consistent with this paradigm, the lung epithelium expresses high levels of glycolytic genes during embryonic development, with increased expression of genes involved in oxidative phosphorylation at later postnatal stages[14] (Extended Data Fig. 1a,b). We sought to determine whether a functional mitochondrial electron transport chain (ETC) was necessary for lung development

by ablating mitochondrial complex I subunit NADH dehydrogenase (ubiquinone) iron-sulfur protein 2 ($Ndufs2$) in the distal lung epithelium during development. We crossed surfactant protein C-Cre (SFTPC-Cre)[15] mice with $Ndufs2^{fl/-}$ mice[16] and Cre-reporter mice ($ROS A26Sor^{CAG-tdTomato}$), which are hereafter referred to as NDUFS2 conditional knockout (cKO) mice ($Ndufs2^{fl/-} SFTPC-Cre;ROSA26Sor^{CAG-tdTomato}$). Because $Sftpc$ is expressed in common distal lung epithelial progenitors at embryonic day (E)10.5 in mice, genes harbouring floxed alleles ($Ndufs2$) and a $loxP$-STOP-$loxP$ cassette ($tdTomato$) are deleted and expressed, respectively, in distal lung epithelial cell populations (club cells, alveolar epithelial type 2 (AT2) cells and alveolar epithelial type 1 (AT1) cells) in these animals upon Cre-mediated recombination[15].

NDUFS2 is a nuclear-encoded core subunit of mitochondrial complex I that is essential for its enzymatic activity. Depleting NDUFS2 causes mitochondrial complex I deficiency (Fig. 1a) and its global depletion results in embryonic lethality[17]. NDUFS2 cKO mice were viable despite decreased abundance of NDUFS2 protein (Fig. 1b) and decreased basal and coupled oxygen consumption rates (OCR) in lung epithelial cells

[1]Division of Pulmonary and Critical Care Medicine, Department of Medicine, Northwestern University, Chicago, IL, USA. [2]Department of Life Science, Dongguk University-Seoul, Goyang-si, Republic of Korea. [3]Instituto de Biomedicina de Sevilla (IBiS), Hospital Universitario Virgen del Rocío, CSIC, Universidad de Sevilla, Seville, Spain. [4]Biochemistry and Molecular Genetics, Northwestern University, Chicago, IL, USA. ✉e-mail: shan@northwestern.edu; nav@northwestern.edu

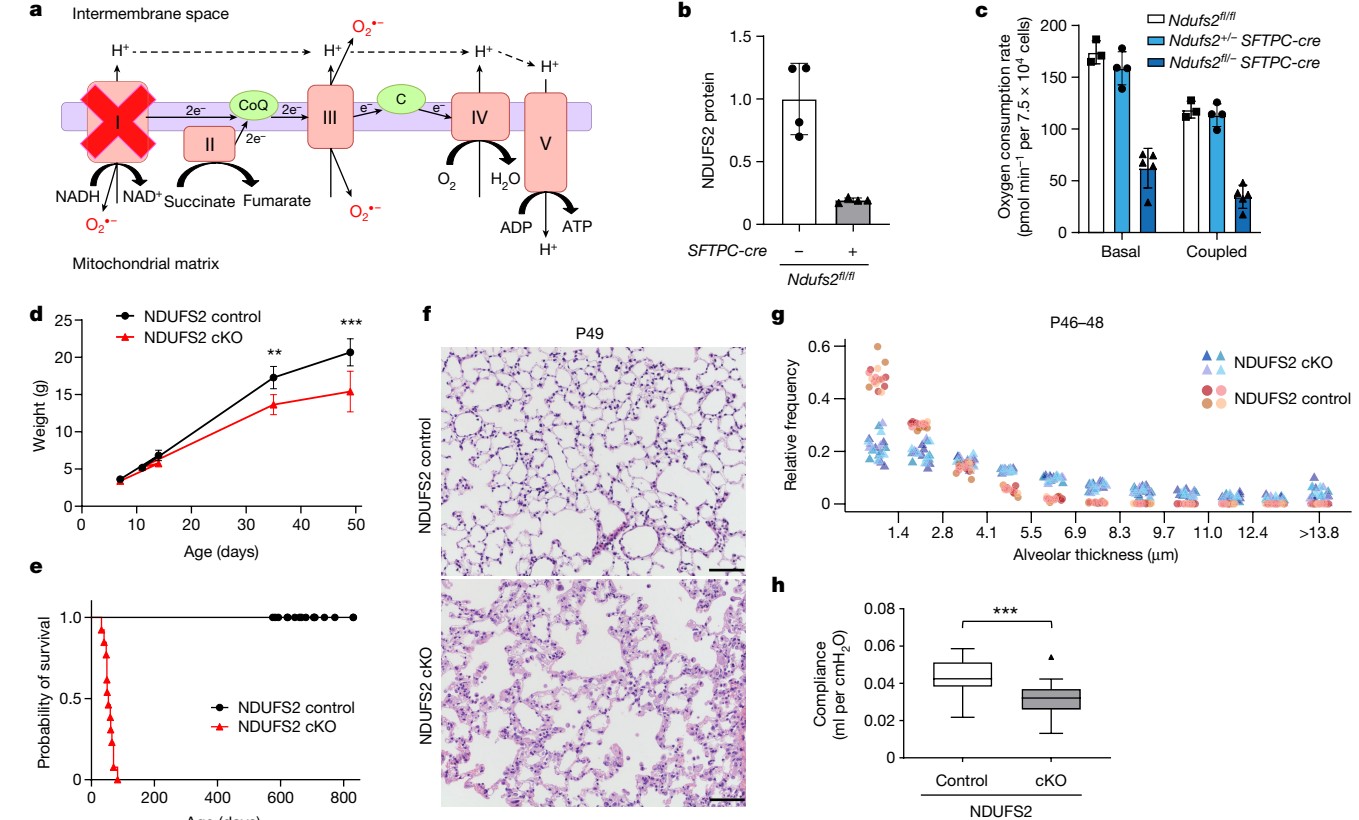

**Fig. 1 | Mitochondrial complex I in lung epithelial cells is necessary for postnatal lung development. a**, Schematic of the mitochondrial ETC in lung epithelial cells of NDUFS2 cKO mice. **b**, Immunoblot analysis of NDUFS2 protein normalized to vinculin in lung epithelial cells isolated from 11-day-old mice. Data represent mean ± s.d. ($n$ = 4 mice in each genotype with technical replicates). **c**, Basal and coupled OCR of lung epithelial cells isolated from 43- to 46-day-old mice. Data represent mean ± s.d. ($Ndufs2^{fl/fl}$ $n$ = 3; NDUFS2 control $n$ = 4; NDUFS2 cKO $n$ = 5 mice with technical replicates). **d**, Body weight in grams (control $n$ = 34; cKO $n$ = 18 mice). Data represent mean ± s.d. **$P$ = 0.0040, ***$P$ = 0.0005 by Mann–Whitney test. **e**, Survival of NDUFS2 control ($n$ = 21) and NDUFS2 cKO ($n$ = 13) mice ($P$ < 0.0001 by log-rank test). **f**, Representative images of lung histology on postnatal day 49 (haematoxylin and eosin stain). Scale bar, 100 µm. **g**, The frequency distribution of alveolar thickness measured in haematoxylin and eosin-stained lung histology of 46- to 48-day-old mice ($n$ = 4 mice, two males and two females per genotype). Four to six randomly selected fields of view from each mouse were evaluated. The $x$ axis shows alveolar thickness bins and the $y$ axis shows the number of alveolar pixels that belong to the respective alveolar thickness bin normalized to the total alveolar pixel count in the image. Each animal is represented by its own colour. Statistical significance for genotype was calculated based on $F$-test for a linear model ($P$ = 4.56 × 10$^{-5}$). **h**, Box plots of lung compliance in 46- to 49-day-old mice (control $n$ = 33; cKO $n$ = 24 mice with technical replicates), $P$ < 0.0001 by Mann–Whitney test.

(Fig. 1c), compared with $Ndufs2^{+/-}$ $SFTPC$-$Cre$;$ROSA26Sor^{CAG\text{-}tdTomato}$ mice (hereafter referred to as NDUFS2 control mice). We have previously reported that lung development and ageing at two years of life in $Ndufs^{+/-}$ mice is similar to what is observed in wild-type mice[17]. NDUFS2 cKO mice displayed diminished postnatal weight gain and died between five to nine weeks after birth (median week 7) (Fig. 1d,e and Extended Data Fig. 1c,d).

Abnormalities at necropsy in NDUFS2 cKO mice were limited to the lung, where the alveolar airspaces were filled with a pink, homogenous material negative for periodic acid–Schiff stain (Extended Data Fig. 1e–g). These findings suggest death from respiratory failure. NDUFS2 cKO mice harvested before death showed hypercellular areas with thickened alveolar walls interspersed between enlarged alveolar spaces (Fig. 1f,g and Extended Data Fig. 1h–q) and disrupted spatial organization between alveolar epithelial cells and endothelial cells and/or fibroblasts (Extended Data Fig. 2a,b), indicating impaired alveolar development. Lung compliance, which developmentally reflects both lung size and lung elastic recoil, was significantly decreased in NDUFS2 cKO mice compared with NDUFS2 control mice (Fig. 1h). In contrast, we could not detect histologic differences between NDUFS2 cKO and NDUFS2 control mice harvested at E13.5 and postnatal day (P) 0 (Extended Data Fig. 2c–g). These findings suggest that structural abnormalities of the

lungs in NDUFS2 cKO mice develop postnatally and are largely limited to the alveolar space.

Surprisingly, we observed increased cellularity in postnatal NDUFS2 cKO lungs compared with NDUFS2 control lungs. NDUFS2 cKO lungs had an increased number of hypertrophic cells expressing an AT2 marker, surfactant protein C and/or the AT1 marker, podoplanin, compared with NDUFS2 control lungs (Extended Data Fig. 1n–q). However, inflammatory cell infiltration and increased apoptosis were absent in NDUFS2 cKO lungs compared with NDUFS2 control lungs (Extended Data Fig. 1h,i,k,l,r,s). Instead, Ki67 expression, a proliferation marker, was increased in NDUFS2 cKO lungs compared with NDUFS2 control mice including in the $Sftpc$ lineage ($tdTomato$)-positive cells (Extended Data Fig. 1j,m,t and Extended Data Fig. 4a). These findings argue against a bioenergetic crisis in NDUFS2 cKO mice causing epithelial cell death. The podoplanin-positive cells in the NDUFS2 cKO mice were thicker and rounder than those in the NDUFS2 control mice (Extended Data Fig. 1n–q). Likewise, some $Sftpc$-expressing cells in NDUFS2 cKO mice did not have the classic cuboidal shape of AT2 cells and instead exhibited a linear and thin shape more typical of AT1 cells (Extended Data Fig. 2b inset and Extended Data Fig. 6f' inset). These findings suggest that the differentiation of epithelial cells in NDUFS2 cKO lungs may be arrested in an intermediate transitional state expressing both AT2

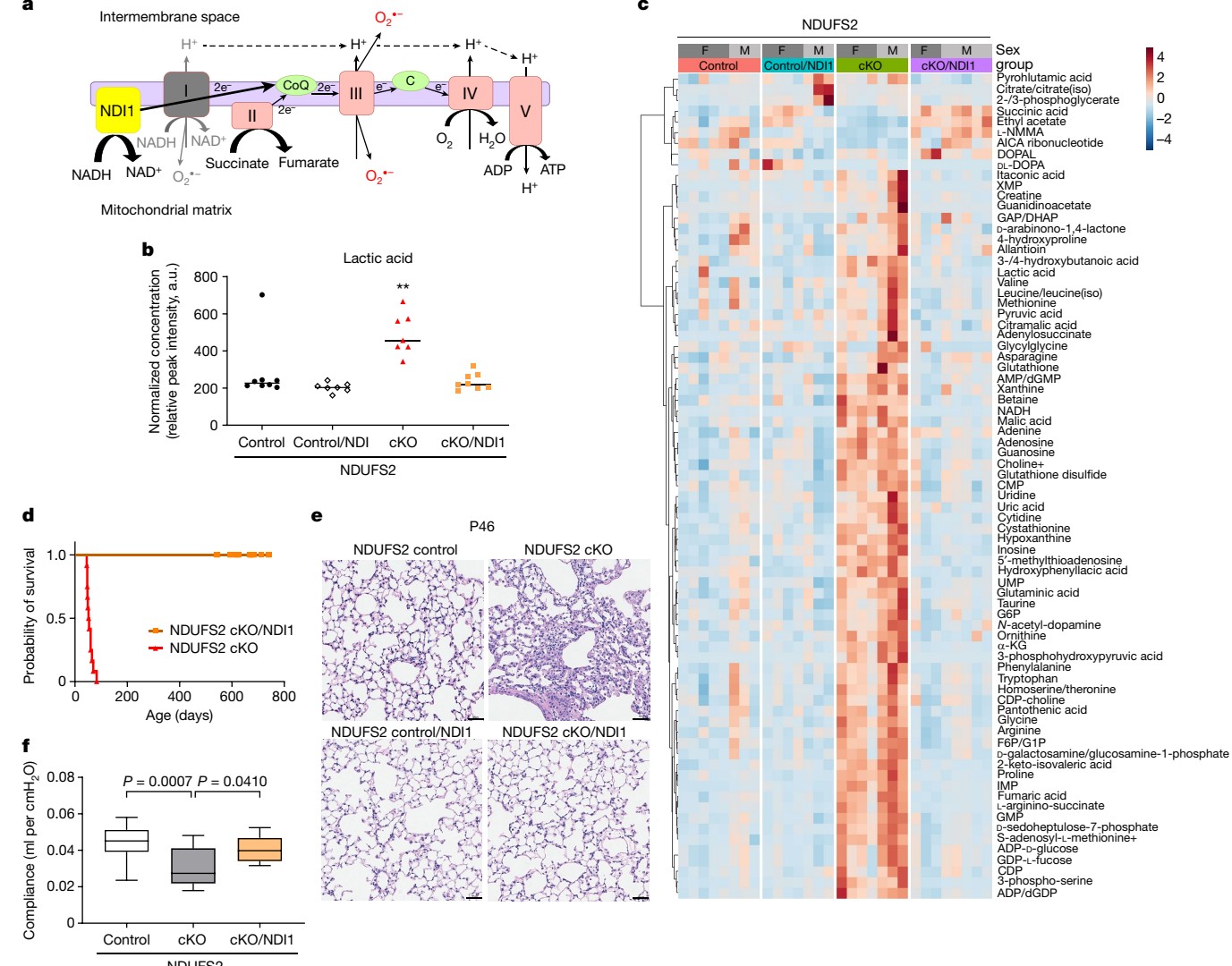

**Fig. 2 | Expression of the yeast NDI1, an alternative NADH dehydrogenase, in lung epithelial cells reverses abnormal postnatal alveolar development in NDUFS2 cKO mice. a**, Schematic of the mitochondrial ETC with ectopic NDI1. **b,c**, Metabolomics analysis of lung epithelial cells isolated from 35-day-old mice (NDUFS2 control $n = 8$; NDUFS2 control/NDI1 $n = 7$; NDUFS2 cKO $n = 7$; NDUFS2 cKO/NDI1 $n = 8$ mice). **b**, Relative abundance of lactic acid. Lines represent median. $P = 0.0013$ by Kruskal–Wallis test **b**. **c**, The heat map displays the relative abundance of significantly changed metabolites **c**. α-KG, α-ketoglutarate; AICA, 5-aminoimidazole-4-carboxamide; DHAP, dihydroxyacetone phosphate; DOPAL, 3,4-dihydroxyphenylacetaldehyde;

F6P, fructose-6 phosphate; G1P, glucose-1 phosphate; G6P, glucose-6 phosphate; GAP, glyceraldehyde 3-phosphate; IMP, inosine monophosphate; L-NMMA, $N^G$-monomethyl-L-arginine; XMP, xanthosine monophosphate. **d**, Survival of NDUFS2 cKO ($n = 12$) and NDUFS2 cKO/NDI1 ($n = 22$) mice ($P < 0.0001$ by log-rank test). **e**, Representative images of littermates' lung histology (haematoxylin and eosin stain) in 46-day-old mice. Scale bar, 50 μm. **f**, Box plots of lung compliance of 46- to 50-day-old mice (NDUFS2 control $n = 21$; NDUFS2 cKO $n = 11$; NDUFS2 cKO/NDI1 $n = 12$ mice with technical replicates). $P = 0.0016$ by one-way analysis of variance. Adjusted $P$ values by Šídák's multiple comparisons test in the graph. a.u., arbitrary units.

and AT1 canonical cell markers. Taken together, our findings suggest a requirement of mitochondrial complex I for postnatal alveolar development.

Mammalian mitochondrial complex I is solely responsible for regenerating mitochondrial NAD$^+$, which is necessary for maintaining oxidative tricarboxylic acid cycle function. It also pumps protons across the mitochondrial inner membrane, contributing to the proton gradient necessary for ATP synthesis. Additionally, it generates superoxide that can control physiology and pathology in some biological contexts. The *Saccharomyces cerevisiae* single-subunit alternative internal NADH dehydrogenase (NDI1) protein regenerates NAD$^+$ by passing electrons to ubiquinone without proton pumping or producing superoxide[7,8] (Fig. 2a). To determine the necessity of these distinct functions of mammalian mitochondrial complex I during lung development, we crossed *Ndufs2[fl/−] SFTPC-Cre* mice with mice that have an *NDI1[LSL]* targeting

construct inserted into the mouse *Rosa26* locus[7]. The resulting mice, which are hereafter referred to as NDUFS2 cKO/NDI1 mice, conditionally delete *Ndufs2* but express yeast NDI1 in the distal lung epithelium upon *Cre*-mediated recombination. Expressing *NDI1* in normal lung epithelium (*SFTPC-Cre;NDI1[LSL]*) does not disrupt lung development or physiology (Extended Data Fig. 3a,b). NDI1 protein expression in lung epithelial cells from NDUFS2 cKO/NDI1 mice almost completely rescued the metabolite dysregulation observed in NDUFS2 cKO mice. NDI1-expressing NDUFS2 cKO lung epithelial cells have a comparable lactate level and ratio of NADH/NAD$^+$ to NDUFS2 control lung epithelial cells (Fig. 2b,c and Extended Data Fig. 3c). NDI1 expression in NDUFS2 cKO mice prevented mortality (Fig. 2d), reversed the histologic abnormalities in alveolar structures (Fig. 2e and Extended Data Fig. 3d) and restored lung compliance (Fig. 2f) a level that was indistinguishable from NDUFS2 control mice. NDUFS2 cKO/NDI1 mice did not have

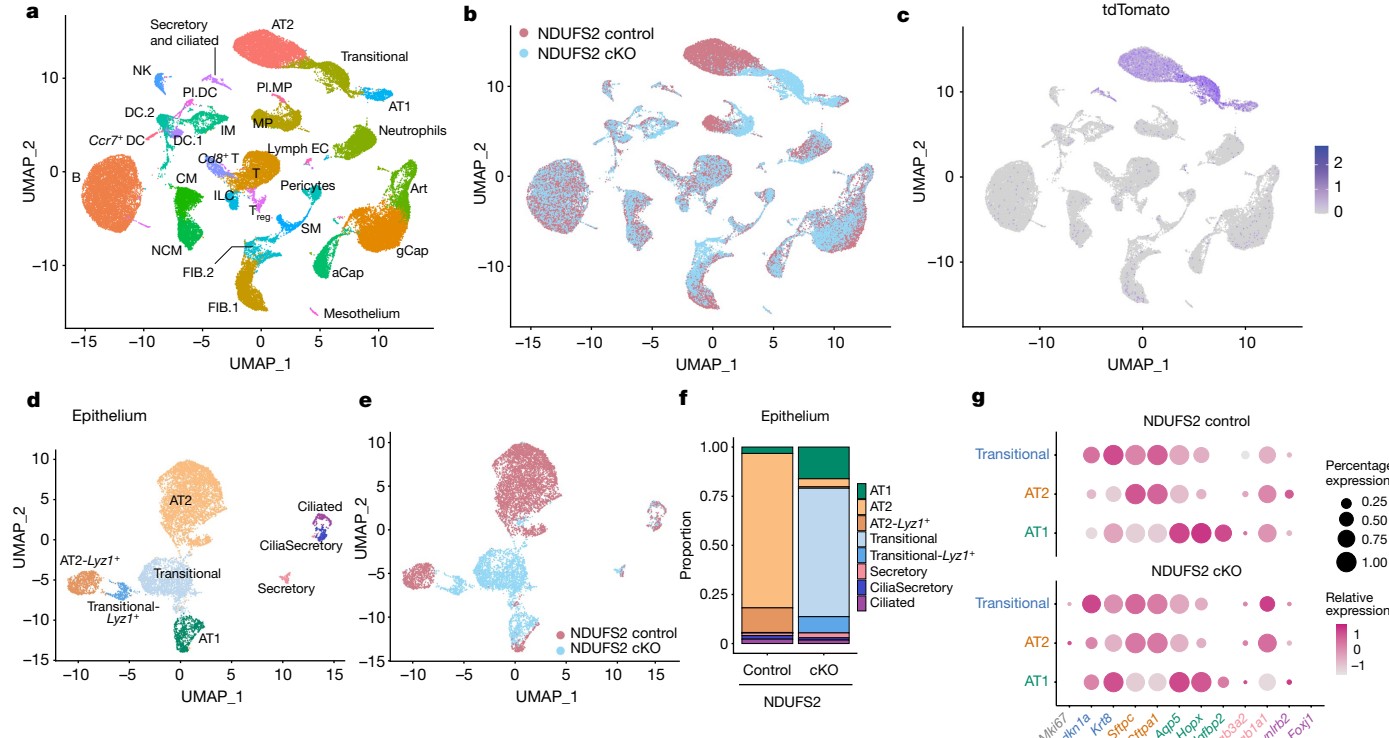

**Fig. 3 | A distinct epithelial population of transitional cells emerges during postnatal lung development in NDUFS2 cKO mice. a**, Uniform manifold approximation and projection (UMAP) plot showing single-cell RNA-seq analysis of 57,886 cells isolated from 21-day-old mouse lungs ($n = 4$ mice (two males and two females) in each genotype). aCAP, alveolar capillary endothelial cells; Art, arterial endothelial cells; B, B cells; CM, classical monocytes; DC.1, dendritic cells 1; DC.2, dendritic cells 2; FIB.1, fibroblasts 1; FIB.2, fibroblasts 2; gCAP, general capillary endothelial cells; ILC, innate lymphoid cells; IM, interstitial macrophages; lymph EC, lymphatic endothelial cells; MP, macrophages; NCM, non-classical monocytes; NK, natural killer cells; Pl.DC, proliferating dendritic cells; Pl.MP, proliferating macrophages; SM, smooth muscle cells; T, T cells; T$_{reg}$, regulatory T cells (Extended Data Table 1). **b**, UMAP plot depicting cell origins with respect to the mouse

genotype. **c**, UMAP plot showing the expression of a *Sftpc* lineage tracing fluorescent marker, tdTomato. Darker colour represents higher expression. **d**, UMAP embedding of lung epithelial cells ($n = 9,322$ cells) coloured by cell type. CiliaSecretory, club cells and ciliated cells. **e**, UMAP plot depicting epithelial cell origins with respect to the mouse genotype. **f**, Bar plots demonstrating the composition of epithelial subclusters in cells from NDUFS2 control and NDUFS2 cKO mice. **g**, Marker gene expression by epithelial cell type in each mouse genotype is displayed in a dot plot, where the size of the dot indicates the proportion of cells within the cell type expressing that gene and higher expression is represented as a darker colour. In this dot plot, AT2 and AT2-*Lyz1*⁺ clusters and Transitional and Transitional-*Lyz1*⁺ clusters were merged into 'AT2' and 'Transitional', respectively.

histological or functional differences compared with NDUFS2 control mice even at 18 to 25 months of life (Extended Data Fig. 3e,f). Previous studies in cancer cells reported that ETC inhibition triggers depletion of aspartate and asparagine, decreasing cell proliferation[18–21]. However, we did not observe decreases in aspartate and asparagine in lung epithelial cells isolated from NDUFS2 cKO mice compared with those from NDUFS2 control mice (Extended Data Fig. 3g,h) and proliferation in the NDUFS2 cKO lung epithelium was preserved (Extended Data Fig. 4a). Collectively, these results indicate that mitochondrial complex I's ability to regenerate NAD⁺ is the dominant function controlling postnatal alveolar development.

To further investigate the histologic abnormalities that we observed in NDUFS2 cKO lungs, we performed single-cell RNA sequencing (RNA-seq) on whole lung single-cell suspensions isolated from both NDUFS2 cKO and control mice at P21. Clustering analysis identified 29 cell types in the whole lung (Fig. 3a and Extended Data Fig. 4b,c). *Ndufs2* was deleted in the distal epithelium in NDUFS2 cKO animals, which confirms proper *Cre*-mediated recombination in these cells (Extended Data Fig. 4d,e). Further clustering analysis with epithelial cells (that is, clusters expressing the canonical epithelial cell marker *Epcam*) identified eight distinct epithelial cell types (Fig. 3d). We observed expansion of an epithelial subpopulation characterized by high expression of *Cdkn1a*, *Krt8* and other cytokeratin genes in NDUFS2 cKO mice compared with NDUFS2 control mice (Fig 3b,e–g and Extended Data Figs. 4c and 5a,b).

These cells share several transcriptional features with intermediate epithelial cell populations described in several settings where AT2 cells are differentiating into AT1 cells, including postnatal mouse lung development[14], mouse models of lung injury[22–24], lung explants from patients with lung fibrosis, including patients with COVID-19 infection[25–27], and differentiating lung organoids derived from mouse and human AT2 cells[28,29]. Accordingly, we refer to these cells as transitional cells hereafter. Transitional cells are *Sftpc* lineage-positive cells (Fig. 3c) and they express both AT2 (*Sftpc*, *Sftpa1*) and AT1 (*Aqp5*, *Hopx*) markers (Fig. 3g and Extended Data Fig. 5b). The overall expression of cell cycle-associated genes was similar across epithelial subpopulations, although the expression of *Cdkn1a* was increased in transitional cells (Extended Data Fig. 5c). Our histologic analysis suggested that AT1 cells in NDUFS2 cKO mice were rounder and less mature than flat, thin mature AT1 cells in NDUFS2 control mice (Extended Data Fig. 1o',q'). Indeed, we found that cells assigned to the AT1 cell cluster in NDUFS2 cKO mice express higher levels of transitional cell marker genes such as *Cdkn1a*, *Krt8* and *Krt18* and a lower level of *Igfbp2*, which is known as a marker for mature, terminally differentiated AT1 cells[30], compared with those in NDUFS2 control mice (Fig 3g and Extended Data Fig. 5d–h). Thus, the ability to capture more AT1 cells from NDUFS2 cKO mice than NDUFS2 control mice in single-cell RNA-seq is likely to result from enhanced liberation of these rounder, less mature cells during tissue dissociation. (Fig 3f and Extended Data Fig. 5a). Furthermore, the loss of

mitochondrial complex I function in the distal lung epithelium resulted in the emergence of a fibroblast subpopulation (Fig. 3b and Extended Data Fig. 4c) that includes populations of fibroblasts characterized by expression of *Sfrp1* and *Timp1* (Extended Data Fig. 6). In mouse models of lung injury and repair, another group identified a similar population of cells they labelled as transitional fibroblasts[31]. Taken together, these results suggest that the loss of mitochondrial complex I in epithelial cells disrupts epithelial and mesenchymal differentiation; thus, it changes cell fate during postnatal lung development.

To identify transcriptional networks that might underlie the accumulation of transitional cells in the absence of mitochondrial complex I function, we performed bulk RNA-seq on lung epithelial cells isolated from NDUFS2 cKO and NDUFS2 control mice at P35. Gene set enrichment analysis suggested an increase in MYC signalling, oxidative phosphorylation and the unfolded protein response pathways in NDUFS2 cKO lung epithelium compared with NDUFS2 control lung epithelium (Extended Data Fig. 7a). Several groups have shown that the integrated stress response (ISR) is activated in response to mitochondrial ETC dysfunction in vitro and in vivo[18,19,32–40]. ISR activation increases phosphorylation of eukaryotic translation initiation factor 2 subunit alpha (eIF2α), which inhibits protein synthesis globally but enhances the translation of selective genes as an adaptive mechanism to counter metabolic stress. The paradoxically translated genes following ISR activation include those encoding transcription factors (ATF3, ATF4 and ATF5) and *Ddit3* (which encodes CHOP), and they can induce the expression of their own genes and genes involved in one-carbon metabolism. Chronic ISR activation can be detrimental and we recently linked activation of the ISR to the development of pulmonary fibrosis[41,42]. Here, we identified induction of the ISR following the loss of mitochondrial complex I function as a potential pathogenic and causal mechanism to explain the impairment of proper epithelial cell differentiation in NDUFS2 cKO mice. We observed an increase in the expression of ISR target genes including *Atf4*, *Atf5* and *Ddit3* in lung epithelial cells from NDUFS2 cKO mice compared with NDUFS2 control mice (Fig. 4a,b and Extended Data Fig. 7b). Analysis of our single-cell RNA-seq data revealed enrichment of *Atf* as well as other ISR target-gene transcripts in transitional cells relative to other epithelial cell populations in the lung (Fig. 4c–e and Extended Data Fig. 7c–e).

Next, we compared transitional cells from NDUFS2 cKO mice with transitional cell populations identified by other investigators. We found that postnatal transitional cells observed in a single-cell atlas of mouse lung development[14] displayed higher expression of ISR signature genes than other epithelial cell types within the atlas (Extended Data Fig. 8a). Notably, transitional cells from the NDUFS2 cKO mice showed an even higher expression of ISR target genes compared with normal postnatal transitional cells (Extended Data Fig. 8b). We integrated our single-cell RNA-seq data from NDUFS2 cKO lungs with the single-cell atlas data from wild-type lungs during postnatal development and analysed RNA velocity. RNA velocity analysis identified postnatal transitional cells that emerged during normal development as differentiating into AT1 cells, but it did not predict a similar differentiation trajectory for transitional cells from NDUFS2 cKO mice, which implies a stalled epithelial differentiation process (Extended Data Fig. 8c,d). We also performed a similar analysis using single-cell RNA-seq data from epithelial cells isolated from hyperoxia-exposed postnatal lungs[43]. AT2 cells from hyperoxia-exposed mice expressed higher levels of ISR target genes compared with the AT2 cells from normoxia-exposed lungs, but these levels were not as high as in the NDUFS2 cKO transitional cells (Extended Data Fig. 8e,f). Similarly, our RNA velocity analysis data suggests that AT2 cells from hyperoxia-exposed mice are differentiating into AT1 cells, while transitional cells from NDUFS2 cKO mice are not (Extended Data Fig. 8g,h). Additionally, we evaluated adult transitional cells in adult lung injury models[23,24,28] and compared them with the transitional cells from NDUFS2 cKO mice. All of the examined adult transitional cells displayed an enriched ISR signature compared with other epithelial cells

within their single-cell RNA-seq datasets (Extended Data Fig. 8i,k,m). Notably, the expression of ISR genes in these adult lung injury and repair models was not as high as the expression of ISR genes in the NDUFS2 cKO transitional cells (Extended Data Fig. 8j,l,n).

Remarkably, administration of the small-molecule ISR inhibitor (ISRIB) significantly extended the lifespan of NDUFS2 cKO mice and reversed most of the histologic abnormalities in alveolar structure observed in these mice (Fig. 4f and Extended Data Fig. 9a–e). Similarly, administration of nicotinamide mononucleotide (NMN), an NAD⁺ precursor, partially rescued the lethality of NDUFS2 cKO mice (Extended Data Fig. 9f–h). Both ISRIB and NMN decreased the ISR signatures in lung epithelial cells of NDUFS2 cKO mice (Extended Data Fig. 10a–c), while the increased NADH/NAD⁺ ratio in NDUFS2 cKO mice was only reduced significantly with NMN as expected (Extended Data Fig. 10d).

To determine whether the impaired alveolar epithelial differentiation we observed in NDUFS2 cKO mice were cell autonomous, we performed organoid and two-dimensional cultures with culture media supplemented with aspartate and asparagine. Postnatal transitional cells normally start to appear from P7 (ref. 14). Therefore, to evaluate early postnatal transcriptomic signatures, we performed bulk RNA-seq of lung epithelial cells isolated from NDUFS2 cKO and control mice at P6. The expression of ISR genes was slightly higher in NDUFS2 cKO lungs at P6 compared with NDUFS2 control lungs, but it was not as high as in P35 NDUFS2 cKO lungs (Extended Data Fig. 11a–c). Moreover, transcriptomic signatures of NDUFS2 control and cKO lung epithelial cells at P6 were not clearly separated in principal component analysis (PCA), suggesting that the critical pathways are disrupted after P6 (Extended Data Fig. 11d,e). Next, we isolated lung epithelial cells from P6 wild-type mice, before transitional cells appear, and cultured them on 2D plastic culture plates, a system in which AT2 cells have long been recognized to spontaneously differentiate into cells resembling AT1 cells. Using RNA-seq, we confirmed that the isolated lung epithelial cells (that is, AT2 cells) lose AT2 cell markers and express AT1 cell markers in this system 72 h after isolation (Extended Data Fig. 11f). However, addition of the mitochondrial complex I inhibitor, piericidin A, to the culture media prevented AT2 cells from expressing AT1 cell markers, possibly through high ISR activation (Extended Data Fig. 11f–h). We then compared the development of three-dimensional organoids using AT2 cells from NDUFS2 control and NDUFS2 cKO mice at P6. Compared with AT2 cells from NDUFS2 control mice, AT2 cells from NDUFS2 cKO mice showed impaired differentiation, as measured by organoid size. Administration of ISRIB increased the organoid size for both genotypes without changing proliferation (Extended Data Fig. 12a–f). Previous results indicate that the mitochondrial protease OMA1 cleaves DELE1, which is released into the cytosol in response to ETC inhibition to activate the ISR through the haem-regulated eIF2α kinase (HRI)[36]. Using CRISPR–Cas9, we created an *Oma1* knockout mouse lung epithelial cell line (MLE-12 *Oma1* KO) and showed that piericidin A increased ATF4 protein abundance in an OMA1-dependent manner (Extended Data Fig. 12g–j). This result suggests that mitochondrial complex I inhibition in lung epithelial cells induces activation of the ISR through the OMA1–DELE1–HRI pathway. Collectively, our results indicate that abnormally high ISR activation causes a cell autonomous barrier to cell differentiation, which alters developmental cell fate in the setting of mitochondrial complex I loss.

Our data showing that expression of yeast NDI1 can reverse the pathology of NDUFS2 cKO mice suggests that inhibition of NAD⁺ regeneration is the key driver of an abnormally high ISR activation and subsequent expansion of transitional cells. These findings are consistent with previous in vitro work that suggests that mitochondrial impairment of NAD⁺ regeneration activates the ISR[18]. Although inhibiting the function of mitochondrial ETC complexes I, III, IV or V decreases NAD⁺ regeneration and alters the NADH/NAD⁺ ratio (Fig. 5a), inhibiting mitochondrial complex II function does not[44]. Thus, we ablated the mitochondrial complex II subunit succinate dehydrogenase subunit D gene (*Sdhd*) in the distal lung epithelium during development by

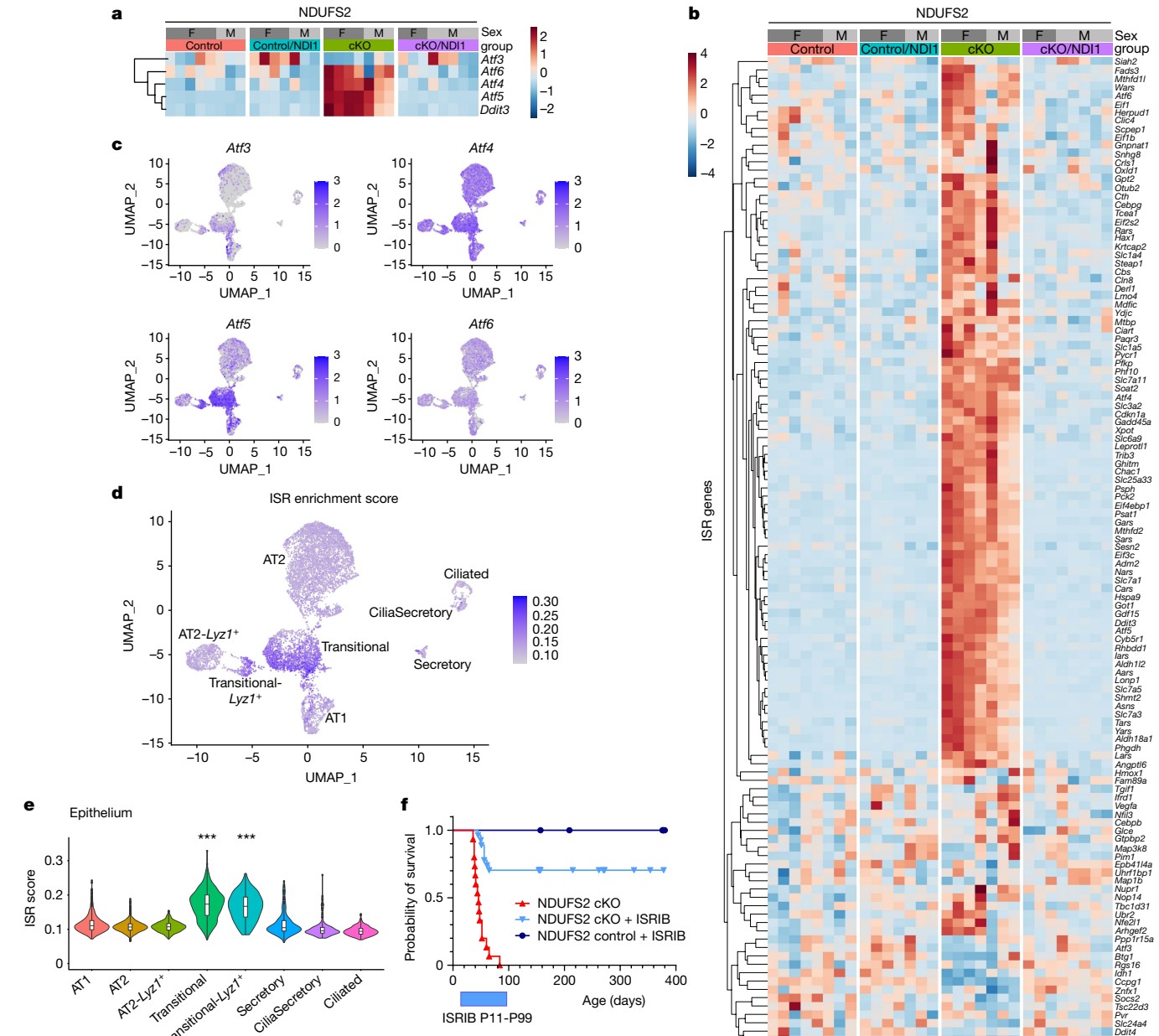

**Fig. 4 | Loss of mitochondrial complex I in lung epithelial cells induces a robust ISR that precludes alveolar development. a,b**, RNA-seq analysis of lung epithelial cells isolated from 35-day-old mice (NDUFS2 control $n = 8$; NDUFS2 control/NDI1 $n = 7$; NDUFS2 cKO $n = 7$; NDUFS2 cKO/NDI1 $n = 8$ mice). **a**, Heat map of ATF transcripts and *Ddit3*. **b**, Heat map of ISR signature gene transcripts. **c**, UMAP plot showing expression of *Atf* genes in single-cell RNA-seq analysis of epithelial cells. **d**, UMAP displaying the level of ISR enrichment score calculated from the overall expression of ISR genes in each epithelial cell. Darker colour indicates a higher ISR enrichment score and thus

a highly enriched ISR gene signature. **e**, Violin plots of ISR enrichment scores in epithelial subclusters. $P < 2.2 \times 10^{-16}$ by Kruskal–Wallis test. Transitional cells and Transitional-*Lyz1*[+] cells have more enriched ISR gene signatures than other epithelial cells (***adjusted $P < 1.0 \times 10^{-29}$ by post hoc pairwise Mann–Whitney test with Holm method; *P* values are in the Source Data). **f**, The ISRIB reduces NDUFS2 cKO lethality. Survival of NDUFS2 cKO mice with or without ISRIB (NDUFS2 cKO $n = 15$; NDUFS2 cKO + ISRIB $n = 27$ mice; $P < 0.0001$ by log-rank test) and NDUFS2 control mice with ISRIB (NDUFS2 control + ISRIB $n = 14$ mice).

crossing *SFTPC-Cre* mice with *Sdhd*[fl/fl] mice[45] (hereafter referred to as SDHD cKO mice, *Sdhd*[fl/fl]*SFTPC-Cre*). SDHD is a nuclear-encoded subunit that is essential for the enzymatic activity of mitochondrial complex II. Lung epithelial cells isolated from SDHD cKO mice displayed a decrease in both basal and coupled OCR (Fig. 5b) and an increase in succinate levels without a change in lactate levels, compared with those from *Sdhd*[fl/fl] mice (hereafter referred to as SDHD control mice) (Fig. 5c,d), which indicates effective deletion of *Sdhd* in the lung epithelial cells of our SDHD cKO mice. Unlike NDUFS2 cKO mice, SDHD cKO mice survived postnatally (Fig. 5e) and their lung compliance

was comparable with that of SDHD control mice at P47–49 (Fig. 5f). Histology of SDHD cKO lungs showed slightly thicker alveolar septa compared with SDHD control lungs (Fig. 5g and Extended Data Fig. 13a). RNA-seq data revealed that lung epithelial cells from SDHD cKO mice displayed an induction of the ISR, but the degree of induction was less than that observed in the NDUFS2 cKO mice (Fig. 5h–j and Extended Data Fig. 13b). It is important to note that succinate levels, increased both in NDUFS2 cKO/NDI1 and SDHD cKO mice, were not elevated by the administration of ISRIB or NMN in NDUFS2 cKO mice (Extended Data Fig. 13c,d). These findings suggest that the ability to regenerate NAD[+]

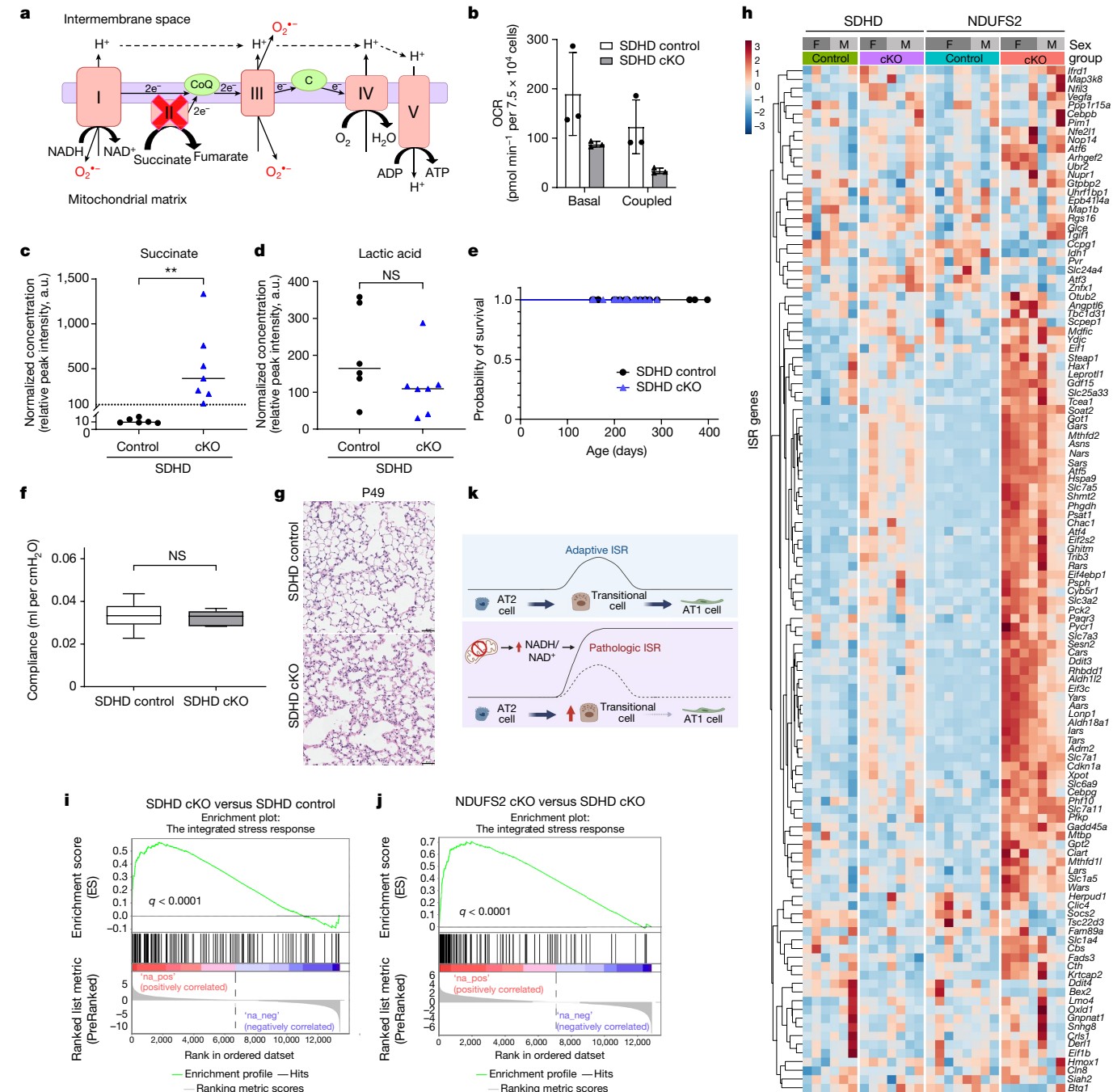

**Fig. 5 | Developmental loss of mitochondrial complex II in lung epithelial cells is not detrimental. a**, Schematic of the mitochondrial electron transport chain in lung epithelial cells of SDHD cKO mice. **b**, Basal and coupled OCR of lung epithelial cells isolated from 4-month-old mice ($n = 3$ mice per genotype with technical replicates). Data represent mean ± s.d. **c,d**, Relative abundance of succinate (**c**) and lactic acid (**d**) in lung epithelial cells isolated from 35-day-old mice (SDHD control $n = 6$; SDHD cKO $n = 7$ mice). Lines represent median. $P = 0.0012$ (**c**) and $P = 0.0734$ (**d**) by Mann–Whitney test. **e**, Survival of SDHD control ($n = 34$) and SDHD cKO ($n = 37$) mice ($P > 0.9999$ by log-rank test). **f**, Box plots of static lung compliance in 47- to 49-day-old mice (SDHD control $n = 15$; SDHD cKO $n = 5$ mice), $P = 0.7354$ by Mann–Whitney test. **g**, Representative images of lung histology on postnatal day 49 (haematoxylin and eosin stain). Scale bar, 50 μm. **h–j**, RNA-seq analysis of lung epithelial cells from 35-day-old mice (SDHD control $n = 6$; SDHD cKO $n = 7$; NDUFS2 control $n = 8$; NDUFS2 cKO $n = 7$ mice). Data in Fig. 4a,b were partly included. **h**, Heat map of ISR signature gene transcripts. **i,j**, Enrichment plot of the ISR signature genes in lung epithelial cells from SDHD cKO versus SDHD control mice (**i**) (normalized enrichment score (NES) 2.15; false discovery rate (FDR) $q < 0.0001$) and those from NDUFS2 cKO versus SDHD cKO mice (**j**) (NES 2.80; FDR $q < 0.0001$). **k**, During the transitional cell state, the adaptive ISR is transiently induced and subsequently subsides as transitional cells differentiate into AT1 cells. Loss of mitochondrial complex I function results in an increase in the NADH/NAD⁺ ratio, leading to persistently high-level activation of the ISR. Chronically high ISR alters cell fate by preventing the successful differentiation of transitional cells into AT1 cells. Fig 5k created with BioRender.com. NS, not significant.

in the lung epithelial cells of SDHD cKO mice is sufficient to prevent abnormally high ISR induction and allows adequate postnatal alveolar development for survival.

In summary, we have demonstrated that mitochondrial complex I-dependent NAD⁺ regeneration controls lung epithelial cell differentiation by preventing pathologic ISR activation. The ISR is transiently

activated during normal lung development and as an adaptive response to mitochondrial ETC inhibition-induced metabolic stress[46]. However, our data indicates that chronic high-level activation of the ISR can be pathologic, and thus alter cell fate during development. Pathologic activation of the ISR leads to the accumulation of transitional cells, characterized by high expression of cytokeratin and cell-cycle regulatory genes. This intermediate transitional cell population has been described in several settings where AT2 cells differentiate into AT1 cells during development and after injury[14,22–29]. Our data suggests that loss of mitochondrial complex I function activates high levels of the ISR, which prevents the successful differentiation of transitional cells into AT1 cells. The resulting failure in postnatal epithelial development causes respiratory failure and death of the animal. The rescue of the NDUFS2 cKO mouse phenotype with expression of NDI1 and the preserved lung development in SDHD cKO mice suggests that the ability of mitochondrial complex I to regenerate $NAD^+$ is required to prevent pathologic ISR activation in lung epithelium during postnatal alveolar development. Thus, this study uncovers an unappreciated role of the ISR and the mitochondria, independent of their role in generating ATP, in dictating lung epithelial cell fate in vivo.

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

## Methods

### Animal models

*Ndufs2*[fl/fl] mice[16], *Sdhd*[fl/fl] mice[45], *SFTPC-Cre* mice[15] and *NDI1*[LSL] mice[7] have previously been described. *ROSA26Sor*[CAG-tdTomato] (Ai14, stock no. 007908) mice were obtained from Jackson Laboratory. All the strains were backcrossed for three generations to C57BL/6J mice in house before breeding and confirmed to be greater than 96% C57BL/6J per the SNP analysis by DartMouse. Animals were housed at Northwestern University animal facility, where the animals were on a 14-h on, 10-h off light cycle, the room temperature range was 21–23 °C and humidity was within a 30–70% range compliant with the guidelines. Our breeding strategies allow only one copy of maternally inherited *Cre* in all experimental mice. ISRIB (AdooQ, A14302) was dissolved in DMSO at 6.25 mg ml⁻¹ and subsequently diluted in sterile saline at 0.25 mg ml⁻¹ and delivered intraperitoneally to mice at a dose of 2.5 mg kg⁻¹ per day every other day in the afternoon from P11 to P99. NMN (Sigma, catalogue no. N3501) was dissolved in sterile saline at 50 mg ml⁻¹ and delivered intraperitoneally to mice at a dose of 500 mg kg⁻¹ per day four times a week between 5 p.m. and 7 p.m. local time, from P11 to P90. Both male and female mice were used in all experiments. All animal procedures were approved by Institutional Animal Care and Use Committee (IACUC) at Northwestern University.

### Lung cell isolation

After mice were euthanized, the pulmonary vasculature was perfused through the right ventricle with Hanks' balanced salt solution (HBSS) until clear. The trachea was cannulated with a 20–24 gauge catheter depending on the mouse's age and/or size, and the lungs were removed en bloc and gently inflated with dispase (Corning, catalogue no. 354235). The trachea and bilateral main bronchi were removed from the inflated lungs before they were incubated in dispase with gentle rocking for 30 min at room temperature. The digested lungs were placed in a petri dish with 25 mM HEPES (Gibco, catalogue no. 15630) buffered DMEM media (Corning, catalogue no. 10-013-CV) with 0.02% DNase I (Sigma, catalogue no. D4513). Any visible proximal airways were removed, and then tissue was torn apart and minced to make a single-cell suspension. The resulting suspension was passed through a 70 µm filter and subsequently a 40 µm filter to remove residual tissue fragments and centrifuged at 250*g* for 5 min at 4 °C. The pelleted cells were resuspended and incubated in BD Pharm Lyse (BD Biosciences, catalogue no. 555899) to remove erythrocytes. The resulting whole lung single-cell suspension was kept in complete DMEM media (DMEM media supplemented with 10% dialyzed fetal bovine serum (FBS) (Peak, catalogue no. PS-FB2), 1x Penicillin-streptomycin (Gibco, catalogue no. 15140), 2 mM L-glutamine (Gibco, catalogue no. 25030), 1x MEM NEAA (Gibco, catalogue no. 11140) and 25 mM HEPES) at 4 °C for further process. For the epithelial cell isolation, the cells were incubated with anti-mouse biotin-conjugated CD45 (BD Biosciences, catalogue no. 553078), CD31 (BD Biosciences, catalogue no. 553371) and CD16/CD32 (BD Biosciences, catalogue no. 553143) antibodies and subsequently with magnetic beads (Promega, catalogue no. Z5482) for negative selection. CD45⁻CD31⁻CD16/CD32⁻ cells were further incubated with EpCAM microbeads (Miltenyi Biotec, catalogue no. 130-105-958) for positive selection. For mice who were 11-days-old or younger, cells were processed for EpCAM positive selection without negative selection. The ages of mice used in cell isolation were selected to evaluate early molecular drivers of the phenotype in the mutant strains and to avoid survivor bias in the assays.

### Lung histology and immunohistochemistry

For embryonic time points, timed mating was performed; the noon on the day of appearance of vaginal plugging in the mother was taken as embryonic day (E) 0.5. Individual embryos were also staged by fetal crown-rump length at the time of euthanasia. The embryos were fixed in 10% neutral-buffered formalin (NBF) for more than 48 h and processed to be embedded in paraffin. For mice who were 11 days or older, after euthanasia and perfusion, the trachea was cannulated and the lungs were inflated with 10% NBF for fixation. Fixed lungs were dehydrated and embedded in paraffin. All paraffin-embedded tissues were prepared for 4 µm thick sections. Immunohistochemistry was performed using primary antibodies against the following epitopes: pro-SftpC (rabbit, Millipore, catalogue no. AB3786; 1:500), podoplanin (Syrian hamster, Abcam, catalogue no. ab11936; 1:2,000), Ki67 (rabbit, Abcam, catalogue no. ab16667; 1:100) and CD45 (rabbit, Abcam, catalogue no. ab10558; 1:1,500). Before primary antibody incubation, sections were incubated with a sodium citrate buffer (pH = 6) at 110 °C for 20 min in a pressure cooker for antigen retrieval. 3,3′-diaminobenzidine was used for chromogenic detection. All staining was completed on an automated platform (IntelliPATH by Biocare Medical). TUNEL assay was performed with terminal transferase (New England BioLabs, catalogue no. M0315L) and Biotin-16-dUTP (Millipore Sigma, catalogue no. 11093070910). Images were acquired using a Nikon microscope and Tissue Gnostics.

### Alveolar thickness quantification

Haematoxylin and eosin-stained images, obtained by TissueGnostics (×20 with a numerical aperture (NA) of 0.50), were processed and quantified using ImageJ/Fiji software (NIH) to measure alveolar septal thickness. Four to six randomly selected fields of view from each mouse lung histology were analysed. The images with proximal airways were excluded. Each colour image was converted to a greyscale image. For segmentation, we performed thresholding with the Huang algorithm. Holes in the segmentation smaller than 1.38 µm (5 pixels) were filled to analyse only the distance to the outside of the vessel (alveolar septal wall). The distance map was then calculated and we counted the number of pixels belonging to the respective alveolar thickness bin and the total pixel count of all alveolar septal walls in each image. Relative frequency was calculated as follows: (number of foreground pixels that belong to respective alveolar thickness bin)/(total number of foreground pixels). To test statistical significance for genotype, we calculated average alveolar thickness in each image (Source Data). Statistical significances were then calculated by *F*-test for the following linear model, where Condition (genotype) denotes whether or not the corresponding mouse was cKO.

$$\text{Thickness} = \beta_0 + \beta_1\text{Condition(genotype)} + \sum_i \beta_i\text{Mouse}_i$$

### RNA in situ hybridization

Multiplex fluorescent in situ hybridization was performed using RNAscope (Advanced Cell Diagnostics (ACD)). As described above, mouse lungs were inflated and fixed with 10% NBF for 24 h at room temperature. Lungs were paraffin embedded and prepared for 4 µm thick sections. Slides were baked for 1 h at 60 °C, deparaffinized in xylene, dehydrated in 100% ethanol and air-dried for 5 min at 60 °C. Sections were treated with hydrogen peroxide (ACD, catalogue no. 322330) for 10 min at room temperature and then heated to mild boil (98–102 °C) in 1x Target Retrieval Reagent (ACD, catalogue no. 322001) for 15 min. Protease plus (ACD, catalogue no. 322330) was applied to sections for 30 min at 40 °C in a HybEZ Oven (ACD, catalogue no. 241000). Hybridization with target probes, preamplifier, amplifier, fluorescent labels and wash buffer (ACD, catalogue no. 320058) were carried out according to ACD instructions for Multiplex Fluorescent Reagent Kit v2 (ACD, catalogue no. 323100). Parallel mouse tissue sections were incubated with positive (ACD, catalogue no. 321811) and negative (ACD, catalogue no. 321831) control probes. Sections were mounted under a no. 1.5 coverslip with ProLong Gold Antifade (Thermo, catalogue no. P36930). Probes used were mouse *Sftpc* (ACD, catalogue no. 314101-C3, NM_011359.2), *Pdgfra* (ACD, catalogue no. 480661-C2, NM_011058.2),

*Car4* (ACD, catalogue no. 468421, NM_007607.2) and *Sfrp1* (ACD, catalogue no. 404981, NM_013834.3). Opal fluorophores (Opal 520 (catalogue no. FP1487001KT), Opal 620 (catalogue no. FP1495001KT) and Opal 690 (catalogue no. FP1497001KT) (Perkin Elmer) were used at 1:1,500 (for 620 and 690) and 1:9,000 (for 520) dilution in Multiplex TSA buffer (ACD, catalogue no. 322809). Images were captured on a Nikon A1C confocal microscope with a ×40 objective and NA of 1.30 (NU-Nikon Cell Imaging Facility). Wavelengths used for excitation included 405 nm, 488 nm, 561 nm and 640 nm.

## Mouse AT2 cell culture

Mouse lung AT2 cells were isolated from 6-day-old mice with EpCAM positive selection as described above. For the classic two-dimensional culture, isolated cells were plated in 48-well cell culture plate (Corning, catalogue no. 353230) at $1.25 \times 10^5$ cells per well and cultured in complete DMEM media (DMEM media supplemented with 10% dialyzed FBS, 1x penicillin-streptomycin, 2 mM L-glutamine, 1x MEM NEAA and 25 mM HEPES). The remaining cells were processed for RNA-seq (culture 0 h). After 56 h of culture, new culture media with or without piericidin A (Cayman, catalogue no. 15379) was added to the cultures to achieve a final concentration of 0.5 μM piericidin A. After 16 h (a total of 72 h of culture), cells in each well were processed for RNA-seq (culture 72 h).

The three-dimensional alveolar organoid cultures were performed as previously described[2,42,47] with modifications. In brief, lung fibroblasts were isolated from 7-week-old wild-type mice with CD45 depletion and cultured for 4–5 passages to expand in DMEM media with 4.5 g l⁻¹ D-glucose, 2 mM L-glutamine, 10% FBS and 1% penicillin-streptomycin as previously described[5]. Immediately before use in organoid culture, fibroblasts were treated with mitomycin-C (Millipore Sigma, catalogue no. M4287) for 2 h. AT2 cells were isolated from 6-day-old mice as described above. AT2 cells and lung fibroblasts (1:10) were suspended in 50% Matrigel (Corning, catalogue no. 356231) and 50% organoid growth media (alpha-MEM media (Thermo Fisher, catalogue no. 41061029) supplemented with 2 mM L-glutamine, 10% FBS, 1% penicillin-streptomycin, 1% Insulin-Transferrin-Selenium (Thermo Fisher, catalogue no. 41400045), 0.002% Heparin, 0.25 ug ml⁻¹ Amphotericin B (Millipore Sigma, catalogue no. A2942) and 2.5 μg ml⁻¹ ROCK inhibitor Y24632 (Selleckchem, catalogue no. S1049)). Then 100 μl of the cell-media-matrigel mixture ($5 \times 10^3$ tdTomato⁺ AT2 cells and $5 \times 10^4$ lung fibroblasts per insert) was plated in a 24-well 0.4 μm Transwell insert (Corning, catalogue no. 3470) and solidified at 37 °C for 5 min before 500 μl of organoid growth media was added under the insert. The next day, organoid growth media was switched to fresh media containing either 1 μM ISRIB or DMSO and changed every other day. After 10 days of culture, organoids were imaged on a Nikon Ti² Widefield in brightfield and red fluorescent protein (RFP) channels with the objective of ×20 and NA of 0.45. Alveolar organoids were defined as a clonal colony with a minimum diameter of 50 microns. Images were processed in Nikon Elements (v.5.11.00) and quantified using ImageJ/Fiji software to evaluate organoid diameters and colony counts. All culture media contained aspartate and asparagine.

## Mitochondrial OCR

The OCR of lung epithelial cells was measured in a Seahorse XF96 extracellular flux analyser (Agilent Bioscience) with Wave v.2.6.3.5 software. Isolated lung epithelial cells as described above were immediately seeded at $7.5 \times 10^4$ cells per well using cell adhesive, Cell-Tak (Corning, catalogue no. 354240) according to the manufacturer's instructions. Basal mitochondrial respiration was assessed by subtracting the non-mitochondrial OCR, measured with 1 μM antimycin A (Sigma, catalogue no. A8674) and 1 μM piericidin A (Cayman, catalogue no. 15379), from baseline OCR. Coupled respiration was determined by subtracting the OCR in the presence of 2 μM oligomycin (Sigma, catalogue no. 75351) from the basal mitochondrial respiration.

## Cell line culture

A mouse lung epithelial cell line (MLE-12; ATCC, CRL-211) was cultured in HITES media (DMEM/F12 (1:1) (Gibco, catalogue no. 11320033), 1x Insulin-Transferrin-Selenium (Gibco, catalogue no. 41400045), 10 nM Hydrocortisone (Sigma, catalogue no. H4001), 10 nM β-oestradiol (Sigma, catalogue no. E2758), 10 mM HEPES (Corning, catalogue no. 25-060-CI), 1x GlutaMAX (Gibco, catalogue no. 35050061)) supplemented with 4% FBS (Atlas Biologicals, catalogue no. F0500A), 1 mM methyl-pyruvate (Sigma-Aldrich, catalogue no. 371173), 400 μM uridine (Sigma-Aldrich, catalogue no. U3003), 50 μM L-Asparagine (Sigma-Aldrich, catalogue no. A4284; in addition to 50 μM L-asparagine in the basal medium), 1x antibiotic-antimycotic solution (Gibco, catalogue no. 15240062) and 2.5 μg ml⁻¹ Plasmocin Prophylactic (Invivogen, ant-mpp)). Cells were incubated at 37 °C, 5% $CO_2$ and 95% humidity.

## Generation of cell lines with knockouts and ectopic expression

A single-guide RNA (sgRNA) oligonucleotide targeting *Oma1* or a non-targeting control sgRNA was cloned into the pSpCas9(BB)-2A-GFP (PX458) plasmid (Addgene, 48138; a gift from F. Zhang at the Massachusetts Institute of Technology), according to the provider's instructions. Oligonucleotide sequences were as follows: sg*Oma1*: 5′-CGTGTGCGATCTCATGGCCC-3′ (targeting the '+' strand in exon 5); non-targeting sgRNA: 5′-GCGAGGTATTCGGCTCCGCG-3′. Both sgRNA-Cas9-2A-GFP vectors were then transfected into MLE-12 cells using jetOPTIMUS transfection reagent (Polyplus). Forty-eight hours after transfection, the GFP⁺ cells were single-cell sorted into 96-well plates using a BD FACSAria cell sorter. The sorted cells were grown in culture for 2–3 weeks and the resultant clonal cell lines were expanded. Knockout of *Oma1* was confirmed by immunoblotting.

*Oma1* coding sequence (NM_025909) was cloned into the pLV-EF1-RFP vector (VectorBuilder) using GenScript service. The pLV-Oma1-EF1-RFP vector or empty vector control, along with pMD2.G and psPAX2 lentiviral packaging vectors, were then transfected into 293T cells (ATCC, CRL-3216, using jetOPTIMUS (Polyplus)) to generate Oma1-RFP or empty vector control-RFP lentivirus, respectively. *Oma1* KO MLE-12 cells were transduced with empty vector control-RFP or Oma1-RFP lentivirus and then RFP⁺ cells were sorted using a BD FACSAria cell sorter. The cells were periodically sorted to maintain high RFP expressions. *Oma1* overexpression was confirmed by immunoblotting. Cells were incubated with 500 nM piericidin A (Cayman, catalogue no. 15379) or 100 nM oligomycin (Sigma, catalogue no. 75351) for 16 h, respectively, before collection for immunoblotting analysis.

## Immunoblot blot analysis

Lung epithelial cells were isolated from 11-day-old mice as described above, washed with ice-cold phosphate buffered saline and stored at −80 °C until processed. Cells were lysed in NP40 cell lysis buffer (ThermoFisher, catalogue no. FNN0021) supplemented with Halt protease inhibitor cocktail (ThermoFisher, catalogue no. 78430). Protein concentrations were measured using the Pierce BCA Protein Assay Kit (Thermo Fisher Scientific, catalogue no. 23225). Immunoblots were performed using the Protein Simple WES/Sally Sue platform (Bio-Techne), a capillary electrophoresis immunoassay, according to the manufacturer's instructions. Protein abundance was quantified using Compass software. Primary antibodies used were anti-NDUFS2 (Abcam, ab192022, 1:200 dilution), anti-Vinculin (Abcam, ab129002, 1:500 dilution; as an NDUFS2 protein loading control), anti-Oma1 (SCBT, sc-515788, 1:50 dilution), anti-ATF4 (CST, 11815S,1:50 dilution) and anti-cofilin (CST, 5175T, 1:30,000, as an OMA1 loading control and 1:10,000 as an ATF4 loading control). Relative abundances of NDUFS2, OMA1 and ATF4 protein were quantified as the peak area of NDUFS2, OMA1 and ATF4 over the peak area of VINCULIN (for NDUFS2) and COFILIN (for OMA1 or ATF4) in each capillary lane, respectively.

## Static lung compliance analysis

Mice were anesthetized and tracheotomized. Respiratory mechanics were assessed using the flexiVent FX equipped with a module 1 (flexiVent FX, SCIREQ Scientific Respiratory Equipment).

## Metabolite measurements

Metabolomics were carried out as previously described[7,48,49] with modifications. Briefly, 35-day-old mice were euthanized and lung epithelial cells were isolated as described above. Cells were washed once with ice-cold HBSS and divided into two dry cell pellets, one of which was frozen and stored at −80 °C for metabolites extraction until all samples were collected. The remaining cell pellet, if any, was processed for RNA-seq (RNA sequencing). To extract metabolites, samples were suspended in 225 µl ultra-cold HPLC-grade methanol/water (80/20, v/v) per one million cells (333 µl ultra-cold HPLC-grade acetonitrile/water (80/20, v/v) per one million cells for SDHD control and SDHD cKO mice) and went through three complete freeze-thaw cycles in −80 °C and 4 °C before high-speed centrifugation at 4 °C. The supernatants, which contained metabolites, were collected and dried in a SpeedVac concentrator (Thermo Savant). The dried metabolites were reconstituted in 50% acetonitrile in analytical-grade water (50/50, v/v) and centrifuged to remove debris. Samples were analysed by ultra-high-performance liquid chromatography and high-resolution mass spectrometry and tandem mass spectrometry (UHPLC-MS/MS). The metabolites extracted with 80% acetonitrile were directly injected into the mass spectrometry without drying and reconstitution. Data were acquired with Xcalibur software (v.4.1; ThermoFisher Scientific). The resulting data were analysed using the MetaboAnalyst software v.5.0 (refs. 50,51) and the MetaboAnalystR package v.4.1.2 (ref. 52). Metabolites were normalized by total ion count for each sample. Significantly different metabolites among groups were identified by one-way analysis of variance followed by Fisher's least significant difference post hoc analysis with FDR < 0.05 and then plotted as a heat map. NADH/NAD$^+$ ratios were calculated from the peak area values of NADH and NAD$^+$ within the same individual sample and compared between groups. Our extraction method may allow interconversion between the reduced and oxidized forms during extraction[53]. Some metabolites were reported as zero because the metabolite levels were low and below the detection limit. Normalized peak areas of individual metabolites (lactate, aspartate, asparagine and succinate) were graphed as arbitrary units (a.u).

## RNA sequencing

Mouse lung epithelial cells were isolated as described above and washed with ice-cold HBSS. The cell pellet was lysed with RLT Plus buffer (Qiagen, catalogue no. 74134) with 1% β-mercaptoethanol and stored at −80 °C until all samples were collected for RNA extraction. RNA was extracted using the RNeasy Plus Mini Kit (Qiagen, catalogue no. 74134), according to the manufacturer's protocol. The quantity and quality of the extracted RNA were assessed using TapeStation 4200 (Agilent). mRNA libraries were prepared using NEBNext Ultra Kit with polyA selection (New England BioLabs, catalogue nos. E7530 and E7490) and sequenced on NextSeq 500 High output for 75 cycles (Illumina) or NextSeq 2000 P2 or P3 100 cycles (Illumina).

## RNA sequencing data analysis

The sequencing data was demultiplexed using bcl2fastq v.2.20.0 provided by Illumina and trimmed using Trimmomatic v.0.39 (ref. 54). Reads were then aligned to the GRCm39 reference genome using the STAR aligner v.2.7.7 (ref. 55) and counts were calculated using HTseq v.0.11.0 (ref. 56). The ComBat-seq[57] package was used to adjust for batch effect on RNA-seq count data related to the multiple library preparations and sequencing from different days. The DESeq2 (ref. 58) package was used to generate a PCA plot to visualize the clustering patterns of the samples based on their gene expression profiles after data

transformation. The edgeR[59] package was used for identifying differentially expressed genes. Using the filterByExpr function in edgeR, lowly expressed counts were filtered out before library normalization. An additive model was created to adjust for sex differences in the samples and the counts were fit to a negative binomial generalized linear model for comparison. The CPM (counts per million reads mapped) matrix was generated using the cpm function in edgeR. Heat maps visualizing expression levels of ATF genes, ISR signature genes and cell marker genes in each sample by genotypes or conditions were generated by pheatmap package (https://github.com/raivokolde/pheatmap/). Gene set enrichment analysis was performed using the gene set enrichment analysis software v.4.2.1 (ref. 60) with hallmark gene sets[61] or a curated list of ISR genes[62] (Supplementary Table 1; a gift from C. Sidrauski at the Calico Life Sciences).

## Single-cell RNA sequencing

Whole lung single-cell suspensions from 21-day-old mice were prepared as described above. Cell concentrations were counted using Cellometer K2 (Nexcelom) with AOPI staining solution (Nexelom, CS2-0106-5mL). Single-cell RNA-seq libraries were prepared using Chromium Next GEM Single Cell 3' Reagent Kits v.3.1 (10x Genomics) aiming to capture around 6,000–10,000 cells per library. After quality checks, single-cell RNA-seq libraries were pooled at an equimolar ratio and sequenced shallowly on MiniSeq High Output 150 cycles (Illumina) to rebalance the pool to adjust for different numbers of cells per library and to achieve even sequencing depth coverage (reads per cell) across libraries on deep sequencing. Deep sequencing was performed on the HiSeq 4000 instrument (Illumina).

## Analysis of single-cell RNA sequencing data

Raw sequencing reads were processed using CellRanger v.6.0.1. Reads were aligned onto GRCm39 reference genome with *tdTomato* gene inserted. Doublets were removed using Scrublet v.0.2.1 (ref. 63) from each library. All downstream analysis of single-cell RNA-seq data was performed using Seurat v.4.0.6 (ref. 64) (in R v.4.1.2), except for the part of the analysis of integrated data with other single-cell datasets[14,43] (see below). Quality control was performed by removing cells with more than 25% of reads from mitochondrial genes and cells with less than 500 detected genes. SCTransform[65] was used to normalize and stabilize the variance of molecular count data before performing PCA on the top 3,000 most variable genes. Cells were then clustered with the FindClusters function based on the Louvain algorithm[66] and UMAP embedding was generated with the RunUMAP function. Cell types of the clusters were manually annotated with known cell-type marker genes based on differentially expressed genes in each cluster detected by the FindAllMarkers function. To further classify major cell-type subsets at high resolution, specifically epithelial and mesenchymal cells, we assessed the expression of each canonical cell marker *Epcam* (epithelial cells), *Pecam1* (endothelial cells), *Col1a1* (mesenchymal cells) and *Ptprc* (immune cells) and identified each major cell subset accordingly. Cells co-expressing markers of different cell types were removed as they were likely to be rare doublets that were not removed during the initial data processing. Each subset was then re-processed with the same normalization and dimensionality reduction approach as described above. For the epithelial (*Epcam*$^+$ clusters), AT1 cells (annotated from epithelial subset), or mesenchymal subset (*Col1a1*$^+$ clusters except mesothelium), the subset cells were re-clustered. Identified epithelial or mesenchymal sub-cell types were annotated with known sub-cell-type markers, respectively, based on gene expression markers in each subcluster generated by the FindAllMarkers function. To evaluate differentially expressed genes by mouse genotype within the AT1 cell type, pseudobulk differential expression analysis was performed using the AggregateExpression function in Seurat[64] and the edgeR[59] package. To evaluate the ISR gene signature, we calculated ISR gene signature scores with the UCell algorithm[67] which calculates gene

enrichment scores for single-cell RNA-seq data based on the Mann–Whitney U statistic without being affected by dataset composition. The ISR gene signature is defined by the same curated list of ISR genes[62] (Supplementary Table 1) as in the above RNA-seq data analysis. The enrichment scores for glycolysis and oxidative phosphorylation gene signatures, retrieved from hallmark gene sets[61], were also calculated with the UCell algorithm. The cell cycle stage for each cell was identified by calculating cell cycle phase scores using the CellCycleScoring function.

### Integration with other single-cell datasets and RNA velocity analysis

To compare the ISR gene enrichment of transitional cells from our dataset with those identified by other investigators, we integrated our count matrices with those from Strunz et al. (high-resolution datasets in GSE141259)[23], Choi et al. (Bleomycin-treated *SPC-Cre*[ERT2]; *R26R*[tdTomato] mice cells in GSE145031)[24] and Kobayashi et al. (GSE141634)[28]. We used the SCTransform integration[68] method to perform data integration between the epithelial cells.

Raw sequencing reads in Negretti et al. (PRJNA674755 and PRJNA693167, except P64)[14] and Hurskainen et al. (PRJNA637911)[43] were processed using CellRanger v.6.0.1 and aligned onto a GRCm39 reference genome, respectively, with the same parameters as described above. Only postnatal epithelial cells were included for data integration. Each processed epithelial dataset was combined with the epithelial dataset in our current study using the SCTransform integration[68]. Then UMAP embedding was conducted with Scanpy v.1.8.1 (ref. 69) and batch balancing was conducted by BBKNN[70]. For the analysis of RNA velocity, spliced and unspliced mRNA count matrices were constructed by using velocyto v.0.17 (ref. 71) and RNA velocity was predicted with scVelo v.0.2.4 (ref. 72) in Python v.3.8.3. All charts and visualization plots were generated with ggplot2 and dittoSeq[73].

### Statistics and reproducibility

All data analysis and statistical tests, other than those specified above, were performed using GraphPad Prism software (v.9.5.0). All statistical tests were performed as two-sided. Descriptive data is presented as mean ± s.d. unless stated otherwise. All box plots are displayed as follows: minimum and maximum are the smallest and largest values, respectively, excluding outliers and the box is drawn from the 25th to 75th percentile with the median in the centre. Numbers of biological replicates are indicated in the figure legends. The investigators were not blinded during experiments and outcome assessments. No statistical method was used to predetermine sample size and experiments were not randomized. $P$ values less than 0.05 were considered as significant unless stated otherwise and depicted as following: *$P < 0.05$; **$P < 0.01$; ***$P < 0.001$. Representative images of lung histology are shown from at least $n = 3$ mice.

### Reporting summary

Further information on research design is available in the Nature Portfolio Reporting Summary linked to this article.

### Data availability

All raw sequencing data (.fastq) generated in this study are available at the NCBI BioProject with the following Accession IDs: PRJNA865889, PRJNA940730, PRJNA940746, PRJNA940973, PRJNA940986 and PRJNA940992. Data from Strunz et al. (GSE141259)[23], Choi et al. (GSE145031)[24], Kobayashi et al. (GSE141634)[28], Negretti et al. (PRJNA674755 and PRJNA693167)[14] and Hurskainen et al. (PRJNA637911)[43] were re-analysed. Molecular Signatures Database (MSigDB)[61] and GRCm39 reference genome were used for analysis. Source data are provided with this paper.

### Code availability

All codes used for analysis are available at https://github.com/MinhoLee-DGU/2023.Han.et.al.Nature.

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

**Acknowledgements** We thank the following core facilities at Northwestern University: Pulmonary NextGen Sequencing Core, Transgenic and Targeted Mutagenesis Laboratory, the Robert H. Lurie Cancer Center (RHLCCC) Metabolomics Core, Mouse Histology and Phenotyping Laboratory, and Northwestern University Center for Advanced Microscopy, supported by NCI CCSG P30-CA060553 awarded to the RHLCC. We thank P. Gao for performing our metabolomics runs and H. Abdala-Valencia for technical assistance with RNA-seq. We thank the following members of the Northwestern University, Department of Medicine, Division of Pulmonary and Critical Care: S. Weinberg, R. Grant, N. Markov and A. Misharin for their helpful comments and A. Chaker for assistance with mouse colony management. Some elements in Fig. 5 were generated with BioRender.com. This work was supported by the National Institutes for Health (NIH): R35CA197532 to N.S.C.; P01HL071643 and P01AG049665 to N.S.C. and G.R.S.B.; K08HL143138 to S.H.; R01HL134800 to C.J.G.; T32

HL076139 to C.R.R.; and K08HL146943 to P.A.R. S.H. was supported by the American Heart Association Career Development Award (19CDA34630070), the Parker B. Francis Fellowship and the Doris Duke Charitable Foundation/Walder Foundation and Feinberg School of Medicine COVID-19 Fund to Retain Clinician Scientists. G.R.S.B. was supported by the Veterans Administration grant CX001777. M.L. and Y.S. were supported by the National Research Foundation of Korea (2022R1A2C2093050), KREONET (Korea Research Environment Open NETwork), which is managed and operated by KISTI (Korea Institute of Science and Technology Information) and the Dongguk University Research Fund. R.P.C. and M.M.H. were supported by the Northwestern University Pulmonary and Critical Care Cugell fellowship.

**Author contributions** S.H. and N.S.C. conceptualized the study, interpreted the data and wrote the manuscript with the input of co-authors. G.R.S.B. and C.J.G. conceptualized the study, interpreted the data and revised the manuscript. M.L., Y.S., P.A.R. and B.K. analysed bulk RNA-seq data. M.L. and Y.S. analysed single-cell RNA-seq data. S.H., R.G., R.P.C., M.M.H., L.A.D. and A.S.F. carried out experiments. L.G. and J.L.B. provided *Ndufs2*$^{fl/fl}$ mice and *Sdhd*$^{fl/fl}$ mice and provided input in the editing of the paper. C.R.R. generated the *NDI1*$^{LSL}$ mice and edited the manuscript.

**Competing interests** The authors declare no competing interests.

## Additional information

**Correspondence and requests for materials** should be addressed to SeungHye Han or Navdeep S. Chandel.

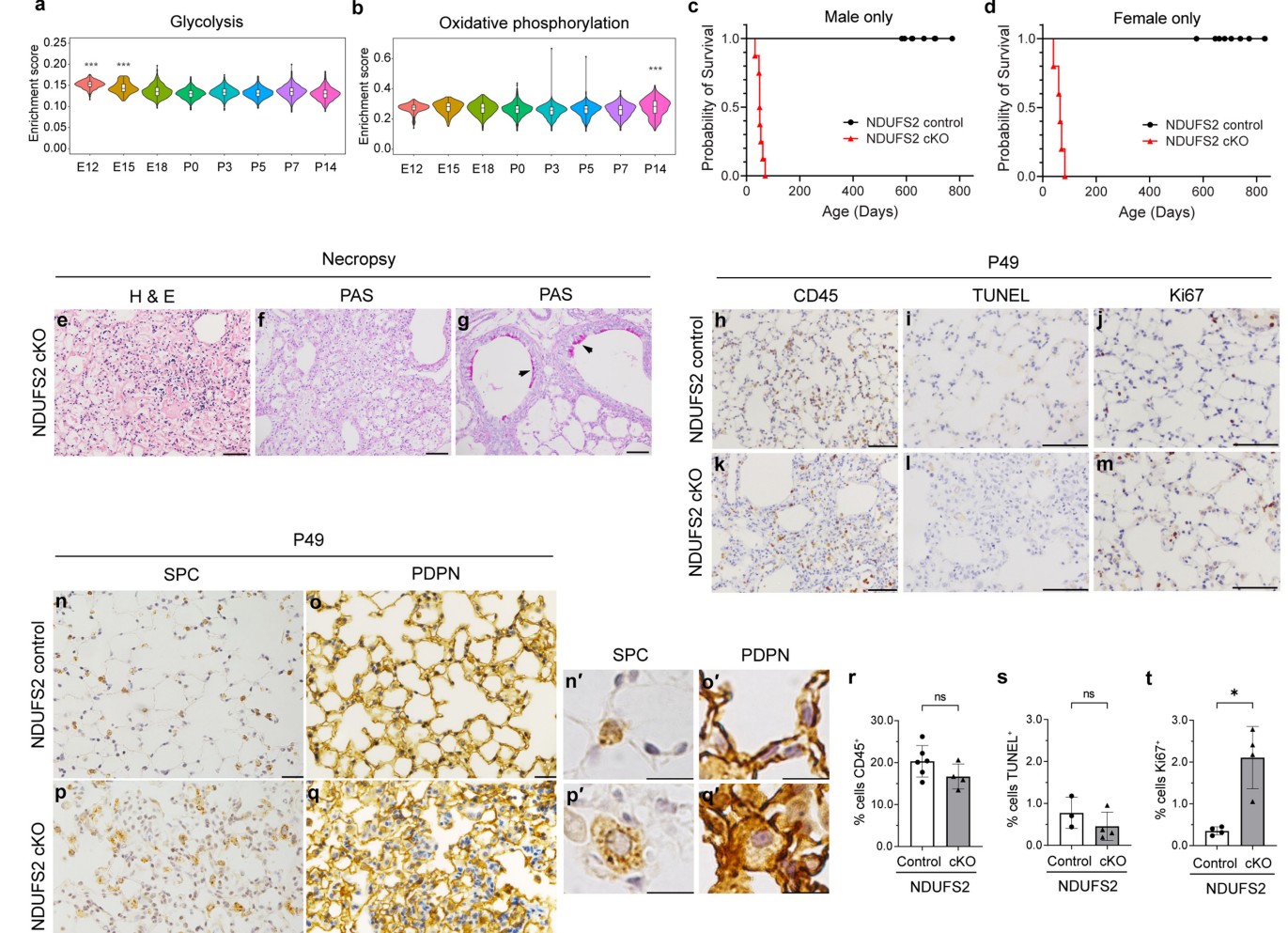

**Extended Data Fig. 1 | Loss of mitochondrial complex I in lung epithelial cells during development induces aberrant hypercellular lung structure with thickened alveolar walls resulting in postnatal death. a–b**, Changes in metabolism-related gene signatures during murine lung epithelial development. Re-analysis of epithelial cells from single-cell RNA-seq data from a mouse lung development atlas[14] (n = 13,445 cells). Violin plots showing enrichment scores of glycolysis (**a**) and oxidative phosphorylation gene signatures (**b**) measured by UCell algorithm[67]. A higher score indicates higher enrichment of the gene signature. The gene lists are retrieved from hallmark gene sets in the Molecular Signatures Database (MSigDB)[61]. Glycolysis gene signatures at E12 and E15 are more enriched than those at P0 – P14, while oxidative phosphorylation gene signature at P14 is more enriched than those at P0 – P7 (*** adjusted $p < 1.0 \times 10^{-5}$ by post-hoc pairwise Mann-Whitney test with Holm method, $p$ values are in the Source Data). $p < 2.2 \times 10^{-16}$ by Kruskal-Wallis test for both glycolysis and oxidative phosphorylation gene enrichment scores. **c**, Survival of male NDUFS2 control (n = 9) and male NDUFS2 cKO (n = 8) mice ($p < 0.0001$ by log-rank test). **d**, Survival of female NDUFS2 control (n = 12)

and female NDUFS2 cKO (n = 5) mice ($p < 0.0001$ by log-rank test). **e–g**, Representative images of lung necropsy from NDUFS2 cKO (n = 4 mice). Alveolar airspaces are filled with pink, homogenous material (hyaline membranes), suggesting the mice died from respiratory failure (**e**). The pink homogenous materials are negative for Periodic acid–Schiff (PAS) stain (**f**). Positive PAS stain (arrowhead) in a different region from the same lung histology section staining (**g**). Scale bar, 100 µm. **h–q**, Lung histology of 49-day-old mice stained for TUNEL assay (apoptosis), CD45 (leukocyte marker), Ki67 (proliferation), surfactant protein C (SPC, AT2 marker), and podoplanin (PDPN, AT1 marker). **n′–q′**, high-magnification images. Scale bars, 100 µm (**h–m**), 20 µm (**n–q**), and 10 µm (**n′–q′**). **r–t**, Quantification of percent CD45[+], TUNEL[+], and Ki67[+] cells (mean ± SD) in 35-day-old mouse lungs. $p = 0.2286$ (TUNEL[+]), $p = 0.2571$ (CD45[+]) and $p = 0.0286$ (Ki67[+]) by Mann-Whitney test. (n = 2540–4930 cells evaluated from at least three randomly selected fields of view in each mouse; NDUFS2 control n = 6 (CD45[+]), n = 3 (TUNEL[+]), n = 4 (Ki67[+]); NDUFS2 cKO n = 4 (CD45[+]), n = 4 (TUNEL[+]), n = 4 (Ki67[+]) mice, both male and female).

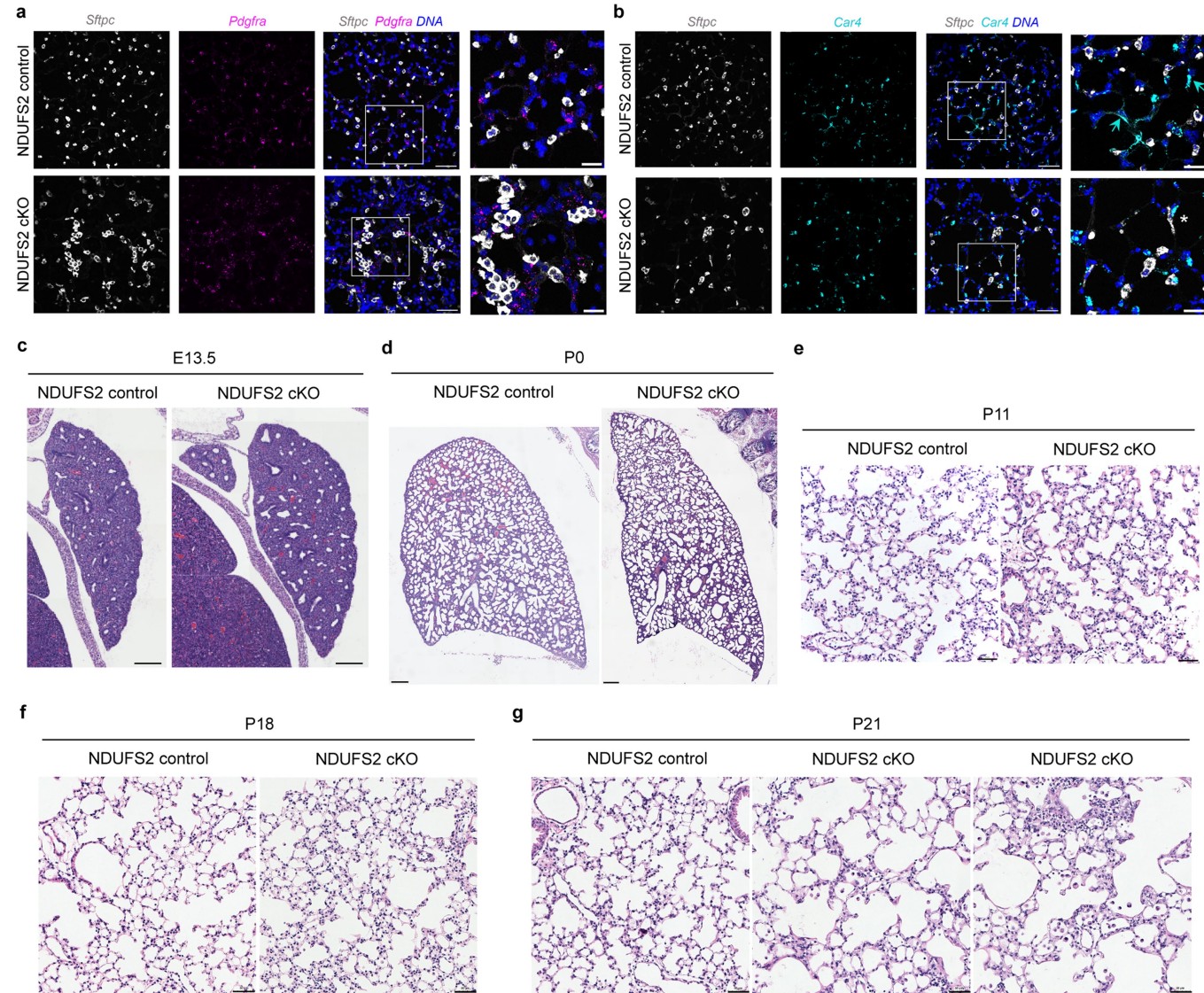

**Extended Data Fig. 2 | *In situ* RNA hybridization with amplification confirms the postnatal disruption of spatial organization between different cell types in NDUFS2 cKO lungs. a–b**, Lung sections from 21-day-old NDUFS2 control and NDUFS2 cKO mice (n = 3 each genotype, both male and female) were hybridized with the indicated target probes (RNAscope®). Magnified images (boxed region) are shown on the right. Representative lung sections (**a**) showing alveolar type 2 (AT2) cells by *Sftpc* (gray) have 1:1 direct contact with *Pdgfra*⁺ fibroblasts (magenta) in NDUFS2 control lungs, whereas 1:1 relationship between *Sftpc*⁺ cells and *Pdgfra*⁺ fibroblasts are lost in NDUFS2 cKO lungs. Hypertrophic *Sftpc*⁺ cells in NDUFS2 cKO lungs cluster next to each other along the alveolar walls while *Sftpc*⁺ AT2 cells in NDUFS2 control lungs locate individually at the corner of alveolar sacs. Representative lung sections (**b**) showing *Car4*⁺ endothelial cells (cyan, arrows) locate next to linear thin *Sftpc*- AT1 cells in NDUFS2 control mice. However, in NDUFS2 cKO lungs, *Car4*⁺ endothelial cells (cyan) are located next to linear thin *Sftpc*⁺ cells (gray, asterisk). Please note that *Sftpc* is an AT2 marker (cuboidal) that does not normally express in linear thin AT1 cells. Scale bars, 50 μm and 20 μm (magnified inset). **c–g**, Mitochondrial complex I in lung epithelial cells is dispensable for antenatal lung development. Representative images of littermates' lung histology (hematoxylin-eosin stain) at different time points (E, embryonic day; P, postnatal day). Branching morphogenesis during antenatal development is not grossly disrupted in NDUFS2 cKO mice compared to NDUFS2 control mice (**c,d**). The subtle differences in alveolar structure between NDUFS2 cKO and NDUFS2 control mice at P11 become apparent by P21 (**e–g**). Scale bars, 200 μm (**c,d**), 50 μm (**e–g**).

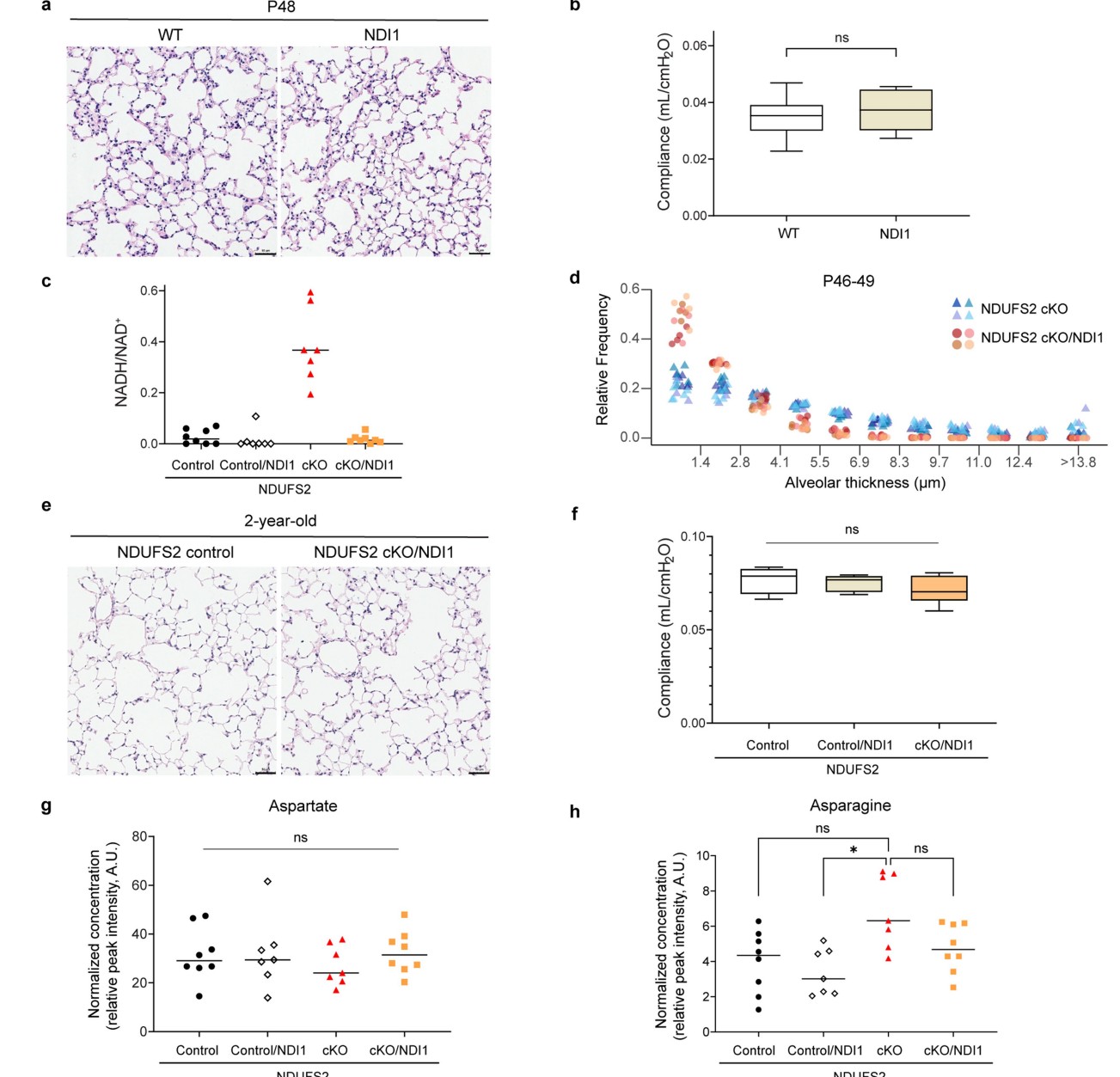

**Extended Data Fig. 3 | Expression of the yeast NDI1 protein in lung epithelial cells does not disrupt lung development or physiology, and restores abnormal alveolar structures in NDUFS2 cKO mice. a**, Representative images of lung histology (hematoxylin-eosin stain) from 48-day-old mice. *NDI1^LSL* mice and *SFTPC-Cre;NDI1^LSL* mice are referred to as WT and NDI1, respectively (scale bar, 50 μm). **b**, Box plots of static lung compliance in 48–49-day-old mice (WT n = 12; NDI1 n = 10 mice with technical replicates), *p* = 0.6744 by Mann-Whitney test. **c**, Intracellular NADH/NAD⁺ ratios from metabolomics analysis of lung epithelial cells isolated from 35-day-old mice (NDUFS2 control n = 8; NDUFS2 control/NDI1 n = 7; NDUFS2 cKO n = 7; NDUFS2 cKO/NDI1 n = 8 mice). *p* = 0.0006 by Kruskal-Wallis test. **d**, The frequency distribution of alveolar thickness measured in hematoxylin-eosin stained lung histology of 46–49-day-old mice (n = 4 mice, two males and two females per genotype). 4–6 randomly selected fields of view from each mouse were evaluated. The *x* axis shows alveolar

thickness bins, and the *y* axis shows the number of alveolar pixels that belong to the respective alveolar thickness bin normalized to the total alveolar pixel count in the image. Each animal is represented by its own color. Statistical significance for genotype was calculated based on F-test for a linear model (*p* = 4.66 x 10⁻⁵). **e**, Representative images of lung histology (hematoxylin-eosin stain) from 2-year-old mice (scale bar, 50 μm), **f**, Box plots of lung compliance in 18–25-month-old mice (NDUFS2 control n = 4; NDUFS2 control/NDI1 n = 5; NDUFS2 cKO/NDI1 n = 15 mice with technical replicates), *p* = 0.3847 by Kruskal-Wallis test. **g–h**, Metabolomics analysis of lung epithelial cells isolated from 35-day-old mice (NDUFS2 control n = 8; NDUFS2 control/NDI1 n = 7; NDUFS2 cKO n = 7; NDUFS2 cKO/NDI1 n = 8 mice). Lines represent median. Relative abundance of aspartate. *p* = 0.7515 by Kruskal-Wallis test (**g**). Relative abundance of asparagine. *p* = 0.0253 by Kruskal-Wallis test. **p* = 0.0259 by Dunn's multiple comparisons test (**h**).

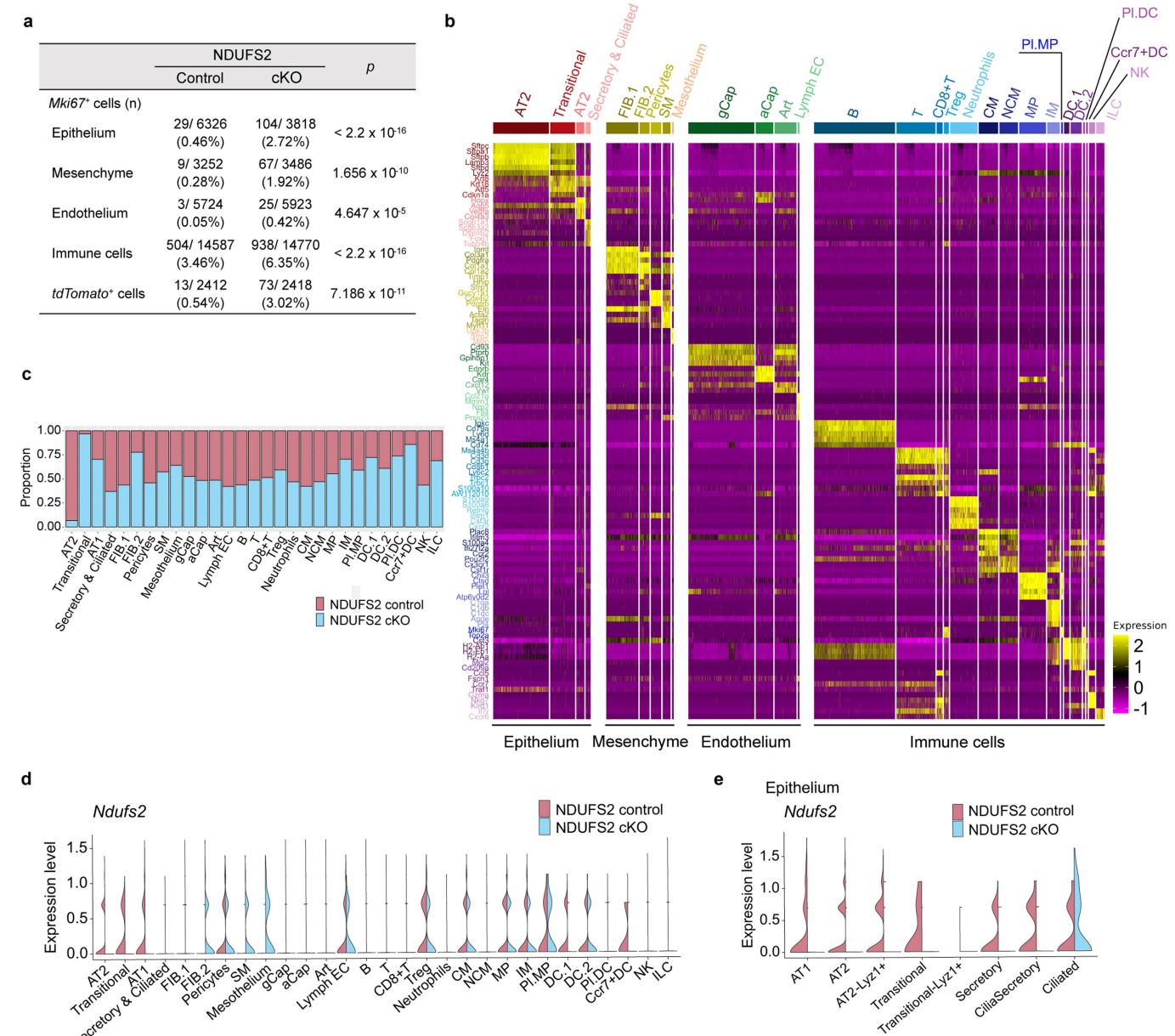

**a**

| | NDUFS2 | | |
| | Control | cKO | p |
| --- | --- | --- | --- |
| *Mki67*+ cells (n) | | | |
| Epithelium | 29/ 6326 (0.46%) | 104/ 3818 (2.72%) | < 2.2 x 10⁻¹⁶ |
| Mesenchyme | 9/ 3252 (0.28%) | 67/ 3486 (1.92%) | 1.656 x 10⁻¹⁰ |
| Endothelium | 3/ 5724 (0.05%) | 25/ 5923 (0.42%) | 4.647 x 10⁻⁵ |
| Immune cells | 504/ 14587 (3.46%) | 938/ 14770 (6.35%) | < 2.2 x 10⁻¹⁶ |
| *tdTomato*+ cells | 13/ 2412 (0.54%) | 73/ 2418 (3.02%) | 7.186 x 10⁻¹¹ |

**Extended Data Fig. 4 | Single-cell RNA-sequencing analysis confirms that *Ndufs2* deletion is specific to distal lung epithelium in NDUFS2 cKO mice.** **a**, *Mki67* expression by genotype in each tissue type or in *Sftpc* lineage (*tdTomato*)-positive cells. *Mki67*+ cells were defined as cells with normalized UMI (unique molecular identifier) counts of *Mki67* > 0 using sctransform.

*p* values by Pearson's chi-squared test. **b**, Heatmap showing expression of selected hallmark identifier genes for each clustered cell type. **c**, Relative contributions to each clustered cell type from NDUFS2 control and NDUFS2 cKO lungs. **d–e**, Expression of *Ndufs2* gene in all clusters (**d**) and epithelial subclusters (**e**). *Ndufs2* was deleted only in the distal lung epithelium.

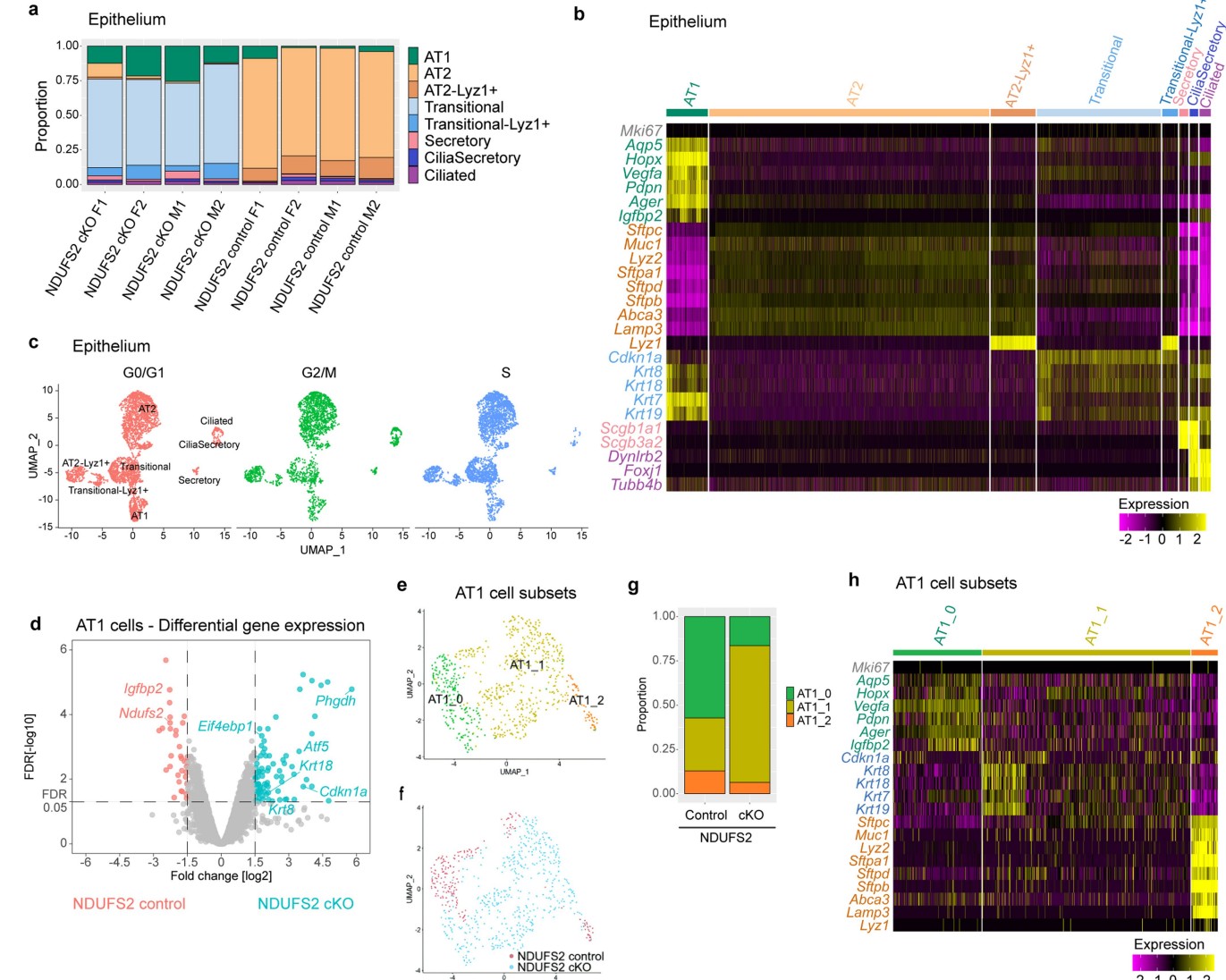

**Extended Data Fig. 5 | Postnatal transitional cells express early basal cell markers. a**, Bar plots demonstrating the composition of epithelial subclusters in each individual mouse (n = 8 mice). The transitional cell cluster was consistently expanded in all four NDUFS2 cKO mice compared with NDUFS2 control mice. M, male; F, female. **b**, Heatmap showing expression of hallmark identifier genes for each epithelial cell type. Early basal cell marker genes (*Krt8, Krt18, Krt7, Krt19*) are highly expressed in transitional cells and some of the AT1 cluster. **c**, Cell-cycle score analysis of epithelial cells was performed and plotted on a UMAP embedding. Cells predicted to be in G0/G1, G2/M, and S phases are shown in separate UMAPs, respectively. No subcluster of epithelial cells was predicted to be in a specific cell-cycle phase. **d**, Volcano plots visualizing the differential gene expression results by mouse genotype in the AT1 cluster from single-cell RNA sequencing analysis. *x* axis shows average log2 fold change, and *y* axis shows −log10 false discovery rate (FDR) *q* value. **e**, UMAP embedding of AT1 cells (n = 723 cells) colored by subcluster. **f**, UMAP plot depicting AT1 cell origins with respect to the mouse genotype. **g**, Bar plots demonstrating the composition of AT1 subclusters in NDUFS2 control and NDUFS2 cKO mice. **h**, Heatmap showing expression of epithelial marker genes in AT1 subclusters. Cells in the AT1_1 cluster express higher level of transitional cell marker genes compared to those in other AT1 subclusters.

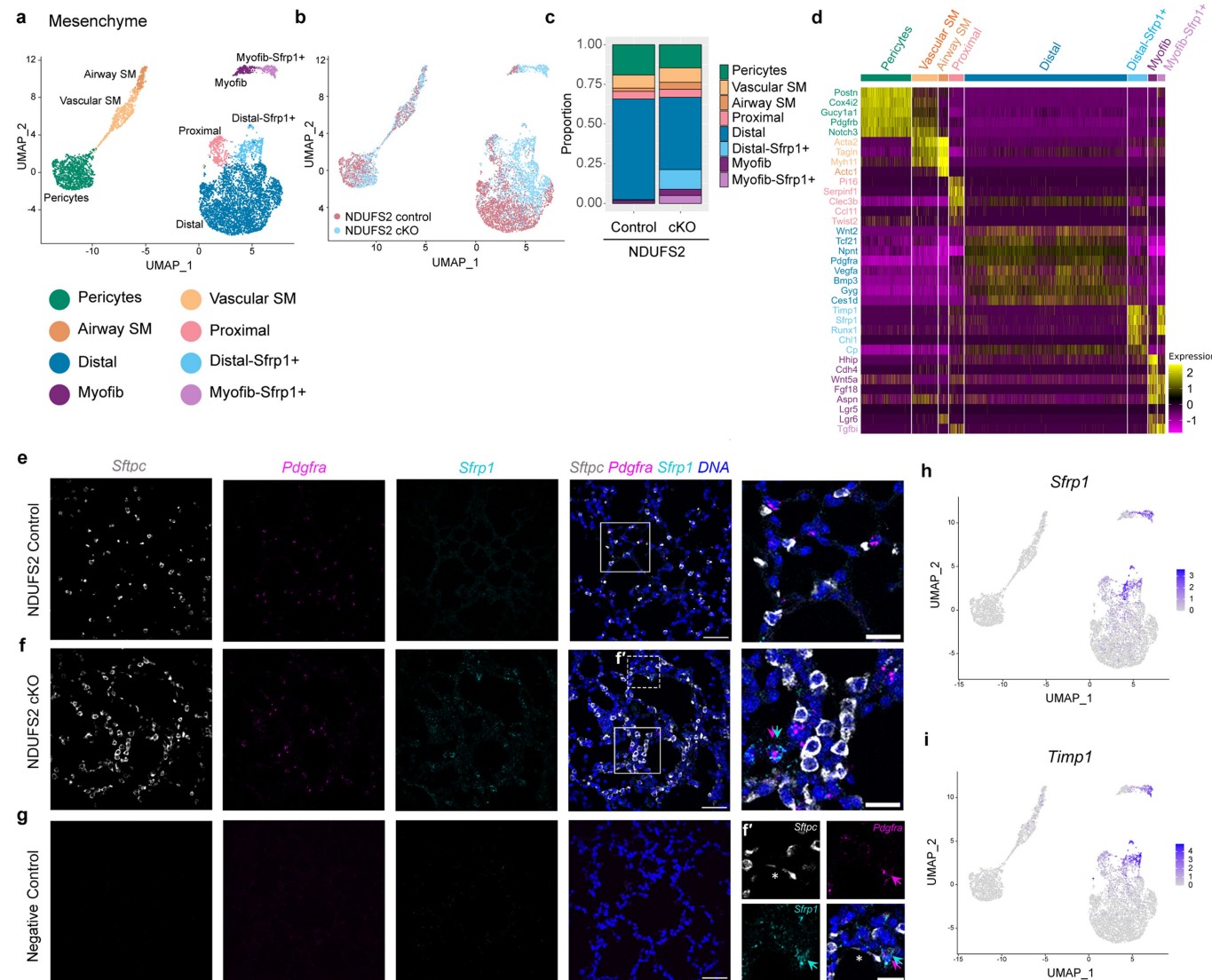

**Extended Data Fig. 6 | *Sfrp1*⁺ mesenchymal cells emerge in NDUFS2 cKO lungs.** Single-cell RNA-seq subclustering analysis with mesenchymal cells (*Col1a1*⁺ clusters except mesothelial cells, n = 6,252 cells) shows expansion of *Sfrp1*⁺ cell populations in NDUFS2 cKO mice compared to NDUFS2 control mice. **a**, UMAP embedding of lung mesenchymal cells, colored by cell type. Annotation per the recently published 3-axes classification system[74]. **b**, UMAP plot depicting mesenchymal cell origins in regard to the mouse genotype. **c**, Bar plots demonstrating the composition of mesenchymal subclusters in cells from NDUFS2 control and NDUFS2 cKO mice. **d**, Heatmap showing selected marker gene expression in different mesenchymal cellular subsets. **e–g**, Lung sections from 21-day-old NDUFS2 control and NDUFS2 cKO mice (n = 3 each

genotype, both male and female) were *in situ* RNA hybridized with indicated target probes (RNAscope®). Magnified images (boxed region) are shown on the right. Representative lung sections (**e–f**) showing subpopulations of fibroblasts, double positive for *Pdgfra*⁺ (magenta, arrow) and *Sfrp1*⁺ (cyan, arrow), emerge in NDUFS2 cKO lungs. Scale bars, 50 μm and 20 μm (magnified inset). **f′**, Magnified images of dotted line boxed region shows a fibroblast, double positive for *Pdgfra*⁺ (magenta, arrow) and *Sfrp1*⁺ (cyan, arrow) is located next to a linear thin *Sftpc*⁺ cell (gray, asterisk). Representative images (**g**) of lung sections *in situ* RNA hybridized with negative control probes. Scale bars, 50 μm and 20 μm (magnified inset). **h–i**, UMAP plots showing expression of *Sfrp1* (**h**) and *Timp1* (**i**) in subclusters of mesenchymal cells. Darker color indicates higher expression.

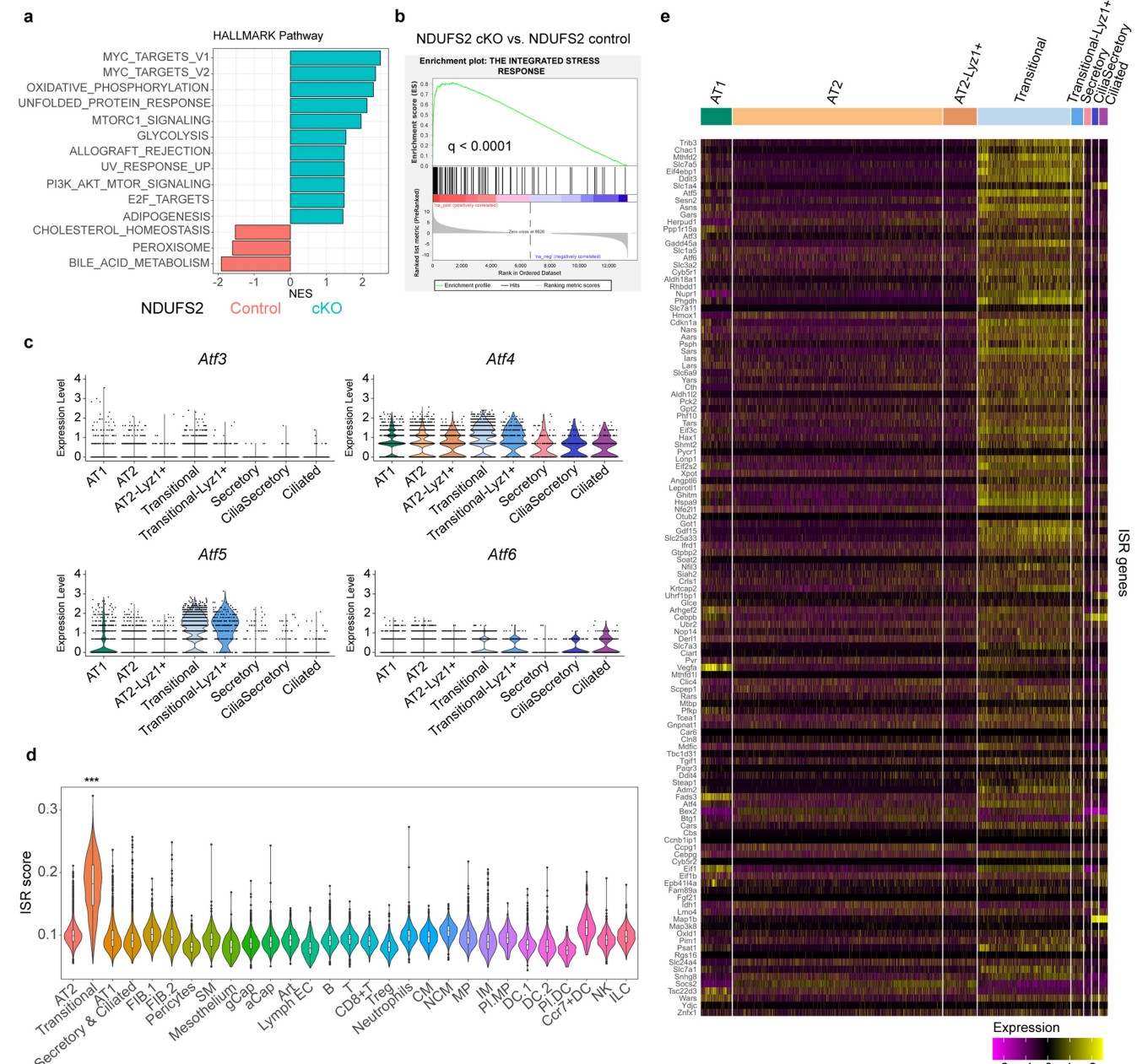

**Extended Data Fig. 7 | Postnatal transitional cells are characterized by increased ISR. a**, Gene set enrichment analysis of top gene signatures that are up-regulated (blue) or down-regulated (red) in lung epithelial cells from NDUFS2 cKO mice (n = 7) compared to NDUFS2 control mice (n = 8). FDR ≤ 0.05. **b**, Enrichment plot of the ISR signature genes in lung epithelial cells from NDUFS2 cKO mice (n = 7) compared to NDUFS2 control mice (n = 8) (normalized enrichment score; 2.80, false discovery rate q value <0.0001). **c**, Expression levels of *Atf* genes in epithelial subclusters are plotted in violin plots. **d**, Violin plots of ISR enrichment scores across all the cell clusters. $p < 2.2 \times 10^{-16}$ by Kruskal-Wallis test. Transitional cells have more enriched ISR gene signature than any other cell types (*** adjusted $p < 2.0 \times 10^{-16}$ by post-hoc pairwise Mann-Whitney test with Holm method, $p$ values are in the Source Data). **e**, Heatmap of ISR signature genes by epithelial subclusters. Results are from RNA-sequencing analysis of lung epithelial cells isolated from fifteen 35-day-old mice (**a–b**, related to Fig. 4a-b), and single-cell RNA sequencing analysis from eight 21-day-old mice (**c–e**).

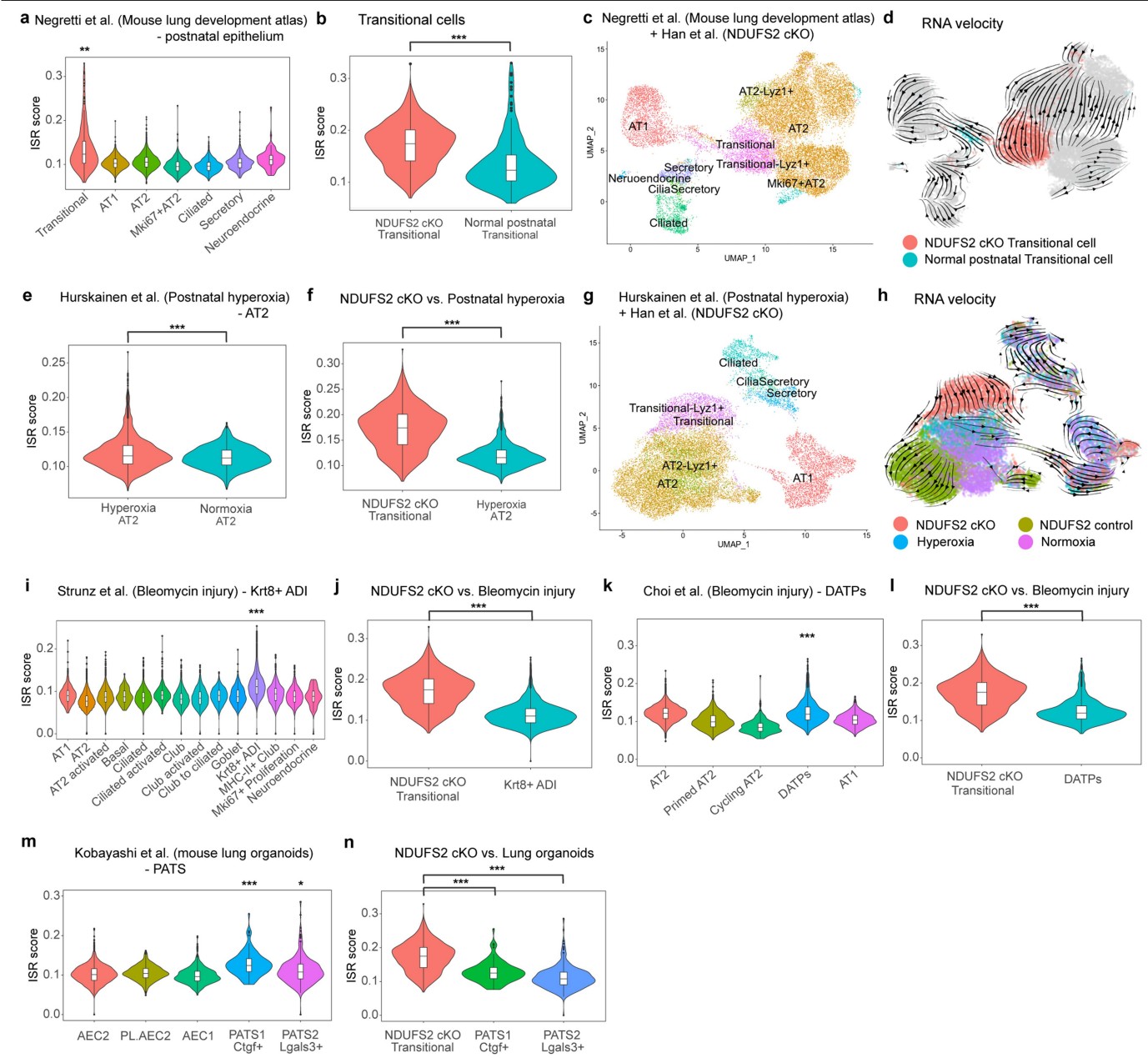

**Extended Data Fig. 8 |** See next page for caption.

**Extended Data Fig. 8 | Postnatal transitional cells from NDUFS2 cKO mice display distinct features compared to those identified in other postnatal or adult lung injury and repair models. a–d**, Re-analysis of postnatal lung epithelium from Negretti et al. (mouse lung development atlas, n = 11,807 cells)[14] integrated with our single-cell RNA-seq data of epithelium (9,322 cells). ISR enrichment scores across the postnatal epithelial cell types within the single-cell mouse lung development atlas (**a**). Higher score indicates higher enrichment of the ISR signature genes. $p < 2.2 \times 10^{-16}$ by Kruskal-Wallis test. Transitional cells have more enriched ISR gene signatures than other epithelial cells (** adjusted $p < 0.01$ by post-hoc pairwise Mann-Whitney test with Holm method, $p$ values are in the Source Data). Violin plots (**b**) showing ISR enrichment score in transitional cells from two single-cell RNA-seq data. $p < 2.2 \times 10^{-16}$ by Mann-Whitney test. UMAP embedding of integrated postnatal lung epithelial cells (**c**) colored by the cell types as annotated in original analyses. RNA-velocity vectors (**d**) were calculated and overlaid on the UMAP embedding. While normal postnatal transitional cells are predicted to differentiate to AT1 cells by RNA velocity analysis, transitional cells in NDUFS2 cKO mice are not. Please note that RNA velocity estimates should be interpreted with caution as they can be biased by a low-dimensional representation[75]. **e–h**, Re-analysis of lung epithelium from Hurskainen et al. (postnatal hyperoxia model, n = 9,975 cells)[43] integrated with our single-cell RNA-seq data of epithelium (9,322 cells). ISR enrichment scores in AT2 cells from hyperoxia-exposed lungs and normoxia-exposed lungs are shown in violin plots (**e**). $p = 1.3 \times 10^{-12}$ by Mann-Whitney test. Violin plots showing ISR enrichment scores in transitional cells from NDUFS2 cKO mice and AT2 cells from hyperoxia-exposed mice (**f**). $p < 2.2 \times 10^{-16}$ by Mann-Whitney test. UMAP embedding of integrated lung epithelial cells (**g**) colored by the same cell type as annotated in original analyses. RNA-velocity vectors were calculated and overlaid on the UMAP plots depicting cell identity by experimental conditions (**h**). **i–j**, Re-analysis of lung epithelium from Strunz et al. (adult bleomycin injury model, n = 32,559 cells)[23] integrated with our single-cell RNA-seq data of epithelium (9,322 cells). ISR enrichment scores across the epithelial cell types within Strunz et al. dataset (**i**). $p < 2.2 \times 10^{-16}$ by Kruskal-Wallis test. ISR gene signatures of Krt8+ ADI cells are more enriched compared to other cell types (*** adjusted $p < 2.0 \times 10^{-16}$ by post-hoc pairwise Mann-Whitney test with Holm method, $p$ values are in the Source Data). Violin plots (**j**) showing ISR enrichment score in transitional cells from two single-cell RNA-seq data. $p < 2.2 \times 10^{-16}$ by Mann-Whitney test. **k–l**, Re-analysis of lung epithelium from Choi et al. (adult bleomycin injury model, n = 12,179 cells)[24] integrated with our single-cell RNA-seq data of epithelium (9,322 cells). ISR enrichment scores across the epithelial cell types within Choi et al. dataset (**k**). $p < 2.2 \times 10^{-16}$ by Kruskal-Wallis test. ISR gene signatures of DATPs are more enriched than primed AT2, cycling AT2, and AT1 (*** adjusted $p < 2.0 \times 10^{-16}$ by post-hoc pairwise Mann-Whitney test with Holm method, $p$ values are in the Source Data). Violin plots (**l**) showing ISR enrichment score in transitional cells from two single-cell RNA-seq data. $p < 2.2 \times 10^{-16}$ by Mann-Whitney test. **m–n**, Re-analysis of lung epithelium from Kobayashi et al. (mouse lung organoids, n = 5,705 cells)[28] integrated with our single-cell RNA-seq data of epithelium (9,322 cells). ISR enrichment scores across the epithelial cell types within Kobayashi et al. dataset (**m**). $p < 2.2 \times 10^{-16}$ by Kruskal-Wallis test. ISR gene signatures of PATS are more enriched than those of other epithelial cells (* adjusted $p < 0.05$, *** adjusted $p < 1.0 \times 10^{-14}$ by post-hoc pairwise Mann-Whitney test with Holm method, $p$ values are in the Source Data). Violin plots (**n**) showing ISR enrichment scores in transitional cells from two single-cell RNA-seq data. $p < 2.2 \times 10^{-16}$ by Kruskal-Wallis test. *** adjusted $p < 2.0 \times 10^{-16}$ by post-hoc pairwise Mann-Whitney test with Holm method.

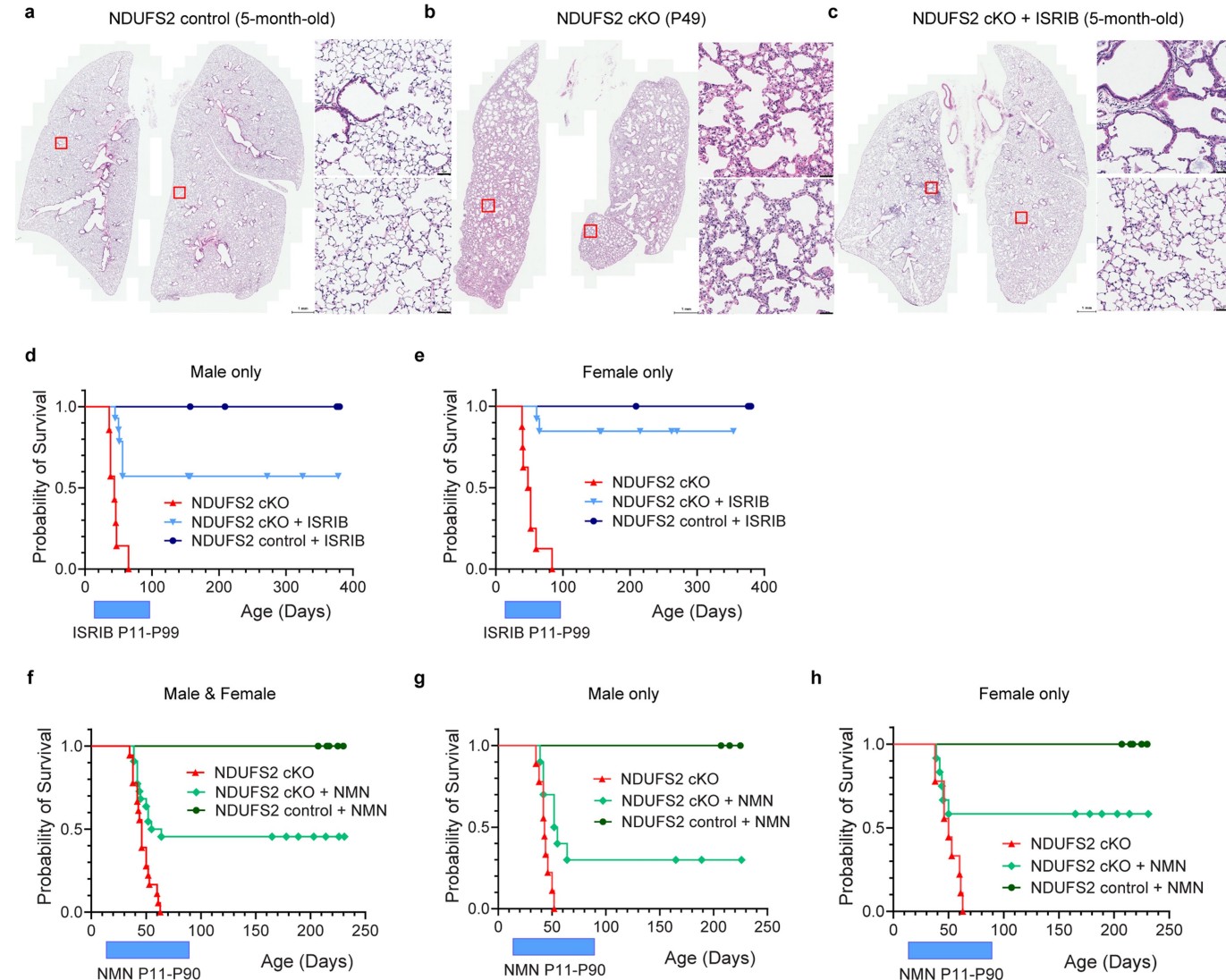

**Extended Data Fig. 9 | Inhibiting the ISR improves structural abnormalities in NDUFS2 cKO mice.** Representative images of lung histology (hematoxylin-eosin stain) from **a**, 5-month-old NDUFS2 control mice, **b**, 49-day-old NDUFS2 cKO mice that did not receive ISRIB, prior to death, and **c**, 5-month-old NDUFS2 cKO mice that received ISRIB. Scale bars, 1 mm (whole lung images) and 50 μm (close-up images). **d**, Survival of male NDUFS2 cKO mice with or without ISRIB (NDUFS2 cKO n = 7; NDUFS2 cKO + ISRIB n = 14 mice; *p* = 0.0002 by log-rank test) and male NDUFS2 control mice with ISRIB (NDUFS2 control + ISRIB n = 8 mice). **e**, Survival of female NDUFS2 cKO mice with or without ISRIB (NDUFS2 cKO n = 8; NDUFS2 cKO + ISRIB n = 13 mice; *p* < 0.0001 by log-rank test) and

female NDUFS2 control mice with ISRIB (NDUFS2 control + ISRIB n = 6 mice). **f**, Survival of NDUFS2 cKO mice with or without NMN (NDUFS2 cKO n = 18; NDUFS2 cKO + NMN n = 22 mice; p = 0.0010 by log-rank test) and NDUFS2 control mice with NMN (NDUFS2 control + NMN n = 13). **g**, Survival of male NDUFS2 cKO mice with or without NMN (NDUFS2 cKO n = 9; NDUFS2 cKO + NMN n = 10 mice; p = 0.0087 by log-rank test) and male NDUFS2 control mice with NMN (NDUFS2 control + NMN n = 7). **h**, Survival of female NDUFS2 cKO mice with or without NMN (NDUFS2 cKO n = 9; NDUFS2 cKO + NMN n = 12 mice; p = 0.0284 by log-rank test) and female NDUFS2 control mice with NMN (NDUFS2 control + NMN n = 6).

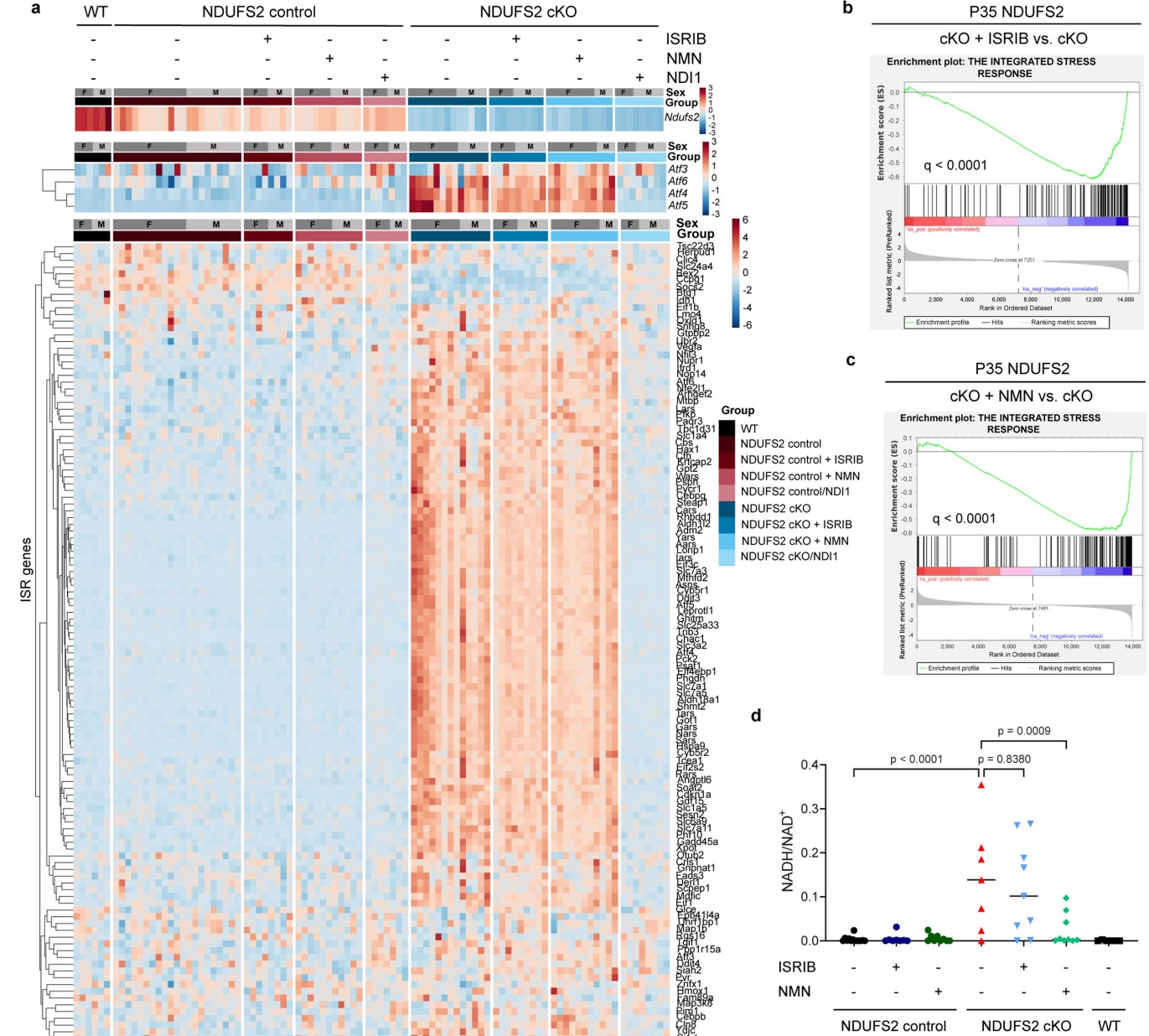

**Extended Data Fig. 10 | Administration of ISRIB or NMN decreases the pathologic ISR activation observed in NDUFS2 cKO mice. a–c**, RNA-sequencing analysis of lung epithelial cells isolated from 35-day-old mice (WT n = 6; NDUFS2 control n = 21; NDUFS2 control + ISRIB n = 8; NDUFS2 control + NMN n = 11; NDUFS2 control/NDI1 n = 7; NDUFS2 cKO n = 13; NDUFS2 cKO + ISRIB n = 9; NDUFS2 cKO + NMN n = 11; NDUFS2 cKO/NDI1 n = 8 mice). Data in Fig. 4a-b were included in the analysis. Heatmaps of *Ndufs2* and ISR signature gene transcripts (**a**). Enrichment plots of the ISR signature genes in lung epithelial cells from NDUFS2 cKO mice with vs. without ISRIB (**b**) (normalized enrichment score; −2.51, false discovery rate q value <0.0001), and NDUFS2 cKO mice with vs. without NMN (**c**) (normalized enrichment score; −2.39, false discovery rate q value <0.0001). **d**, Intracellular $NADH/NAD^+$ ratios in lung epithelial cells from 35-day-old mice (NDUFS2 control n = 13; NDUFS2 control + ISRIB n = 8; NDUFS2 control + NMN n = 9; NDUFS2 cKO n = 7; NDUFS2 cKO + ISRIB n = 9; NDUFS2 cKO + NMN n = 9; WT n = 6 mice). $p < 0.0001$ by ANOVA. Adjusted p values by Šídák's multiple comparisons test were provided in the graph.

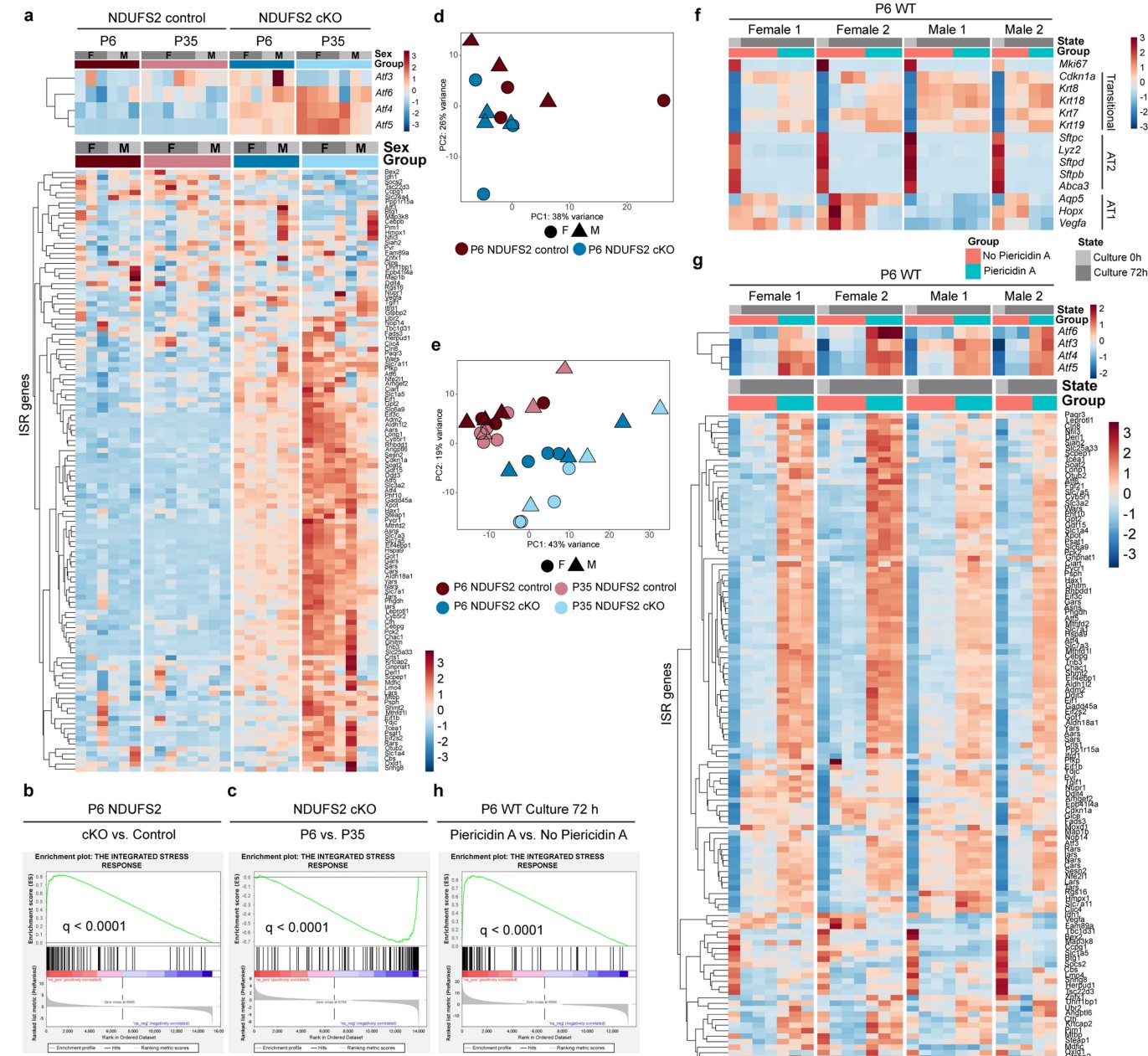

**Extended Data Fig. 11 | Inhibition of mitochondrial complex I prevents AT2 to AT1 differentiation *in vitro*. a**–**e**, RNA sequencing analysis of lung epithelial cells isolated from 6-day-old mice and 35-day-old mice (P6 NDUFS2 control n = 6, P6 NDUFS2 cKO n = 6; P35 NDUFS2 control n = 8; P35 NDUFS2 cKO n = 7 mice). P35 data is from Fig. 4a,b. Heatmaps of ATF and ISR signature gene transcripts (**a**). **b**, Enrichment plot of the ISR signature genes in lung epithelial cells from P6 NDUFS2 cKO vs. P6 NDUFS2 control mice (normalized enrichment score; 3.10, false discovery rate q value <0.0001). **c**, Enrichment plot of the ISR signature genes in lung epithelial cells from P6 NDUFS2 cKO vs. P35 NDUFS2 cKO mice (normalized enrichment score; −2.71, false discovery rate q value <0.0001).

**d**–**e**, Principal components analysis (PCA) shows transcriptomic signatures of NDUFS2 control and NDUFS2 cKO lung epithelial cells at P6 were not clearly separated. **f**–**h**, RNA sequencing analysis of 2-D cultured AT2 cells isolated from 6-day-old wild-type (WT) mice. Cells were incubated with the mitochondrial complex I inhibitor, piericidin A (500 nM) for 16 h before processed for RNA isolation. Heatmaps of expression of cell type marker genes (**f**), and ISR signature genes including ATF transcripts (**g**). Enrichment plot of the ISR signature genes (**h**) in 2-D cultured AT2 cells with vs. without piericidin A (normalized enrichment score; 2.92, false discovery rate q value <0.0001).

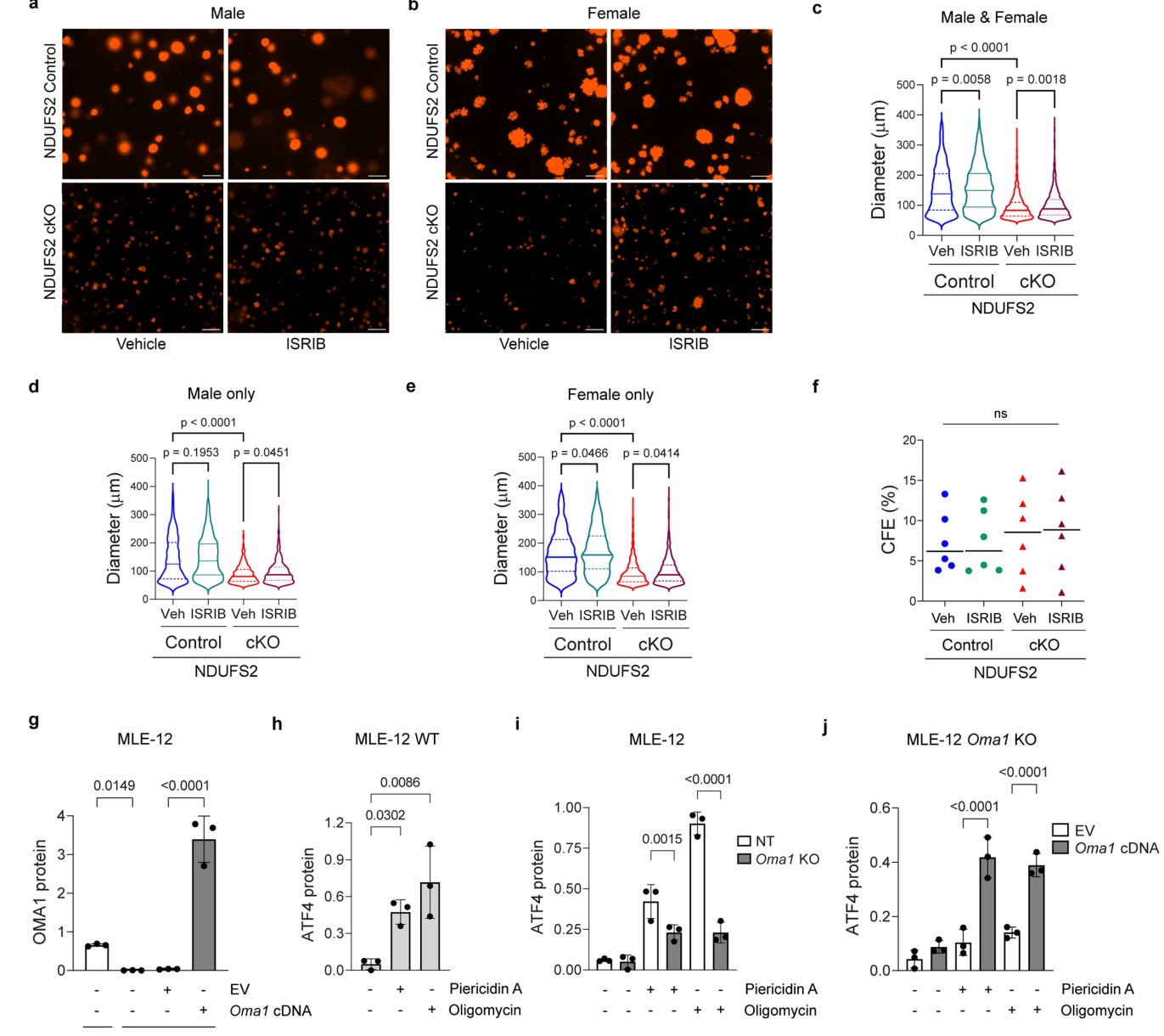

**Extended Data Fig. 12 | ISRIB improves limited cell growth of NDUFS2 cKO AT2 cells in 3-D organoid culture. a–f**, 3-D alveolar organoid cultures with AT2 cells isolated from 6-day-old *Sftpc* lineage traced (*tdTomato*) NDUFS2 control and NDUFS2 cKO mice (n = 6 mice (three males and three females) per genotype with technical replicates). Representative images of alveolar organoid cultures (**a**–**b**). Scale bar, 500 μm. Violin plots of organoid diameters (**c**–**e**), a proxy for alveolar organoid differentiation. The diameters of NDUFS2 cKO organoids were smaller than those of NDUFS2 control organoids, which was improved by ISRIB administration. *p* values by Šídák's multiple comparisons test. Colony forming efficiency (CFE), a proxy for organoid proliferation, is shown (**f**). Proliferation is preserved in NDUFS2 cKO AT2 cells. Lines represent median.

*p* = 0.9652 by Kruskal-Wallis test. **g**, Immunoblot analysis of OMA1 protein adjusted by COFILIN in mouse lung epithelial cell line (MLE-12). Data represents mean ± S.D. of three independent experiments. *q* values by two-stage linear step-up procedure of Benjamini, Krieger and Yekutieli. **h–j**, Immunoblot analysis of ATF4 protein adjusted by COFILIN in MLE-12. Data represents mean ± S.D. of three independent experiments. Cells were incubated with piericidin A (500 nM) or oligomycin (100 nM) for 16 h to inhibit complex I or V (positive control), respectively. *q* values by two-stage linear step-up procedure of Benjamini, Krieger and Yekutieli. EV, empty vector; NT, non-targeting control. All cell culture media contained aspartate and asparagine.

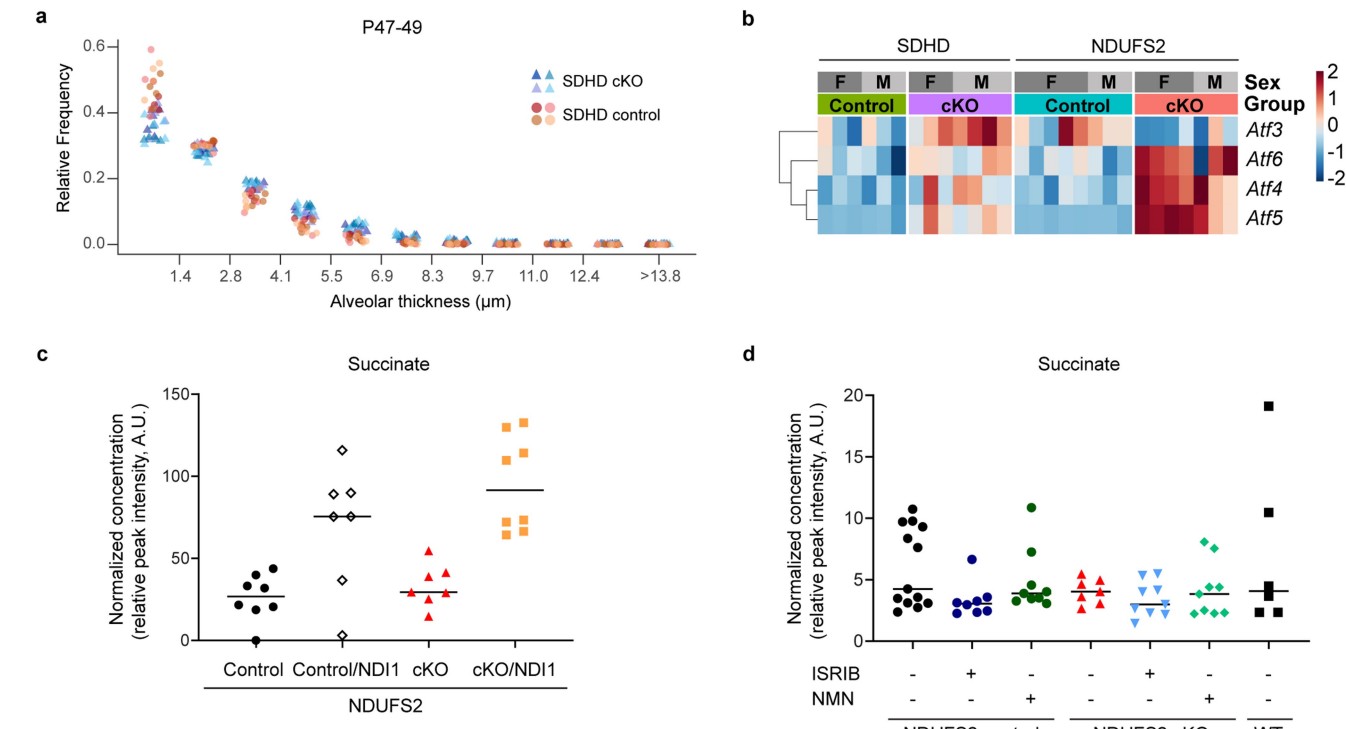

**Extended Data Fig. 13 | Loss of mitochondrial complex II in lung epithelial cells induces mild ISR, which is lower than the ISR induction due to loss of mitochondrial complex I. a**, The frequency distribution of alveolar thickness measured in hematoxylin-eosin stained lung histology of 47–49 day-old mice (n = 4 mice, two males and two females per genotype). 4–6 randomly selected fields of view from each mouse were evaluated. The x axis shows alveolar thickness bins, and the y axis shows the number of alveolar pixels that belong to the respective alveolar thickness bin normalized to the total alveolar pixel count in the image. Each animal is represented by its own color. Statistical significance for genotype was calculated based on F-test for a linear model

$(p = 4.65 \times 10^{-3})$. **b**, RNA-seq analysis of lung epithelial cells from 35-day old mice (SDHD control n = 6; SDHD cKO n = 7; NDUFS2 control n = 8; NDUFS2 cKO n = 7 mice). Heatmap of ATF transcripts in lung epithelial cells. **c**, Relative abundance of succinate in lung epithelial cells isolated from 35-day old mice (NDUFS2 control n = 8; NDUFS2 control/NDI1 n = 7; NDUFS2 cKO n = 7; NDUFS2 cKO/NDI1 n = 8 mice). **d**, Relative abundance of succinate in lung epithelial cells isolated from 35-day old mice (NDUFS2 control n = 13; NDUFS2 control + ISRIB n = 8; NDUFS2 control + NMN n = 9; NDUFS2 cKO n = 7; NDUFS2 cKO + ISRIB n = 9; NDUFS2 cKO + NMN n = 9; WT n = 6 mice). Lines represent median. $p = 0.2333$ by Kruskal-Wallis test.

## Extended Data Table 1 | Identified cell populations

| Abbreviation | Cell type | Abbreviation | Cell type |
|---|---|---|---|
| aCap | Alveolar capillary endothelial cells | Lymph EC | Lymphatic endothelial cells |
| Art | Arterial endothelial cells | Mesothelium | Mesothelial cells |
| AT1 | alveolar epithelial type I cells | MP | Macrophages |
| AT2 | Alveolar epithelial type II cells | NCM | Non-classical monocytes |
| B | B cells | Neutrophils | Neutrophils |
| Ccr7+ DC | CCR7+ dendritic cells | NK | Natural killer cell |
| CD8+ T | CD8+ T cells | Pericytes | Pericytes |
| CM | Classical monocytes | Pl.DC | Proliferating dendritic cells |
| DC.1 | Dendritic cells 1 | Pl.MP | Proliferating macrophages |
| DC.2 | Dendritic cells 2 | Secretory & Ciliated | Club cells and ciliated cells |
| FIB.1 | Fibroblasts 1 | SM | Smooth muscle cells |
| FIB.2 | Fibroblasts 2 | T | T cells |
| gCap | General capillary endothelial cells | Transitional | Transitional cells |
| ILC | Innate lymphoid cells | Treg | Regulatory T cells |
| IM | Interstitial macrophages | | |

Cell types identified from single-cell RNA sequencing analysis in Fig. 3a are listed.

# Reporting Summary

## Statistics

For all statistical analyses, confirm that the following items are present in the figure legend, table legend, main text, or Methods section.

| n/a | Confirmed | |
|---|---|---|
| ☐ | ☒ | The exact sample size ($n$) for each experimental group/condition, given as a discrete number and unit of measurement |
| ☐ | ☒ | A statement on whether measurements were taken from distinct samples or whether the same sample was measured repeatedly |
| ☐ | ☒ | The statistical test(s) used AND whether they are one- or two-sided<br>*Only common tests should be described solely by name; describe more complex techniques in the Methods section.* |
| ☐ | ☒ | A description of all covariates tested |
| ☐ | ☒ | A description of any assumptions or corrections, such as tests of normality and adjustment for multiple comparisons |
| ☐ | ☒ | A full description of the statistical parameters including central tendency (e.g. means) or other basic estimates (e.g. regression coefficient) AND variation (e.g. standard deviation) or associated estimates of uncertainty (e.g. confidence intervals) |
| ☐ | ☒ | For null hypothesis testing, the test statistic (e.g. $F$, $t$, $r$) with confidence intervals, effect sizes, degrees of freedom and $P$ value noted<br>*Give P values as exact values whenever suitable.* |
| ☒ | ☐ | For Bayesian analysis, information on the choice of priors and Markov chain Monte Carlo settings |
| ☐ | ☒ | For hierarchical and complex designs, identification of the appropriate level for tests and full reporting of outcomes |
| ☐ | ☒ | Estimates of effect sizes (e.g. Cohen's $d$, Pearson's $r$), indicating how they were calculated |

*Our web collection on statistics for biologists contains articles on many of the points above.*

## Software and code

Policy information about availability of computer code

| | |
|---|---|
| Data collection | Oxygen consumption data was collected using Wave 2.6.3.5 software. Immunoblot data was collected using a Wes by ProteinSimple and Compass for SW software 5.0.1. Nikon A1C confocal microscope, Nikon Ti2 Widefield microscope, and TissueGnostics imaging software system (TissueFAXS 7.1) were used to obtain images. RNA-seq data was collected using Illumina NextSeq 500 system. Raw BCL read files were demultiplexed using bcl2fastq V2.20.0 (Illumina), and trimmed using Trimmomatic (version 0.39). For metabolomics, high-resolution HPLC–tandem mass spectrometry was performed on a Q-Exactive (ThermoFisher Scientific) in line with an electrospray source and an UltiMate 3000 (ThermoFisher Scientific) and data were collected using Xcalibur 4.1 software. Single-cell RNA-seq data was obtained from HiSeq 4000 instrument (Illumina), and raw sequencing reads were processed using CellRanger v6.0.1. |
| Data analysis | Image processing and analysis was performed using freely (ImageJ/Fiji 1.53 [NIH]) or commercially available software (Nikon Elements (5.11.00)). Immunoblot data were analyzed using Compass for SW software 5.0.1 (ProteinSimple).<br>RNA-seq data was analyzed using the R package edgeR. Reads were then aligned to the GRCm39 reference genome using the STAR aligner V2.7.7, and counts were calculated using HTseq V0.11.0. The ComBat-seq package was used to adjust for batch effect on RNA-seq count data. Metabolomic data were analyzed using the MetaboAnalyst software V5.0 and the MetaboAnalystR package V4.1.2. Single-cell RNA-seq data analyses were performed using Seurat v4.0.6 in R v4.1.2 and Scanpy v1.8.1 in Python v3.8.3. Doublets were removed using Scrublet v0.2.1 from each library. RNA velocity was calculated with velocyto v0.17 and scVelo v0.2.4. UCell algorithm was used to evaluate gene signature in single-cell datasets. All code used for analysis is available at https://github.com/MinhoLee-DGU/2023.Han.et.al.Nature<br>All other statistical analyses were performed using GraphPad Prism 9.5.0. |

For manuscripts utilizing custom algorithms or software that are central to the research but not yet described in published literature, software must be made available to editors and reviewers. We strongly encourage code deposition in a community repository (e.g. GitHub). See the Nature Portfolio guidelines for submitting code & software for further information.

## Data

Policy information about availability of data

All manuscripts must include a data availability statement. This statement should provide the following information, where applicable:
- Accession codes, unique identifiers, or web links for publicly available datasets
- A description of any restrictions on data availability
- For clinical datasets or third party data, please ensure that the statement adheres to our policy

All raw sequencing data (.fastq) generated in this study are available at the NCBI BioProject with the following Accession IDs: PRJNA865889, PRJNA940730, PRJNA940746, PRJNA940973, PRJNA940986, and PRJNA940992.
Strunz et al. (GSE141259), Choi et al. (GSE145031), Kobayashi et al. (GSE141634), Negretti et al. (PRJNA674755 and PRJNA693167), Hurskainen et al. (PRJNA637911), Molecular Signatures Database (MSigDB), and GRCm39 reference genome were used for analysis.

# Field-specific reporting

Please select the one below that is the best fit for your research. If you are not sure, read the appropriate sections before making your selection.

☒ Life sciences        ☐ Behavioural & social sciences        ☐ Ecological, evolutionary & environmental sciences

For a reference copy of the document with all sections, see nature.com/documents/nr-reporting-summary-flat.pdf

# Life sciences study design

All studies must disclose on these points even when the disclosure is negative.

| | |
|---|---|
| Sample size | All experiments were performed using sample sizes based on standard protocols in the field. We made every effort to avoid excessive or needless use of animals. No statistical tests were used to predetermine sample sizes. We used sample sizes commonly used in literature in the field. We used statistical analysis consistent with the sample size for each experiment and found sufficient statistical power with the sample sizes used in our study. |
| Data exclusions | No animal data were excluded from analyses.<br>Single-cell RNA-seq: Poor quality cells with less than 500 detected genes and a high percentage of mitochondrial genes (>25%) were excluded from analyses. |
| Replication | All experimental data were reliably reproduced in multiple independent experiments as indicated in the figure legends. For in vivo experiments, multiple mice were used in at least two independent cohorts to ensure reproducibility. |
| Randomization | Experiments were not randomized. Transgenic mice were predetermined by mouse genotype and therefore could not be randomized. All mice were sex- and age-matched, and littermates when possible. |
| Blinding | Investigators were not blinded. Blinding was not relevant in this study, as groups consisted of previously genotyped mice or treated cell lines in order to have correct experimental and control groups. |

# Reporting for specific materials, systems and methods

We require information from authors about some types of materials, experimental systems and methods used in many studies. Here, indicate whether each material, system or method listed is relevant to your study. If you are not sure if a list item applies to your research, read the appropriate section before selecting a response.

### Materials & experimental systems

| n/a | Involved in the study |
|---|---|
| ☐ | ☒ Antibodies |
| ☐ | ☒ Eukaryotic cell lines |
| ☒ | ☐ Palaeontology and archaeology |
| ☐ | ☒ Animals and other organisms |
| ☒ | ☐ Human research participants |
| ☒ | ☐ Clinical data |
| ☒ | ☐ Dual use research of concern |

### Methods

| n/a | Involved in the study |
|---|---|
| ☒ | ☐ ChIP-seq |
| ☒ | ☐ Flow cytometry |
| ☒ | ☐ MRI-based neuroimaging |

## Antibodies

| | |
|---|---|
| Antibodies used | Antibodies used for immunoblot: anti-Vinculin (abcam, ab129002, clone EPR8185; 1:500 dilution); anti-NDUFS2 (abcam, ab192022, clone EPR16266; 1:200 dilution); anti-Oma1 (SCBT, sc-515788, clone H-11; 1:50 dilution), anti-ATF4 (CST, 11815S, clone D4B8; 1:50 |

dilution), and anti-Cofilin (CST, 5175T, clone D3F9; 1:10,000 or 1:30,000 dilution).

Antibodies used for immunohistochemistry: anti-CD45 (abcam, ab10558, 1:1500 dilution), anti-Ki67 (abcam, ab16667, clone SP6; 1:100 dilution), anti-proSftpC (Millipore, AB3786; 1:500 dilution), anti-Podoplanin (abcam, ab11936, clone RTD4E10; 1:2000 dilution).

Antibodies used for cell isolation: anti-mouse biotin-conjugated CD45 (BD Biosciences, 553078, clone 30-F11), anti-mouse biotin-conjugated CD31 (BD Biosciences 553371, clone MEC 13.3), anti-mouse biotin-conjugated CD16/CD32 (BD Biosciences 553143, clone 2.4G2), and anti-mouse EpCAM microbeads (Miltenyi Biotec, 130-105-958) without dilution.

| Validation | The antibodies used in this study were tested by the manufacturer.

-anti-Vinculin (abcam, ab129002, clone EPR8185). This antibody can be found in 206 citations. The manufacturer also provides antibody testing data: https://www.abcam.com/vinculin-antibody-epr8185-ab129002.html

-anti-NDUFS2 (abcam, ab192022, clone EPR16266). This antibody can be found in 3 citations. The manufacturer also provides antibody testing data and knockout validation: https://www.abcam.com/ndufs2-antibody-epr16266-ab192022.html

-anti-Oma1 (SCBT, sc-515788, clone H-11). This antibody can be found in 31 citations. The manufacturer also provides antibody testing data: https://www.scbt.com/p/oma1-antibody-h-11

-anti-ATF4 (CST, 11815S, clone D4B8). This antibody can be found in 661 citations. The manufacturer also provides antibody testing data: https://www.cellsignal.com/products/primary-antibodies/atf-4-d4b8-rabbit-mab/11815

-anti-Cofilin (CST, 5175T, clone D3F9). This antibody can be found in 245 citations. The manufacturer also provides antibody testing data: https://www.cellsignal.com/products/primary-antibodies/cofilin-d3f9-xp-rabbit-mab/5175

-anti-CD45 (abcam, ab10558). This antibody can be found in 282 citations. The manufacturer also provides antibody testing data: https://www.abcam.com/cd45-antibody-ab10558.html

-anti-Ki67 (abcam, ab16667, clone SP6). This antibody can be found in 1744 citations. The manufacturer also provides antibody testing data and knockout validation: https://www.abcam.com/ki67-antibody-sp6-ab16667.html

-anti-proSftpC (Millipore, AB3786). This antibody can be found in 16 citations. The manufacturer also provides antibody testing data: https://www.emdmillipore.com/US/en/product/Anti-Prosurfactant-Protein-C-proSP-C-Antibody,MM_NF-AB3786

-anti-Podoplanin (abcam, ab11936, clone RTD4E10). This antibody can be found in 65 citations. The manufacturer also provides antibody testing data:  https://www.abcam.com/podoplanin--gp36-antibody-rtd4e10-bsa-and-azide-free-ab11936.html

-anti-mouse biotin-conjugated CD45 (BD Biosciences, 553078, clone 30-F11). This antibody can be found in 4 citations. The manufacturer also provides antibody testing data: https://www.bdbiosciences.com/en-us/products/reagents/flow-cytometry-reagents/research-reagents/single-color-antibodies-ruo/biotin-rat-anti-mouse-cd45.553078

-anti-mouse biotin-conjugated CD31 (BD Biosciences 553371, clone MEC 13.3). This antibody can be found in 14 publications. The manufacturer also provides antibody testing data: https://www.bdbiosciences.com/en-us/products/reagents/flow-cytometry-reagents/research-reagents/single-color-antibodies-ruo/biotin-rat-anti-mouse-cd31.553371

-anti-mouse biotin-conjugated CD16/CD32 (BD Biosciences 553143, clone 2.4G2). This antibody can be found in 15 citations. The manufacturer also provides antibody testing data: https://www.bdbiosciences.com/en-lu/products/reagents/flow-cytometry-reagents/research-reagents/single-color-antibodies-ruo/biotin-rat-anti-mouse-cd16-cd32.553143

-anti-mouse EpCAM microbeads (Miltenyi Biotec, 130-105-958). The antibody can be found in 5 publications. The manufacturer provides antibody testing data: https://www.miltenyibiotec.com/US-en/products/cd326-epcam-microbeads-mouse.html#130-105-958

# Eukaryotic cell lines

Policy information about cell lines

| Cell line source(s) | MLE-12 and 293T were purchased from ATCC. |

| Authentication | Neither of the cell lines used were authenticated. |

| Mycoplasma contamination | Cell lines tested negative for mycoplasma contamination. Cells were checked periodically. |

| Commonly misidentified lines (See ICLAC register) | These cell lines are not listed in the database of commonly misidentified cell lines maintained by ICLAC. |

# Animals and other organisms

Policy information about studies involving animals; ARRIVE guidelines recommended for reporting animal research

| Laboratory animals | Genetically modified mice on C57Bl/6J background were used. The age of the mice used in this study is ranged from newborn to 25-month-old, and specified in figure legends for each experiment. Both male and female mice were used in all experiments. Animals were housed at Northwestern University animal facility, where the animals were on a 14-h on, 10-h off light cycle, room temperature |

range was 21-23°C, and humidity was within 30-70 % range compliant to the guidelines.
Ndufs2 floxed mice were genotyped using the following primers: Forward 5' - ATAAGAGTGGATAGGATGTTT - 3' ; flox reverse 5' - CATTTCTCCCTTCCCGTC - 3' ; and null reverse 5'-AGTGGCAGAACAATAGAGTGATCCAGGG-3'
Sdhd floxed mice were genotyped using the following primers: Sdhd Forward 5' - GGAAGGCTCCAAGGGTGCAG - 3' ; and Sdhd Reverse 5' - CACATACACGCAGGCACTGG - 3'
SFTPC-Cre mice were genotyped using the following primers: Cre Forward: 5'-GCAGAACCTGAAGATGTTCGCGAT-3' ; Cre Reverse: 5'-AGGTATCTCTGACCAGAGTCATCC-3' ; Internal Control Forward: 5'-CTAGGCCACAGAATTGAAAGATCT-3' ; and Internal Control Reverse: 5'-GTAGGTGGAAATTCTAGCATCATCC-3'
NDI1-LSL mice were genotyped using the following primers: Rosa26 Fwd  5' – GAGTTCTCTGCTGCCTCCTG; Rosa26 Rev 5' – CCGACAAAACCGAAAATCTG; and WPRE B Fwd 5' – GACGAGTCGGATCTCCCTTT.
ROSA26Sor CAG-tdTomato mice were genotyped using the following primers: 5'-GGC ATT AAA GCA GCG TAT CC-3' ;  5'-CTG TTC CTG TAC GGC ATG G-3' ; 5'-CCG AAA ATC TGT GGG AAG TC-3' ; and  5'-AAG GGA GCT GCA GTG GAG TA-3'

| Wild animals | This study did not involve wild animals. |
| Field-collected samples | This study did not involve samples collected from the field. |
| Ethics oversight | All mouse work was done in accordance with Northwestern University Institutional Animal Care and Use Committee (IACUC). |

Note that full information on the approval of the study protocol must also be provided in the manuscript.

