## [Peer Review File · Nature]

Manuscript Title: Mitochondrial integrated stress response controls lung epithelial cell fate

Reviewer Comments & Author Rebuttals

Reviewer Reports on the Initial Version:

Referees' comments:

Referee #1 (Remarks to the Author):

In the manuscript „Mitochondrial integrated stress response controls lung epithelial cell fate during development“ by Han et al., the authors generated a mouse model with a lung epithelial-specific deletion of the mitochondrial complex I subunit NDUFS2 (NDUFS2 cKO) to investigate its role in lung development. NDUFS2 cKO mice die at an early postnatal age of approximately seven weeks due to respiratory failure accompanied by abnormal alveolar morphology and fluid accumulation in the lung, as revealed by histological examination. Using bulk and single-cell RNA-sequencing combined with immune histological experiments, the authors suggest a block in differentiation of lung epithelial cells in the transitional stage between AT2-to-AT1 cells as the molecular mechanism to explain the observed phenotype. The authors also claim that the cell fate defect is caused by the mitochondrial dysfunction elicited by the loss of complex I activity and the subsequent downstream activation of the mitochondrial integrated stress response. Expressing yeast NDI1 (an enzyme capable of NAD⁺ regeneration but without proton pumping activity present in complex I) in the NDUFS2 cKO mice rescued the histological phenotype and the death of these mice pinpointing the failure of NAD⁺ regeneration as the primary cause of the death of the cNDUFS2 mice. The NDUFS2 cKO mice phenotype could also be rescued by intraperitoneal injection of ISRIB, an inhibitor of the ISR, strengthening the hypothesis of pathogenic hyperactivation of the ISR as the cause of the respiratory failure.

This study presents the exciting hypothesis that the extent of mitochondrial dysfunction and subsequent ISR activation is vital for the switching between physiological to pathological cellular differentiation in the lung. Although potentially interesting, this study suffers from major technical and conceptual limitations that undermine the strength of the authors' conclusions. In addition, the role of mitochondrial function in regulating postnatal development and the effects of OXPHOS impairment and metabolic imbalances on ISR upregulation are already well-described. Therefore, the paper lacks the novelty required for this journal unless some more mechanistic aspects are provided.

Major Comments

1. The characterization of the mouse model is not satisfactory. First, the authors use NDUFS2^{fl/-}; SFTPC-Cre^{+/+} as control mice, but at least for the initial description of the mouse model (experiments shown in Fig 1) a proper control should be used (NDUFS2^{fl/fl}; SFTPC-Cre^{-/-}) to prove that the chosen control mice behave indeed like Cre-negative mice. Additionally, it is unclear why the control mice in Fig. 1a,b differ from those used in the rest of the manuscript. To ensure that the NDUFS2 KO is cell-specific, it is recommended that the authors perform an immunohistochemistry analysis for NDUFS2 in lung epithelial cells. Additionally, the leakage of the Cre should be assessed by looking at the tdTomato-expression combined with specific AT1 and AT2 markers by histology. On a related note, it is strongly recommended that the impaired differentiation, shown only by sc-RNA-seq, is further validated (for example, by an in vitro differentiation assay). These control experiments will rule out the possibility that wild-type cells repopulate the regions where NDUFS2 KO cells are located. This hypothesis is corroborated by the observation that the increased cellularity in the NDUFS2 KO animals arises from hypertrophic AT2-

expressing markers, which according to the authors' model, should not be NDUFS2 KO cells that express AT1 markers instead. Addressing this question is crucial to determine whether the rescue of the lung defects caused by ISRIB is caused by the repopulation of NDUFS2 KO cells or of wild-type cells (see comment 2 below).

The age of mice used in the study differs between different experiments (for instance, some experiments were performed using 11-day-old mice, whereas others used 43-day-old mice). Similarly, several analyses were performed at different stages during the mouse embryonic and postnatal development (for example, some experiments were performed at P34, while others at P47 or P49), making conclusions difficult to draw. Although these parameters might not significantly affect the results, consistency between all experiments is required. Finally, for metabolomic and RNA-seq analyses, the authors used both male and female mice, whereas for all other experiments, it is not clear whether they used mice of a specific gender. Given that the two genders show a difference in their metabolic and transcriptional responses (Fig.2c and Fig.4b, respectively), it would be important to test whether this affects the conclusions or at least discuss these differences appropriately.

2. The authors' central claim is that the primary cause of the respiratory failure in cNDUFS2 mice is the perturbed NAD⁺ generation causing the activation of the mitoISR and subsequently the block in differentiation. There are a few issues with this model.

First, given the central role of the NAD⁺/NADH ratio for all conclusions of this paper, the authors should provide this measurement for every used mouse model (cNDUFS2, cNDUFS2/NDI1, Sdh) and applied treatment (ISRIB). Additional measurements of NAD/NADH imbalance should also be included, such as lactate to pyruvate ratio. These critical controls will confirm that the NAD⁺/NADH ratio is affected as expected and is indeed the trigger for the observed phenotypes. In addition, the authors should show that altering NADH/NAD ratio without affecting the ETC would phenocopy the effect (using approaches proposed by Mick et al. eLife 2020, for instance, using LbNOX). It is also unclear how these NADH/NAD ratio changes can elicit the mitoISR. The authors should provide at least a minimal mechanism to corroborate this connection. For instance, is it a low asparagine level (due to a decrease in aspartate) as proposed by Mick et al. eLife 2020?

Secondly, the authors should corroborate the presence of a mitoISR by reporting additional data beyond the transcriptional signature (changes at the protein levels of ISR markers, and potentially the expected metabolic changes, considering that they have the data). The authors should distinguish whether what they observed is triggered by the integrated stress response coming from the mitochondrial dysfunction or the ER stress, considering the current lack of additional ISR markers. They could distinguish mito- and ER stress-induced ISR by looking at the cleavage of DELE1 by western blot (like in Fessler et al., 2020; Guo et al., 2020). In addition, the authors should confirm that ISRIB reverses these parameters, and acts specifically on lung epithelial cells on NDUFS2 KO lung cells (using the markers described above). It will be vital to confirm that ISRIB restores the proportion of AT2 vs AT1 cells in the lung and that these cells are NDUFS2 KO. These experiments would exclude that any non-cell-autonomous effect is responsible for rescuing the phenotype.

Finally, the mechanism behind the dose-dependent effect of ISR on cellular differentiation has not been described. The authors should clarify how the ISR can affect the expression of genes involved in differentiation and whether acute vs chronic ISR activation affects different transcriptional programmes. For instance, the authors should consider that some factors downstream of complex I loss (including metabolic and transcriptional changes) restored by NDI could contribute to the phenotype. Intriguingly, both SDHD KO and NDI1 (in cNDUFS2 KO) expression led to succinate accumulation. The authors should discuss the possibility that succinate accumulation is sufficient to promote alveolar development and should provide convincing evidence to exclude this hypothesis and consider the established role of this metabolite in orchestrating chromatin changes. In addition, the authors should be able to explain how the suppression of ISR could revert the

differentiation phenotype in cells with mitochondrial dysfunction. Considering their model, would inhibiting the transient ISR observed in normal development be detrimental? Appropriate characterization of the effects of ISRIB in wild-type animals is warranted to address this vital concern.

Of note, not all these experiments have necessarily to be performed in vivo, but at least some evidence of the underlying mechanism that connects complex I defect and ISR in the lung differentiation (considering that these links are generally cell-type specific) should be provided.

Minor Comments

1. Given that the morphological abnormalities of the lung observed in the histological analyses are one of the main phenotypes of the cNDUFS2 Ko mice, a quantification for experiments in Figs. 1f, 2e, 5f, Ext Fig2, Ext Fig 4b,d, Ext Fig9 is required (for example by counting the nuclei and alveolar area).
2. The enlarged cell size in cNDUFS2 KO lungs seen with PDPN-staining (Ext Fig 3k,m) should be quantified.
3. The authors state that in the NDUFS2 KO animals there is an accumulation of transitional cells due to impaired differentiation of AT2 to AT1 cells. Could the authors clarify why in Extended Data Fig.5c there is a bigger proportion of AT1 cells in control cells compared to NDUFS2 KO cells?
4. Why is the quantification of the TUNEL and Ki67-stainings (Ext fig 3n, o) not corresponding to the shown histological images (Ext Fig 3d,g and f, i) and why were these two analyses performed on mice of different ages (P49 and P35)?
5. For some experiments the number of biological replicates was unclear. This information should be added to each figure legend. In some experiments (for example Fig 1c, Fig 5b, Ext Fig 3n) only two biological replicates have been used. A minimum of n=3 biological replicates is required for all experiments.
6. All omics data should be provided as spreadsheets.

Referee #2 (Remarks to the Author):

Nature Review

General Comments:

This exceptionally well-written manuscript reports on an exciting and understudied aspect of lung development—the role of metabolic regulation in the differentiation from AT2 to AT1 cells. The authors do an exceptional job of integrating their findings with recently published single-cell sequencing data. The conclusions drawn—that there is a role for mitochondrial complex I in regulating normal alveologenesis and AT2 differentiation and that the mechanism driving this may be related to the integrated-stress response (and associated pathways)—are generally well-supported by the data shown. While this original work has the potential to represent a major leap forward in our thinking about a fundamental developmental process (with implications for cancer biology, adult lung injury, and in the development of other organs), there are some critical additional experiments/points of clarification that would greatly strengthen the findings as they are

presented.

Major:

1. The histology presented in the main figures and extended data would benefit from quantification of the defect using standard morphometric approaches (mean linear intercepts/radial alveolar counts, airspace volume density). Given the limitations of the number of images that can be included in a figure set, this additional rigor of quantification is necessary and also useful given the comparison made to a neonatal injury model, for which such quantification is available.
2. Considerable attention is devoted to the differences in epithelium in the cKO mice. One of the most striking features is the increased cellularity in the CKO mice (of cells that are not CD45+ immune cells). Clarifying the identity of these cells is critical to our understanding of the developmental phenotype of this model. While the authors note in Extended data figure 3 an increase in the number of SP-C+ type 2 cells, this does not fully explain the phenotype. Additional immunostaining (or RNA in situ hybridization) for other relative markers of cells in the alveolar niche (Pdgfra for fibroblasts, Car4 for alveolar capillaries) would add greatly to this description.
3. Similarly, examination of the UMAP and the stacked area plots for the single-cell sequencing data, demonstrates significant differences in the fibroblast populations between the cKO and control cells. The "fibroblast 2" population is much more enriched in the cKO mice and the "fibroblast 1 population" is more enriched in the control mice. What are these different fibroblasts and the transcriptomic features of these cell? Applying the recently published 3-axes classification system to alveolar mesenchyme (Chen, Development, 2022) or any of the recently published papers defining alveolar fibroblasts is critical. While the genetic defect is in the AEC2 cell, epithelial metabolic derangements are clearly having an effect on the other cells in the alveolar niche.
4. While the Ki67 staining by IHC is clear, it is not shown which cells are proliferating. Presumably the SP-C cells are contributing to the, but additional information is needed. This could be accomplished with Ki67 immunofluorescence in the presence of other alveolar cell markers (Ki67, CD45, Pdgfra, Pecam, Hopx) or from the single-cell sequencing data by reporting which cells have increased expression of Mki67.
5. A dot plot or heat map showing the hallmark genes that were used to label each cell type is required to allow these data to be independently interpreted.
6. A statement about data availability (and whether this dataset will be deposited to GEO with publication) is missing. In the interest of transparency, many groups also make their analysis code available (in the form of a Github link or other sharing site).
7. Was ISRIB given to control mice during development? The authors findings suggest that some ISR response is necessary during normal development and AT2 differentiation, but that exceptionally elevated ISR impairs normal differentiation. If not performed, this would be a critical control experiment.
8. How to the "pathologic" transitional cells transcriptionally compare to the Krt8+ populations identified during adult lung injury (Strunz 2020, Choi 2020, Kobayashi 2020)?
9. This developmental manipulation of metabolism has only subtle differences during the saccular and early alveolar stages, and a more prominent phenotype only detected during later alveologenesis. Some temporal-spatial validation of the critical pathways that are disrupted at earlier timepoints would be essential in determining when/where alveologenesis is disrupted. This could be accomplished by immunostaining/RNA in situ hybridization of earlier timepoints with Ki67 and markers of the ISR (counterstaining for relevant alveolar cells predicted by the single-cell sequencing data).

Minor:

1. The predictions set forth from the RNA velocity index are intriguing. Because RNA velocity analysis is predicated on the same nearest-neighbors map upon which the UMAP is based, the outcome of scVelo analysis can be biased by dimensionality-reduced data. Recent work shows that velocity analysis has the potential to become circular:
<https://www.biorxiv.org/content/10.1101/2022.06.19.494717v2>. While no analysis method is without limitations, some caution should be mentioned when interpreting these findings.

Referee #3 (Remarks to the Author):

This manuscript from Chandel and colleagues describes the discovery of a critical role for mitochondrial complex I in regulating alveolar epithelial type 1 (AT1) generation from AT2 cells. This biological process is of clear general interest from the perspective of understanding mechanisms of lung repair in the context of respiratory viruses and the COVID-19 pandemic. More generally, it is likely that the discovery reported here could be of general relevance in understanding mechanisms energetic regulation of repair and cellular development.

Moreover, to me, the very surprising and exciting finding here is that complex I doesn't appear to be critical for AT1 generation via its contribution to generating mitochondrial ATP. Instead, the authors describe an unexpected role for complex I in triggering the integrated stress response, which appears to be the major mode through which AT1 generation is repressed. This finding in particular I think will be of broad interest to the metabolism community.

Overall, the study is clearly laid out and the experiments are performed to a high standard, and I support publication based on the clear general interest. I only have a few suggestions for the authors to further bolster their conclusions.

1. The SDH vs complex I data are an important addition and provide evidence for the conclusion that complex I depletion drives ISR via NADH accumulation. Since SDH depletion will also affect other processes (succinate accumulation, etc.) it would be interesting to see this corroborated with by another intervention that modulates mitochondrial or cellular NADH directly. LbNOX, targeted to mitochondria or cytosol would test the importance of the ratio of these pools. Or perhaps, supplementation with NAD precursors, although these manipulations may be less tractable in vivo.
2. An alternative path could be to examine interventions that elevate NADH to see if they phenocopy C1 depletion, e.g. hypoxia, which would also provide an interesting physiologic angle to this newfound mechanism.
3. Some discussion/speculation in the paper is warranted as to how elevated NADH/NAD⁺ could regulate ISR since this will be an exciting area of future research.

Author Rebuttals to Initial Comments:

Response to the Referees' comments:

Referee #1 (Remarks to the Author): In the manuscript, "Mitochondrial integrated stress response controls lung epithelial cell fate during development" by Han et al., the authors generated a mouse model with a lung epithelial-specific deletion of the mitochondrial complex I subunit NDUFS2 (NDUFS2 cKO) to investigate its role in lung development. NDUFS2 cKO mice die at an early postnatal age of approximately seven weeks due to respiratory failure accompanied by abnormal alveolar morphology and fluid accumulation in the lung, as revealed by histological examination. Using bulk and single-cell RNA-sequencing combined with immune histological experiments, the authors suggest a block in differentiation of lung epithelial cells in the transitional stage between AT2-to-AT1 cells as the molecular mechanism to explain the observed phenotype. The authors also claim that the cell fate defect is caused by the mitochondrial dysfunction elicited by the loss of complex I activity and the subsequent downstream activation of the mitochondrial integrated stress response. Expressing yeast NDI1 (an enzyme capable of NAD⁺ regeneration but without proton pumping activity present in complex I) in the NDUFS2 cKO mice rescued the histological phenotype and the death of these mice pinpointing the failure of NAD⁺ regeneration as the primary cause of the death of the cNDUFS2 mice. The NDUFS2 cKO mice phenotype could also be rescued by intraperitoneal injection of ISRIB, an inhibitor of the ISR, strengthening the hypothesis of pathogenic hyperactivation of the ISR as the cause of the respiratory failure.

This study presents the exciting hypothesis that the extent of mitochondrial dysfunction and subsequent ISR activation is vital for the switching between physiological to pathological cellular differentiation in the lung. Although potentially interesting, this study suffers from major technical and conceptual limitations that undermine the strength of the authors' conclusions. In addition, the role of mitochondrial function in regulating postnatal development and the effects of OXPHOS impairment and metabolic imbalances on ISR upregulation are already well-described. Therefore, the paper lacks the novelty required for this journal unless some more mechanistic aspects are provided.

We thank the reviewer for the comments on an "exciting hypothesis" and appreciate the reviewer's suggestions to improve the manuscript. In response, we have performed a series of new experiments that directly address the reviewer's questions and comments. These new data substantially strengthen the revised manuscript. A point-by-point response follows.

A key question that is yet to be answered in the mitochondrial field is whether mitochondrial-dependent ISR signaling is pathogenic or adaptive. We would argue this distinction is dependent on the magnitude and the duration of the ISR activation. This is best illustrated by the pioneering work of Peter Walter's group in elucidating the molecular mechanisms of ISR activation and signaling. As the figure on the right shows (from Peter Walter's excellent review in *Science*: PMID: 32327570), reduced or increased ISR activation can be maladaptive. The ISR is needed for adaptation to a particular stress but the inability to resolve the ISR or its

hyperactivation can be maladaptive. Pharmacologic strategies to increase (Sephin1) or decrease the ISR (ISRIB) in a particular pathology can be used to bring the ISR to an optimal range. Note, these pharmacologic strategies have been shown to be weak activators or inhibitors and thus can restore the optimal ISR rather than ablating or hyperactivating it. In the revised manuscript, we present new RNA sequencing data to show ISRIB partially inhibits the ISR thereby restoring it to a range that allows alveolar epithelial differentiation to proceed even in the presence of mitochondrial complex I deficiency. While our study identifies a maladaptive role for excessive ISR activation during development, we hypothesize that complete inhibition of this pathway would be maladaptive. Testing this hypothesis would require more potent methods to prevent ISR activation. Indeed, there is an excellent new study published during the revision of this manuscript from Thomas Langer's group that demonstrated complete genetic ablation of OMA1-DELE1-HRI pathway, which precludes mitochondrial dependent activation of the ISR in the setting of mitochondrial complex IV deficiency in the heart, is maladaptive (PMID: 36113464).

A key set of future experiments will be to determine the signals from mitochondria that cause "optimal" activation of the ISR during successful alveolar differentiation, and to determine which downstream effect of ISR activation--the sustained decrease in protein translation or the activation of ATF4/5 or both--prevents successful differentiation in the setting of mitochondrial complex I deficiency.

Major Comments

1. The characterization of the mouse model is not satisfactory. First, the authors use *NDUSF2^{fl/-}*; *SFTPC-Cre^{+/+}* as control mice, but at least for the initial description of the mouse model (experiments shown in Fig 1) a proper control should be used (*NDUSF2^{fl/fl}*; *SFTPC-Cre^{-/-}*) to prove that the chosen control mice behave indeed like Cre-negative mice. Additionally, it is unclear why the control mice in Fig. 1a,b differ from those used in the rest of the manuscript.

We used *Ndufs2^{+/-}SFTPC-Cre* as a control to compare with *Ndufs2^{fl/-}SFTPC-Cre*, a knockout mouse. Our control has the same conditions as the cKO except for the distal lung epithelial deletion of *Ndufs2* at E10.5, and thus allows us to evaluate cKO without being confounded by potential effects of *Cre* or global *Ndufs2* heterozygosity on lung development.

In specific response to the reviewer's concern, we published a manuscript in which we reported *Ndufs2* heterozygous mice do not have any detectable phenotypes even at an advanced age (PMID: 35338200). In that manuscript, we reported lung development was normal in heterozygous animals. As shown in Fig 1c and Fig 5b, *Ndufs2^{+/-}SFTPC-Cre* lung epithelial cells show similar levels of OCR as that of *Ndufs2^{fl/fl}* lung epithelial cells and are almost indistinguishable from wild-type cells at the level of the measured transcriptome and metabolome.

To ensure that the NDUF52 KO is cell-specific, it is recommended that the authors perform an immunohistochemistry analysis for NDUF52 in lung epithelial cells. Additionally, the leakage of the Cre should be assessed by looking at the tdTomato-expression combined with specific AT1 and AT2 markers by histology.

Developmental *SFTPC-Cre* mediated recombination deletes or expresses gene alleles in all distal lung epithelial cell populations: club (secretory) cells, AT1 cells, and AT2 cells as reported by Brigid Hogan's group who developed these mice (PMID: 15716345). We

validated recombination in distal lung epithelial cells and excluded non-specific recombination in other lung cells using single-cell RNA sequencing of whole lung cell suspension. Specifically, we queried these data for expression of the gene encoding *tdTomato* (*Sftpc* lineage) and *Ndufs2*, which are induced and deleted, respectively, by *Cre*-mediated recombination in our strain. These data (Fig 3c and Extended data Fig 4d-e) provide an unbiased assessment of *Cre* recombination efficiency and specificity that is superior to immunostaining.

On a related note, it is strongly recommended that the impaired differentiation, shown only by sc-RNA-seq, is further validated (for example, by an *in vitro* differentiation assay). These control experiments will rule out the possibility that wild-type cells repopulate the regions where NDUFS2 KO cells are located. This hypothesis is corroborated by the observation that the increased cellularity in the NDUFS2 KO animals arises from hypertrophic AT2-expressing markers, which according to the authors' model, should not be NDUFS2 KO cells that express AT1 markers instead. Addressing this question is crucial to determine whether the rescue of the lung defects caused by ISRIB is caused by the repopulation of NDUFS2 KO cells or of wild-type cells (see comment 2 below).

We thank the reviewer for this excellent suggestion. It is important to note that *in vitro* versus *in vivo* metabolic requirement of any given cell type is now appreciated to be distinct by the community. As the reviewer suggests, we generated alveolar epithelial 3-D organoids, which recapitulate some aspects of alveolar development *in vitro* in a simplified system including lung epithelial cells from NDUFS2 control and NDUFS2 cKO animals cultured with identical supporting cells, wild-type fibroblasts. Our data demonstrate that loss of NDUFS2 results in smaller-sized organoids, a proxy for organoid differentiation, and that organoid size is partially rescued by ISRIB. These results suggest that the failure of alveolar epithelial differentiation in mice with epithelial-specific *Ndufs2* knockout is cell autonomous and is at least partially attributable to the activation of the ISR. Importantly, the epithelial cells used for these organoids were isolated from mice at P6, before differences between NDUFS2 cKO and control mice were detectable.

We complement the 3-D organoid culture system with the classic 2-D alveolar epithelial cell culture, where isolated AT2 cells spontaneously differentiate into cells resembling AT1 cells on 2-D plastic culture plates over days. Using RNA-seq, we confirmed that, in 2-D cultures, AT2 cells lose AT2 cell markers and express AT1 cell markers 72 hours after isolation. However, adding mitochondrial complex I inhibitor, Piericidin A, in culture media prevents AT2 cells from expressing AT1 cell markers (See Extended Data Fig 10f-h).

The age of mice used in the study differs between different experiments (for instance, some experiments were performed using 11-day-old mice, whereas others used 43-day-old mice). Similarly, several analyses were performed at different stages during the mouse embryonic and postnatal development (for example, some experiments were performed at P34, while others at P47 or P49), making conclusions difficult to draw. Although these parameters might not significantly affect the results, consistency between all experiments is required.

We thank the reviewer for this comment. Our selection of time points is driven by the observed phenotypes in the mutant strain. Many NDUFS2 cKO mice die between postnatal day 40 to 50. Thus, to avoid survivor bias, we conducted metabolomics and RNA-seq at P35 when NDUFS2 cKO mice clearly have histologic abnormalities but are still alive. To assess potential early molecular drivers of the phenotype, we conducted single-cell RNA seq at postnatal day 21 when histologic abnormalities in NDUFS2 cKO lungs start to

become obvious. Lung compliance was measured at P46-49 due to technical limitations imposed by the small tracheal size of younger mice. We performed immunoblot assays for the NDUFS2 protein at P11, the earliest time point we could obtain enough protein. We have clarified the rationale for these time points in the revised manuscript.

Finally, for metabolomic and RNA-seq analyses, the authors used both male and female mice, whereas for all other experiments, it is not clear whether they used mice of a specific gender. Given that the two genders show a difference in their metabolic and transcriptional responses (Fig.2c and Fig.4b, respectively), it would be important to test whether this affects the conclusions or at least discuss these differences appropriately.

Thank you for pointing out our omission. We used both male and female in all experiments. In the revised manuscript, we have separated the analysis by sex.

2. The authors' central claim is that the primary cause of the respiratory failure in cNDUFS2 mice is the perturbed NAD⁺ generation causing the activation of the mitoISR and subsequently the block in differentiation. There are a few issues with this model.

First, given the central role of the NAD⁺/NADH ratio for all conclusions of this paper, the authors should provide this measurement for every used mouse model (cNDUFS2, cNDUFS2/NDI1, Sdhd) and applied treatment (ISRIB). Additional measurements of NAD/NADH imbalance should also be included, such as lactate to pyruvate ratio. These critical controls will confirm that the NAD⁺/NADH ratio is affected as expected and is indeed the trigger for the observed phenotypes. In addition, the authors should show that altering NADH/NAD ratio without affecting the ETC would phenocopy the effect (using approaches proposed by Mick et al. eLife 2020, for instance, using LbNOX). It is also unclear how these NADH/NAD ratio changes can elicit the mitoISR. The authors should provide at least a minimal mechanism to corroborate this connection. For instance, is it a low asparagine level (due to a decrease in aspartate) as proposed by Mick et al. eLife 2020?

Thank you for these suggestions. We have provided the NADH/NAD⁺ ratios for the NDUFS2 cKO mouse strains and found that NDI1 and NMN supplementation normalized these ratios. We did not measure NADH/NAD⁺ ratios in SDHD cKO mice as other groups have shown that NADH/NAD⁺ ratios are not affected in these animals. In fact, complex II inhibition is the only complex within the ETC known not to increase NADH/NAD⁺. We show that NDI1 expression reduced the increased level of lactate resulting from the loss of NDUFS2 (Fig 2b). To directly address the reviewer's concern, we complemented the NDI1 rescue experiment with an experiment in which we supplemented NDUFS2 cKO mice with NMN. We provide new RNA-seq data to show the administration of either NMN or ISRIB *in vivo* ameliorated the pathologic ISR activation in lung epithelial cells in the setting of mitochondrial complex I loss (Fig 5b, Extended Data Fig 9h-i). Consistent with a critical role for the NADH/NAD⁺ ratio in the observed phenotype, NMN partially rescued the death of NDUFS2 cKO animals (Fig 5a).

An important distinction in our *in vivo* studies compared to the *in vitro* studies of Mick et al. eLIFE 2020 is that we did not observe decreases in aspartate or asparagine levels which would trigger GCN2-dependent ISR response. Moreover, in cancer cells, impairment of ETC triggers decreases in aspartate and asparagine, decreasing cell proliferation and tumor growth (PMID: 33609439; PMID: 29941933; PMID: 29941931). However, in our study, lung epithelial cell proliferation was at least preserved, and perhaps enhanced (increased Ki67 expression and increases in transcriptional markers of proliferation; see Extended Data Fig 1j, m, t, and Extended Data Fig 4a). Furthermore, in our metabolomic analysis, we

observed the levels of aspartate and asparagine between NDUFS2-cKO and control mice were similar (Extended Data Fig 3g, h). Finally, in the new experiments with organoids/2-D cultures and *Oma1* knockout cell lines (below), we supplemented the media with aspartate and asparagine.

Secondly, the authors should corroborate the presence of a mitoISR by reporting additional data beyond the transcriptional signature (changes at the protein levels of ISR markers, and potentially the expected metabolic changes, considering that they have the data). The authors should distinguish whether what they observed is triggered by the integrated stress response coming from the mitochondrial dysfunction or the ER stress, considering the current lack of additional ISR markers. They could distinguish mito- and ER stress-induced ISR by looking at the cleavage of DELE1 by western blot (like in Fessler et al., 2020; Guo et al., 2020). In addition, the authors should confirm that ISRIB reverses these parameters, and acts specifically on lung epithelial cells on NDUFS2 KO lung cells (using the markers described above). It will be vital to confirm that ISRIB restores the proportion of AT2 vs AT1 cells in the lung and that these cells are NDUFS2 KO. These experiments would exclude that any non-cell-autonomous effect is responsible for rescuing the phenotype.

Thank you for these excellent suggestions. It is important to note that we could not find a suitable DELE1 antibody to assess cleavage. In fact, the field does not have a proper antibody that detects DELE1. In the Kampmann paper, the authors state “Antibodies that failed to detect DELE1 included: Abcam ab189958, 1:500; Santa Cruz Biotech sc-515080, 1:100; Proteintech 21904-1-AP, 1:500; Biorbyt ABIN1031350, 1:500; and Invitrogen PA5-34403, 1:500”. Therefore, to address the reviewer’s concern, we used CRISPR to create an *Oma1* knockout in a mouse lung epithelial cell line with features resembling AT2 cells (MLE-12). Our results demonstrate the mitochondrial complex I inhibitor Piericidin A increases ATF4 protein abundance in an OMA-1 dependent manner (Extended Data Fig 11g-j). Note, we used uridine, methyl-pyruvate (for aspartate production), and asparagine in the culture media to allow cell proliferation. This is similar to *in vivo* data where cell proliferation and aspartate/asparagine levels are preserved. Thus, these findings implicate OMA1/DELE1 for the activation of the ISR in response to complex I inhibition in lung epithelial cells.

Finally, the mechanism behind the dose-dependent effect of ISR on cellular differentiation has not been described. The authors should clarify how the ISR can affect the expression of genes involved in differentiation and whether acute vs chronic ISR activation affects different transcriptional programs. For instance, the authors should consider that some factors downstream of complex I loss (including metabolic and transcriptional changes) restored by NDI could contribute to the phenotype. Intriguingly, both SDHD KO and NDI1 (in cNDFS2 KO) expression led to succinate accumulation. The authors should discuss the possibility that succinate accumulation is sufficient to promote alveolar development and should provide convincing evidence to exclude this hypothesis and consider the established role of this metabolite in orchestrating chromatin changes. In addition, the authors should be able to explain how the suppression of ISR could revert the differentiation phenotype in cells with mitochondrial dysfunction. Considering their model, would inhibiting the transient ISR observed in normal development be detrimental? Appropriate characterization of the effects of ISRIB in wild-type animals is warranted to address this vital concern.

Of note, not all these experiments have necessarily to be performed *in vivo*, but at least some evidence of the underlying mechanism that connects complex I defect and ISR in the lung differentiation (considering that these links are generally cell-type specific) should be provided.

The reviewer asks important questions about the interaction between the ISR and transcriptional programs necessary for the differentiation of AT2 cells. Our new data generated in 3-D organoid cultures show that complex I inhibition is sufficient to impair alveolar epithelial differentiation and the administration of ISRIB is sufficient to rescue impaired AT2 differentiation, suggesting the effects of ISR are cell autonomous. We used a mouse lung epithelial cell line to show that activation of the ISR in response to complex I inhibition was OMA1 dependent. Despite decades of careful studies, the precise mechanisms underlying the differentiation of AT2 cells into AT1 cells are incompletely understood, and it is therefore difficult to localize the interaction of the ISR with this program. We have added this to the revised text.

We provide additional data *in vivo* showing a partial rescue of the developmental phenotype in NDUFS2 cKO mice by NMN, further supporting the importance of the low NADH/NAD⁺ ratio for normal alveolar epithelial development *in vivo*. The reviewer raises an interesting point regarding succinate. NDI1 does increase succinate but we did not observe succinate accumulation with ISRIB administration or NMN treated mice (Extended Data Fig 12e-f). Thus, it is unlikely that succinate alone is sufficient to promote postnatal alveolar development. But perturbing succinate, fumarate or L-2HG would be an interesting line of future investigation with respect to lung development. We have addressed this in the revised text.

We found that NDUFS2 control mice that received ISRIB did not have detectable abnormalities in the lung and the transcriptomic and metabolomic signatures of lung epithelial cells did not change significantly in response to ISRIB. We have also observed no detrimental effects of ISRIB in adult wild-type mice in a different setting, a bleomycin-induced lung fibrosis model, where ISRIB shows no change in lung mechanics, lung collagen, or histology in young adult (4-6 months) or old (18-24 months) mice that did not receive bleomycin (PMID: 33972447). Furthermore, we found ISRIB did not impair differentiation in AT2 cells from NDUFS2 control mice in organoid culture. When interpreting these results, it is important to remember that Peter Walter has shown ISRIB is a weak inhibitor of the ISR. Hence, we have been careful to limit our interpretation of our results as suggesting a pathologic role of chronic activation of the high ISR during lung development and avoided any conclusions as to whether its activation might be necessary for lung development.

Minor Comments

1. Given that the morphological abnormalities of the lung observed in the histological analyses are one of the main phenotypes of the cNDUFS2 Ko mice, a quantification for experiments in Figs. 1f, 2e, 5f, Ext Fig2, Ext Fig 4b,d, Ext Fig9 is required (for example by counting the nuclei and alveolar area).

We have measured and quantified alveolar wall thickness in NDUFS2 cKO animals, NDUFS2 cKO/NDI1 rescue, and SDHD cKO animals.

2. The enlarged cell size in cNDUFS2 KO lungs seen with PDPN-staining (Ext Fig 3k,m) should be quantified.

We have measured and quantified alveolar wall thickness in NDUFS2 cKO animals, NDUFS2 cKO/NDI1 rescue, and SDHD cKO animals. PDPN+ AT1 cells are very large thin cells, and recent evidence suggests that these cells may cover multiple alveoli sacs by

folding their cytoplasm (PMID: 26586225, PMID: 31548395). Therefore, quantifying PDPN+ AT1 cells on 2D images will less likely provide meaningful information on the phenotype of NDUFS2 cKO lungs.

3. The authors state that in the NDUFS2 KO animals there is an accumulation of transitional cells due to impaired differentiation of AT2 to AT1 cells. Could the authors clarify why in Extended Data Fig.5c there is a bigger proportion of AT1 cells in control cells compared to NDUFS2 KO cells?

Lung digestion protocols poorly preserve alveolar type I cells, which are large, flat, and thin. As a result, AT1 cells are systematically under-sampled in all single-cell RNA-sequencing data from the murine and human lungs. Under-sampling of AT1 cells is substantial-- typically AT1 cells represent <10% of alveolar epithelial cells in single-cell RNA-sequencing data but are known to represent ~50% of alveolar epithelial cells from careful morphologic studies using EM (PMID: 7103258). By contrast, transitional cells are cuboidal and smaller than AT1 cells—much closer to an AT2 cell shape. We, therefore, interpret the finding that AT1 cells are less abundant in the control animals compared to the NDUFS2 cKO animals in RNA-seq data but more abundant *in vivo* histologically, as evidence of failed differentiation in the NDUFS2-cKO animals. Specifically, we suspect the cells transcriptionally defined as AT1 cells using single-cell RNA-sequencing are smaller and thicker in the NDUFS2 cKO animals than those in control animals and are therefore more susceptible to liberation during digestion. This conclusion is supported by histologic examination where thin membranous cells typical for AT1 cells are lacking in NDUFS2 cKO animals.

4. Why is the quantification of the TUNEL and Ki67-stainings (Ext fig 3n, o) not corresponding to the shown histological images (Ext Fig 3d,g and f, i) and why were these two analyses performed on mice of different ages (P49 and P35)?

Thank you for this question. We now have provided corresponding images that better represent the findings. As described above, we quantified P35 histology images to avoid survivor bias as many NDUFS2 cKO mice would have died by P49.

5. For some experiments the number of biological replicates was unclear. This information should be added to each figure legend. In some experiments (for example Fig 1c, Fig 5b, Ext Fig 3n) only two biological replicates have been used. A minimum of n=3 biological replicates is required for all experiments.

We have clarified the number of biological replicates and the sex of the animals in all of the figures. Note, all our experiments are based on biological replicates $n \geq 3$ with both female and male animals.

6. All omics data should be provided as spreadsheets.

We have provided spreadsheets of marker genes characterizing individual clusters and subclusters.

Referee #2 (Remarks to the Author):

General Comments:

This exceptionally well-written manuscript reports on an exciting and understudied aspect of lung development—the role of metabolic regulation in the differentiation from AT2 to AT1 cells. The authors do an exceptional job of integrating their findings with recently published single-cell sequencing data. The conclusions drawn—that there is a role for mitochondrial complex I in regulating normal alveologenesis and AT2 differentiation and that the mechanism driving this may be related to the integrated-stress response (and associated pathways)—are generally well-supported by the data shown. While this original work has the potential to represent a major leap forward in our thinking about a fundamental developmental process (with implications for cancer biology, adult lung injury, and in the development of other organs), there are some critical additional experiments/points of clarification that would greatly strengthen the findings as they are presented.

We thank the reviewer for the positive comments and helpful critiques. Our data are consistent with evolving data highlighting the importance of the ISR in alveolar epithelial development. In particular, a careful study using time series single-cell RNA sequencing and morphologic analysis from the Sucre lab identified transitional cells during normal postnatal lung development. Their data argue against a purely pathologic role of these cells, instead suggesting their transient accumulation represents a common transition in cellular development. Our data build on these findings by showing this transition can be pathologically interrupted by mitochondrial activation of the ISR. Furthermore, our new data that were generated in response to the reviewer suggest that the disruption of temporal and spatial organization between different cell types contributes to impaired postnatal lung development, supporting the model of lung development – functional lung development is a result of the appropriate temporal and spatial organization of multiple cell types.

Major:

1. The histology presented in the main figures and extended data would benefit from quantification of the defect using standard morphometric approaches (mean linear intercepts/radial alveolar counts, airspace volume density). Given the limitations of the number of images that can be included in a figure set, this additional rigor of quantification is necessary and also useful given the comparison made to a neonatal injury model, for which such quantification is available.

Thank you for this suggestion. As the main histologic abnormalities in NDUFS2 cKO lungs compared to NDUFS2 control are thickened alveolar walls with hypercellular areas, we have measured and quantified alveolar wall thickness in each genotype of animals (Fig 1g, Extended Data Fig 3d, 12a).

2. Considerable attention is devoted to the differences in epithelium in the cKO mice. One of the most striking features is the increased cellularity in the CKO mice (of cells that are not CD45+ immune cells). Clarifying the identity of these cells is critical to our understanding of the developmental phenotype of this model. While the authors note in Extended data figure 3 an increase in the number of SP-C+ type 2 cells, this does not fully explain the phenotype. Additional immunostaining (or RNA in situ hybridization) for other relative markers of cells in the alveolar niche (Pdgfra for fibroblasts, Car4 for alveolar capillaries) would add greatly to this description.

Thank you for this suggestion. We performed RNA *in situ* hybridization (RNAScope) to evaluate the hypercellularity of cKO lungs. Hypertrophic *Sftpc*+ cells are observed in the cKO lungs explaining the increased cellularity. The hypertrophic *Sftpc*+ cells in the cKO lungs tend to cluster next to each other along the alveolar walls rather than locate individually at the corner of alveolar sacs as in the control lungs. Clusters of cKO *Sftpc*+ cells are observed in the cKO lungs.

cells lose the 1:1 direct contact with *Pdgfra*⁺ fibroblasts that we observed in control animals (and have been reported by others) (Extended Data Fig 2a). Also as suggested by the reviewer (below), we went on to use single-cell RNA-sequencing to examine a population of fibroblasts characterized by increased expression of *Sfrp1*⁺ that have been reported to expand in murine models of bleomycin-induced lung injury (DOI: 10.1101/2022.07.11.499594). These fibroblasts are distinct from fibroblasts found in normal lung development. In the new analysis, we show this population of fibroblasts was expanded in *NDUFS2* cKO mice compared to *NDUFS2* control mice (Extended Data Fig 6). Also as reported by others, we expect that *Car4*⁺ endothelial cells would be located next to AT1 cells (*Sftpc* negative cells), but we observed *Car4*⁺ cells next to *Sftpc*⁺ cells in cKO lungs (Extended Data Fig 2b).

3. Similarly, examination of the UMAP and the stacked area plots for the single-cell sequencing data, demonstrates significant differences in the fibroblast populations between the cKO and control cells. The “fibroblast 2” population is much more enriched in the cKO mice and the “fibroblast 1 population” is more enriched in the control mice. What are these different fibroblasts and the transcriptomic features of these cell? Applying the recently published 3-axes classification system to alveolar mesenchyme (Chen, Development, 2022) or any of the recently published papers defining alveolar fibroblasts is critical. While the genetic defect is in the AEC2 cell, epithelial metabolic derangements are clearly having an effect on the other cells in the alveolar niche.

Thank you for this excellent comment and suggestion. Further clustering analysis with fibroblasts identified 8 distinct fibroblast cell types (Extended Data Fig 6). We observed the expansion of two fibroblast subpopulations characterized by high expression of *Sfrp1* and *Timp1* in *NDUFS2* cKO mice compared to *NDUFS2* control mice. These cells share several transcriptional features with a fibroblast subpopulation observed in lung injury models (bioRxiv, DOI: 10.1101/2022.07.11.499594). As suggested by the reviewer, these findings hint at a possible pathologic feedback loop between epithelial cells and fibroblasts when alveolar development is impaired. We have added this to our revised manuscript.

4. While the Ki67 staining by IHC is clear, it is not shown which cells are proliferating. Presumably the SP-C cells are contributing to the, but additional information is needed. This could be accomplished with Ki67 immunofluorescence in the presence of other alveolar cell markers (Ki67, CD45, *Pdgfra*, *Pecam*, *Hopx*) or from the single-cell sequencing data by reporting which cells have increased expression of *Mki67*.

Thank you for this question. We felt a more quantitative answer would come from an analysis of our single-cell RNA sequencing data. We confirmed that the overall number of alveolar epithelial cells expressing *Mki67* was higher in *NDUFS2* cKO lungs than that of *NDUFS2* control lungs, consistent with the IHC results. All types of cells including *tdTomato* (*Sftpc* lineage) positive cells have increased expression of *Mki67* in *NDUFS2* cKO lungs compared to *NDUFS2* control lungs (Extended Data Fig 4a). These results clearly show that the loss of mitochondrial complex I function in lung epithelial cells induces proliferation in AT2 cells, and we did not detect evidence of increased cell death. The findings that other cell populations also proliferate suggest this might be a global response to impaired epithelial development and explains the non-AT2 cell proliferation the reviewer astutely noted in our Ki67 images. We have added this finding in the revised manuscript.

5. A dot plot or heat map showing the hallmark genes that were used to label each cell type is required to allow these data to be independently interpreted.

Thank you. A marker gene heatmap has been added (Extended data fig 4b).

6. A statement about data availability (and whether this dataset will be deposited to GEO with publication) is missing. In the interest of transparency, many groups also make their analysis code available (in the form of a Github link or other sharing site).

Thank you. We have deposited all sequencing data at the NCBI BioProject with the following Accession IDs: PRJNA865889, PRJNA940730, PRJNA940746, PRJNA940973, PRJNA940986, and PRJNA940992. Currently, reviewers can access the data via the URLs listed below:

- PRJNA865889: <https://www.ncbi.nlm.nih.gov/bioproject/PRJNA865889>
- PRJNA940730: <https://dataview.ncbi.nlm.nih.gov/object/PRJNA940730?reviewer=738nhoasprbamdp0is4vkdr02i>
- PRJNA940746: <https://dataview.ncbi.nlm.nih.gov/object/PRJNA940746?reviewer=3jp34ibnstin0caie4afd0vkf>
- PRJNA940973: <https://dataview.ncbi.nlm.nih.gov/object/PRJNA940973?reviewer=u8ot8p7snuhbvhmfgnirookfg6>
- PRJNA940986: <https://dataview.ncbi.nlm.nih.gov/object/PRJNA940986?reviewer=8n7kia0iqb8n9ou0im3vjdjaek>
- PRJNA940992: <https://dataview.ncbi.nlm.nih.gov/object/PRJNA940992?reviewer=pa9qbufk0nfjnb6v73dptthp3>

We have shared our code at <https://github.com/MinhoLee-DGU/2023.Han.et.al.Nature>

7. Was ISRIB given to control mice during development? The authors findings suggest that some ISR response is necessary during normal development and AT2 differentiation, but that exceptionally elevated ISR impairs normal differentiation. If not performed, this would be a critical control experiment.

Thank you for this comment. Control mice did receive ISRIB and showed no premature mortality or impairment in lung development. Transcriptomic and metabolomic signatures of lung epithelial cells in control mice treated with ISRIB at P35 were not significantly different from untreated mice (Fig 5b). We have also observed no detrimental effects of ISRIB in young adult (4-6 months old) or old (18-24 months old) wild-type mice in our previously published work (PMID: 33972447). Furthermore, we found ISRIB did not impair differentiation in AT2 cells from control mice in organoid culture (Extended Data Fig 11a-f). It is important to remember that Peter Walter has shown ISRIB is a weak inhibitor of the ISR. Hence, we have been careful in the manuscript to limit our interpretation of our results as suggesting a pathologic role of chronic activation of the high ISR during lung development and avoided any conclusions as to whether its activation might be necessary for postnatal lung development.

8. How do the “pathologic” transitional cells transcriptionally compare to the Krt8+ populations identified during adult lung injury (Strunz 2020, Choi 2020, Kobayashi 2020)?

Thank you for this question. The Krt8+ intermediate epithelial cell populations in adult murine models of bleomycin-induced lung injury and murine organoids, as well as during normal and impaired postnatal lung development secondary to hyperoxia, all showed enriched ISR gene signatures compared to other epithelial cell populations. However, the levels of genes associated with ISR activation were higher in NDUFS2 cKO mice than they were in any of these conditions. These data are included in Extended Data Fig 8.

9. This developmental manipulation of metabolism has only subtle differences during the saccular and early alveolar stages, and a more prominent phenotype only detected during later alveologenesis. Some temporal-spatial validation of the critical pathways that are disrupted at earlier timepoints would be essential in determining when/where alveologenesis is disrupted. This could be accomplished by immunostaining/RNA in situ hybridization of earlier timepoints with Ki67 and markers of the ISR (counterstaining for relevant alveolar cells predicted by the single-cell sequencing data).

Thank you for this question. According to the mouse lung development atlas data (Negretti 2021), postnatal transitional epithelial cells start to appear at P7. Therefore, we performed bulk RNA-seq on lung epithelial cells isolated from P6 mice to evaluate transcriptomic signatures in NDUFS2 control and cKO mice. The ISR signature was enriched in NDUFS2 cKO lungs at P6 compared to NDUFS2 control lungs, but not as high as that in P35. Moreover, transcriptomic signatures of control and cKO lung epithelial cells at P6 were not clearly separated in PCA analysis, indicating the critical pathways are disrupted after P6 (Extended data fig 10a-e).

Minor: 1. The predictions set forth from the RNA velocity index are intriguing. Because RNA velocity analysis is predicated on the same nearest-neighbors map upon which the UMAP is based, the outcome of scVelo analysis can be biased by dimensionality-reduced data. Recent work shows that velocity analysis has the potential to become circular: <https://www.biorxiv.org/content/10.1101/2022.06.19.494717v2>. While no analysis method is without limitations, some caution should be mentioned when interpreting these findings.

We completely agree with the reviewer. We considered removing the RNA velocity analysis because of these limitations but still find the results intriguing. Accordingly, these data remain in the Supplement where we have included a discussion of the limitations.

Referee #3 (Remarks to the Author):

This manuscript from Chandel and colleagues describes the discovery of a critical role for mitochondrial complex I in regulating alveolar epithelial type 1 (AT1) generation from AT2 cells. This biological process is of clear general interest from the perspective of understanding mechanisms of lung repair in the context of respiratory viruses and the COVID-19 pandemic. More generally, it is likely that the discovery reported here could be of general relevance in understanding mechanisms energetic regulation of repair and cellular development.

Moreover, to me, the very surprising and exciting finding here is that complex I doesn't appear to be critical for AT1 generation via its contribution to generating mitochondrial ATP. Instead, the authors describe an unexpected role for complex I in triggering the integrated stress response, which appears to be the major mode through which AT1 generation is repressed. This finding in particular, I think will be of broad interest to the metabolism community.

Overall, the study is clearly laid out and the experiments are performed to a high standard, and I support publication based on the clear general interest. I only have a few suggestions for the authors to further bolster their conclusions.

We would like to thank the reviewer for the insightful comments. We share the reviewer's excitement that our findings suggest a new mechanism by which mitochondria can activate signaling pathways that control cell fate with implications for both development and disease, independent of ATP production. The reviewer has suggested important experiments, including supplementation with NAD⁺ precursors, which have substantially strengthened our conclusions.

1. The SDH vs complex I data are an important addition and provide evidence for the conclusion that complex I depletion drives ISR via NADH accumulation. Since SDH depletion will also affect other processes (succinate accumulation, etc.) it would be interesting to see this corroborated with by another intervention that modulates mitochondrial or cellular NADH directly. LbNOX, targeted to mitochondria or cytosol would test the importance of the ratio of these pools. Or perhaps, supplementation with NAD precursors, although these manipulations may be less tractable in vivo.

Thank you for this excellent suggestion. To address this comment, we complemented the NDI1 rescue experiment of NADH/NAD⁺ ratio with an experiment of postnatal supplementation of NDUFS2 cKO mice with NMN. Importantly, we provide new RNA-seq data which demonstrate the administration of either NMN or ISRIB diminishes the pathologic ISR activation in lung epithelial cells due to the loss of NDUFS2 (Fig 5b). Furthermore, NMN partially rescued the death of NDUFS2 cKO animals. Like NDI1, NMN supplementation lowered NADH/NAD⁺ ratios in lung epithelial cells while ISRIB did not, providing further support for the hypothesis that an increase in the NADH/NAD⁺ ratio acts upstream of ISR activation to impair postnatal lung development.

2. An alternative path could be to examine interventions that elevate NADH to see if they phenocopy C1 depletion, e.g. hypoxia, which would also provide an interesting physiologic angle to this newfound mechanism.

Thank you for this suggestion. Exposure to hypoxia as described by Mootha and colleagues (2016 publication in *Science*) is an interesting approach to alleviating mitochondrial dysfunction. But NDUFS2 cKO mice die of respiratory failure and subjecting them to hypoxia would be challenging and almost certainly be confounding. By contrast, in the Mootha study, the NDUFS4 mice died from unknown causes but were not reported to suffer from respiratory failure. Moreover, this group has not yet provided a mechanism to explain the impressively extended lifespan of NDUFS4 KO mice exposed to hypoxia. It seems like these mice die from immune dysregulation as suggested by the excellent study from Simon Johnson and colleagues (see JCI insight paper PMID: 35050903).

3. Some discussion/speculation in the paper is warranted as to how elevated NADH/NAD⁺ could regulate ISR since this will be an exciting area of future research.

This is an excellent suggestion. We investigated one potential mechanism, which would involve GCN2-dependent ISR activation in response to reduced levels of aspartate and asparagine levels as suggested by Christofk and Mootha papers (PMID: 33609439, PMID: 32463360). We provide evidence that *in vivo* aspartate and asparagine levels, metabolites required for proliferation, do not decrease upon loss of NDUFS2. This is consistent with the observation that we don't see cell proliferative defects, as these metabolites are linked to proliferation. Furthermore, we see an OMA1 dependency of the ISR signaling consistent with *in vivo* data from Thomas Langer's group in the heart (PMID: 36113464), even in cells cultured in aspartate and asparagine containing media.

We are bit puzzled how NADH/NAD⁺ ratio would activate the OMA1-DELE1 pathway. The original CRISPR screen paper used oligomycin or FCCP, which are known to increase or decrease mitochondrial membrane potentials, respectively. Note oligomycin but not FCCP would increase NADH/NAD⁺ ratio. We are planning follow-up experiments to this work to understand how the increased NADH/NAD⁺ ratio triggers OMA1-DELE1-HRI dependent ISR.

Reviewer Reports on the First Revision:

Referees' comments:

Referee #1 (Remarks to the Author):

The authors made a significant effort to address all of the reviewer's concerns by presenting additional data, which substantially strengthened their conclusions, and clarifications in the text. The paper has much improved and I trust this paper will be a landmark in the field.

Nevertheless, it would be important if the authors could address the following minor points before final acceptance of their work:

1. It is crucial that a NAD⁺/NADH (and possibly lactate, for consistency) measurements are provided also for the SDHD KO mice. Even though previous reports show these measurements upon SDHD KO (as the authors correctly cited in the 'response to the reviewers' and in the revised manuscript), it is important that this measurement is also performed in this specific cell system, considering the importance of this parameter for their conclusions.

2. In Fig. 1b and in Extended Data Fig. 11g-j a representative image of the western blots should be provided.

3. The method of quantification of the alveolar thickness in Fig. 1g, Extended Data Fig. 3d and Extended Data Fig. 12a is not well described. What does the "number of events" refer to? To ensure data reproducibility the authors should also provide share the script of the plugin used for the quantification.

4 For accuracy and completeness, a representative histology image of a control mouse should be added in Fig. 2e.

5 The plot in Fig. 3g is rather confusing and could be removed.

6 The rescue in the expression levels of ISR-related genes upon ISRIB and NMN treatments is not clear in the heat map shown in Fig. 5. Plotting the fold change in the expression of Atf3-6 genes upon ISRIB and NMN treatments would likely reflect better the effect of these treatments in the expression of these genes. This graph could be added to the figure along with the heatmap.

7 The alveolar thickness should be quantified for the SDHD KO mice shown in Fig. 6f.

8 The authors showed in Extended Data Fig. 3h that asparagine levels increased in the NDFS2 KO, upon activation of the mitochondrial ISR. The authors should comment on this result given that it is one important finding of the study that further strengthens their main hypothesis about the activation of ASNS in these conditions.

9 The graph in the Extended Data Fig. 4d could be simplified by replacing the violin plot with a histogram. This graph seems indeed to show a bimodal distribution of data.

Referee #3 (Remarks to the Author):

The authors have been extremely thoughtful and rigorous in their responses to the critiques from myself and the other reviewers.

In particular, the addition of the NAD modulation data, in conjunction with the ISRIB data provide

a compelling and exciting case for the model of NAD/NADH imbalance acting as the upstream mediator of retrograde signaling from mitochondria via ISR.

I see this revised manuscript as a landmark in the field of mitochondrial metabolic signaling and mitochondrial disease, and fully support publication.

Author Rebuttals to First Revision:

Referees' comments:

Referee #1 (Remarks to the Author):

The authors made a significant effort to address all of the reviewer's concerns by presenting additional data, which substantially strengthened their conclusions, and clarifications in the text. The paper has much improved and I trust this paper will be a landmark in the field.

We thank the reviewer for the positive comments.

Nevertheless, it would be important if the authors could address the following minor points before final acceptance of their work:

1. It is crucial that a NAD⁺/NADH (and possibly lactate, for consistency) measurements are provided also for the SDHD KO mice. Even though previous reports show these measurements upon SDHD KO (as the authors correctly cited in the 'response to the reviewers' and in the revised manuscript), it is important that this measurement is also performed in this specific cell system, considering the importance of this parameter for their conclusions.

We thank the reviewer for this comment and for providing appropriate context. The reviewer accurately states that inhibition of complex II does not lead to an increase in NADH/NAD⁺ ratio, as established by fundamental biochemical principles. In addition to the reports cited in our manuscript and by the reviewer, in a recent paper in eLIFE, Lucas Sullivan's lab reported that mitochondrial complex I function regenerates NAD⁺ in SDH knockout cells and concomitant inhibition of complex I enables the SDH knockout cells to adapt in this system (PMID: 36883551). Hence, like the reviewer, we would argue that maintenance of normal NADH/NAD⁺ ratios in complex II inhibited cells is settled biology. Nevertheless, to address the reviewer's comment, we measured NAD⁺, NADH, lactate, pyruvate, alpha-ketoglutarate, and succinate in frozen samples from the original mice in the manuscript. As expected, we found elevated succinate levels in lung epithelial cells from SDHD cKO mice compared to SDHD control mice, while levels of lactate (as suggested by the reviewer), pyruvate, and alpha-ketoglutarate were similar. The succinate and lactate levels are reported in the revised Fig 6c-d. Unfortunately, we were unable to detect NAD⁺ or NADH in these samples by mass-spectrometry. We suspect this resulted from our use of acetonitrile to extract metabolites from these samples, which can interfere with the detection of NAD⁺ or NADH. These differ from other samples in the manuscript, in which methanol was used for metabolite extraction with minimal interference of NAD⁺ and NADH levels. We used acetonitrile extraction for these experiments as we had fewer mice for these studies compared to the other strains and acetonitrile can extract most metabolites from fewer cells. At the time of the experiments and currently, we lack sufficient numbers of the multiple transgenic animals in our colony required to repeat these assays using methanol extraction.

2. In Fig. 1b and in Extended Data Fig. 11g-j a representative image of the western blots should be provided.

We performed immunoblots with the Protein Simple WES/Sally Sue platform (Bio-Techne, Minneapolis, MN), a capillary electrophoresis immunoassay, that provides quantitative values of the peak area for a detected protein (PMID: 26044028). We published a paper in Nature Immunology using this method to detect inflammasome activation (PMID: 35484407). We have provided spreadsheets of the peak area values of the detected proteins.

3. The method of quantification of the alveolar thickness in Fig. 1g, Extended Data Fig. 3d and Extended Data Fig. 12a is not well described. What does the “number of events” refer to? To ensure data reproducibility the authors should also provide share the script of the plugin used for the quantification.

We have modified our graphs to improve clarity, and added a detailed description to the methods section of how we quantified alveolar thickness. We analyzed 4-6 randomly selected fields of view (H&E images) from each mouse (n=4, two female and two male per genotype). Using ImageJ/Fiji software (NIH), we counted the number of pixels belonging to the respective alveolar thickness bin and the total pixel count of all alveolar septal walls in each image. The x axis of the resulting graphs are the alveolar thickness bins, and the y axis is the pixel count within the respective thickness bin normalized to the total pixel count of all alveolar septal walls in the image. We also provide a macro file with a series of ImageJ commands that we used in this analysis.

4. For accuracy and completeness, a representative histology image of a control mouse should be added in Fig. 2e.

We have added representative images of NDUFS2 control and NDUFS2 control/NDI1 lung histology in Fig 2e.

5. The plot in Fig. 3g is rather confusing and could be removed.

We have modified Fig 3g to clearly show the difference of marker gene expression by cell type and mouse genotype.

6. The rescue in the expression levels of ISR-related genes upon ISRIB and NMN treatments is not clear in the heat map shown in Fig. 5. Plotting the fold change in the expression of Atf3-6 genes upon ISRIB and NMN treatments would likely reflect better the effect of these treatments in the expression of these genes. This graph could be added to the figure along with the heatmap.

We interpret the reviewers comment as requesting statistical support for our conclusion that the expression of genes induced by the ISR differs in epithelial cells as a function of genetic and pharmacologic interventions. Since the ISR is a collective gene response rather than a response driven by any single gene, our authors with expertise in statistics suggested the reviewer’s comment is better addressed by a Gene Set Enrichment Analysis (GSEA) of the ISR signature genes. The enrichment plots comparing the effect of ISRIB or NMN on the ISR in NDUFS2 cKO animals, with statistical significance indicated, are now provided in Fig 5c-d.

7. The alveolar thickness should be quantified for the SDHD KO mice shown in Fig. 6f.

Thank you. It is provided in Extended Data Fig 12a.

8. The authors showed in Extended Data Fig. 3h that asparagine levels increased in the NDUFS2 KO, upon activation of the mitochondrial ISR. The authors should comment on this result given that it is one important finding of the study that further strengthens their main hypothesis about the activation of ASNS in these conditions.

We thank the reviewer for this comment. Indeed, the ATFs target ASNS, which is an important output of the ISR. The reviewer is also correct that asparagine levels in NDUFS2 cKO are maintained or elevated compared to NDUFS2 control mice. However, we have refrained from calling this out in the manuscript as support of our hypothesis because these differences were not statistically significant.

9. The graph in the Extended Data Fig. 4d could be simplified by replacing the violin plot with a histogram. This graph seems indeed to show a bimodal distribution of data.

We agree the data are bimodal as shown in the violin plots. As the reviewer suggested, we made density plots, but the comparison between NDUFS2 control and NDUFS2 cKO is arguably more difficult to visualize with this approach. Instead, we modified the previous violin plots to make them easier to interpret (Extended Data Fig 4d, e).

Referee #2 (Remarks to the Author):

In general, I am very satisfied with the additional data and explanations to address my concerns. They have been meticulous in attention to detail in addressing the other reviewer concerns as well. I have two minor concern that can easily be addressed with modification of the interpretation of these data, and in general feel that the novelty of this work makes it suitable for a broader audience.

We thank the reviewer for the positive comments.

1. Minor concern A: the proportion graphs in figure 3f and in extended data figure 4c the KO appear to show fewer AT2 cells in the KO. This is contrast to what is described in the rest of the data and in the histology photos, where there appears to be an increase in AT2 cells clustered together. I think it is likely that in the expansion of transitional cells, there are many truly transitional cells being labeled as AT1 cells due to significant number of overlapping marker genes. It is also possible that the expanded clusters of AT2 cells (shown in extended data 2) do not dissociate well into single cell suspension and therefore are sequenced as doublets. Some attention to these contrasting findings or a modification in the way the data are being displayed would be helpful. Generally, there is a growing appreciate in the field for decreased survival cells with flatter, irregular shapes (e.g., AT1 cells) through the sorting and processing required of single-cell seq. It is possible that the “more rounded” AT1 cells in the KO make have a technical advantage in making it through sequencing. Further, understanding the nuance in hallmark gene expression between KO and WT mice might be helpful in looking at the difference especially between the KO-AT1 cells and the WT-AT1 cells, especially since the histology of AT1 cells differs so grateful in the KO relative to controls. Modifying the dot plot in Figure 3 g to segregate by genotype (or pulling out the AT1, AT2 and transitional cells) would be helpful.

We thank the reviewer for these comments. Like the reviewer, we suspect that many of the hypertrophic AT2-marker positive cells and cuboidal AT1-marker positive cells in NDUFS2 cKO mice are in fact transitional cells. We have modified our manuscript to clarify our findings.

We agree with the reviewer that mature flattened AT1 cells are less likely to be successfully liberated from the extracellular matrix with an intact cell membrane by the tissue digestion procedure, compared to less mature, more cuboidal cells. We also agree that failed differentiation of transitional cells into mature AT1 cells in the NDUFS2 cKO mice will lead to a preponderance of cells with a more transitional phenotype clustering with AT1 cells in NDUFS2 cKO compared with NDUFS2 control mice. We were therefore excited by the reviewer’s suggestion to test the latter hypothesis. Specifically, the reviewer suggests that the expression of transitional markers should be increased and the expression of mature AT1 markers decreased in cells annotated as AT1 cells in NDUFS2 cKO mice compared to NDUFS2 control mice. Indeed, this is true. We queried our scRNAseq data and found that cells assigned to the AT1 cluster in NDUFS2 cKO mice expressed higher levels of transitional cell marker genes such as *Krt8*, *Krt18*, and *Cdkn1a*. Furthermore, the expression of *Igfbp2*, identified as a marker for mature, terminally differentiated AT1 cells (PMID: 29463737), was lower in cells annotated as AT1 cells from NDUFS2 cKO mice than those from NDUFS2 control mice. These new results have been added in Extended Data Fig 5d-h.

2. Minor concern B: the creation of two “clusters” during analysis of AT2 single-cell seq that segregate on the basis of *Lyz1* expression is well known to the field. Many investigators lump both groups of AT2 cells together for this reason.

Thank you for this suggestion. We have modified Fig 3g by combining AT2 and AT2-*Lyz1*+ into “AT2 cells”, and Transitional cells and Transitional-*Lyz1*+ into “Transitional cells”.

Referee #3 (Remarks to the Author):

The authors have been extremely thoughtful and rigorous in their responses to the critiques from myself and the other reviewers.

In particular, the addition of the NAD modulation data, in conjunction with the ISRIB data provide a compelling and exciting case for the model of NAD/NADH imbalance acting as the upstream mediator of retrograde signaling from mitochondria via ISR.

I see this revised manuscript as a landmark in the field of mitochondrial metabolic signaling and mitochondrial disease, and fully support publication.

We thank the reviewer for the positive comments.

Reviewer Reports on the Second Revision:

Referees' comments:

Referee #1 (Remarks to the Author):

The authors have satisfactorily addressed all the remaining concerns. Although NAD levels could not be measured in the SDH KO animals, the supported evidence is sufficiently strong and I support the publication of the work. Congratulations to the authors.

Referee #2 (Remarks to the Author):

In this revised manuscript, the authors have thoroughly addressed my concerns and suggestions. In looking for markers of the transitional cell population in the AT1 and AT2 cells from the knockout, they have generated data that strongly support a role for metabolism and metabolic dynamics in epithelial differentiation.

This manuscript truly represents a groundbreaking shift in our understanding of the mechanisms of epithelial differentiation and organogenesis that is likely to have a high impact in lung biology and more broadly in other organs as well.